# Active Learning with Neural Networks: Insights from Nonparametric Statistics

**Yinglun Zhu**
Department of Computer Sciences
University of Wisconsin-Madison
Madison, WI 53706
yinglun@cs.wisc.edu

**Robert Nowak**
Department of Electrical and Computer Engineering
University of Wisconsin-Madison
Madison, WI 53706
rdnowak@wisc.edu

## Abstract

Deep neural networks have great representation power, but typically require large numbers of training examples. This motivates deep active learning methods that can significantly reduce the amount of labeled training data. Empirical successes of deep active learning have been recently reported in the literature, however, rigorous label complexity guarantees of deep active learning have remained elusive. This constitutes a significant gap between theory and practice. This paper tackles this gap by providing the first near-optimal label complexity guarantees for deep active learning. The key insight is to study deep active learning from the nonparametric classification perspective. Under standard low noise conditions, we show that active learning with neural networks can provably achieve the minimax label complexity, up to disagreement coefficient and other logarithmic terms. When equipped with an abstention option, we further develop an efficient deep active learning algorithm that achieves $\mathsf{polylog}(\frac{1}{\varepsilon})$ label complexity, without any low noise assumptions. We also provide extensions of our results beyond the commonly studied Sobolev/Hölder spaces and develop label complexity guarantees for learning in Radon $\mathrm{BV}^2$ spaces, which have recently been proposed as natural function spaces associated with neural networks.

## 1 Introduction

We study active learning with neural network hypothesis classes, sometimes known as *deep active learning*. Active learning agent proceeds by selecting the most informative data points to label: The goal of active learning is to achieve the same accuracy achievable by passive learning, but with much fewer label queries (Settles, 2009; Hanneke, 2014). When the hypothesis class is a set of neural networks, the learner further benefits from the representation power of deep neural networks, which has driven the successes of passive learning in the past decade (Krizhevsky et al., 2012; LeCun et al., 2015). With these added benefits, deep active learning has become a popular research area, with empirical successes observed in many recent papers (Sener and Savarese, 2018; Ash et al., 2019; Citovsky et al., 2021; Ash et al., 2021; Kothawade et al., 2021; Emam et al., 2021; Ren et al., 2021). However, due to the difficulty of analyzing a set of neural networks, rigorous label complexity guarantees for deep active learning have remained largely elusive.

To the best of our knowledge, there are only two papers (Karzand and Nowak, 2020; Wang et al., 2021) that have made the attempts at theoretically quantifying active learning gains with neural networks. While insightful views are provided, these two works have their own limitations. The guarantees provided in Karzand and Nowak (2020) only work in the $1d$ case where data points are uniformly sampled from $[0, 1]$ and labeled by a well-seperated piece-wise constant function in a noise-free way (i.e., without any labeling noise). Wang et al. (2021) study deep active learning by

36th Conference on Neural Information Processing Systems (NeurIPS 2022).

linearizing the neural network at its random initialization and then analyzing it as a linear function; moreover, as the authors agree, their error bounds and label complexity guarantees can in fact be *vacuous* in certain cases. Thus, it's fair to say that up to now researchers have not identified cases where deep active learning are provably near minimax optimal (or even with provably non-vacuous guarantees), which constitutes a significant gap between theory and practice.

In this paper, we bridge this gap by providing the first near-optimal label complexity guarantees for deep active learning. We obtain insights from the nonparametric setting where the conditional probability (of taking a positive label) is assumed to be a smooth function (Tsybakov, 2004; Audibert and Tsybakov, 2007). Previous nonparametric active learning algorithms proceed by partitioning the action space into exponentially many sub-regions (e.g., partitioning the unit cube $[0,1]^d$ into $\varepsilon^{-d}$ sub-cubes each with volume $\varepsilon^d$), and then conducting local mean (or some higher-order statistics) estimation within each sub-region (Castro and Nowak, 2008; Minsker, 2012; Locatelli et al., 2017, 2018; Shekhar et al., 2021; Kpotufe et al., 2021). We show that, with an appropriately chosen set of neural networks that *globally* approximates the smooth regression function, one can in fact recover the minimax label complexity for active learning, up to disagreement coefficient (Hanneke, 2007, 2014) and other logarithmic factors. Our results are established by (i) identifying the "right tools" to study neural networks (ranging from approximation results (Yarotsky, 2017, 2018) to complexity measure of neural networks (Bartlett et al., 2019)), and (ii) developing novel extensions of agnostic active learning algorithms (Balcan et al., 2006; Hanneke, 2007, 2014) to work with a set of neural networks.

While matching the minimax label complexity in nonparametric active learning is existing, such minimax results scale as $\Theta(\mathrm{poly}(\frac{1}{\varepsilon}))$ (Castro and Nowak, 2008; Locatelli et al., 2017) and do not resemble what is practically observed in deep active learning: A fairly accurate neural network classifier can be obtained by training with only a few labeled data points. Inspired by recent results in *parametric* active learning with abstention (Puchkin and Zhivotovskiy, 2021; Zhu and Nowak, 2022), we develop an oracle-efficient algorithm showing that deep active learning provably achieves $\mathrm{polylog}(\frac{1}{\varepsilon})$ label complexity when equipped with an abstention option (Chow, 1970). Our algorithm not only achieves an exponential saving in label complexity (*without any low noise assumptions*), but is also highly practical: In real-world scenarios such as medical imaging, it makes more sense for the classifier to abstain from making prediction on hard examples (e.g., those that are close to the boundary), and ask medical experts to make the judgments.

## 1.1 Problem setting

Let $\mathcal{X}$ denote the instance space and $\mathcal{Y}$ denote the label space. We focus on the binary classification problem where $\mathcal{Y} := \{+1, -1\}$. The joint distribution over $\mathcal{X} \times \mathcal{Y}$ is denoted as $\mathcal{D}_{\mathcal{X}\mathcal{Y}}$. We use $\mathcal{D}_{\mathcal{X}}$ to denote the marginal distribution over the instance space $\mathcal{X}$, and use $\mathcal{D}_{\mathcal{Y}|x}$ to denote the conditional distribution of $\mathcal{Y}$ with respect to any $x \in \mathcal{X}$. We consider the standard active learning setup where $x \sim \mathcal{D}_{\mathcal{X}}$ but its label $y \sim \mathcal{D}_{\mathcal{Y}|x}$ is only observed after issuing a label query. We define $\eta(x) := \mathbb{P}_{y \sim \mathcal{D}_{\mathcal{Y}|x}}(y = +1)$ as the conditional probability of taking a positive label. The Bayes optimal classifier $h^\star$ can thus be expressed as $h^\star(x) := \mathrm{sign}(2\eta(x) - 1)$. For any classifier $h : \mathcal{X} \to \mathcal{Y}$, its (standard) error is calculated as $\mathrm{err}(h) := \mathbb{P}_{(x,y) \sim \mathcal{D}_{\mathcal{X}\mathcal{Y}}}(h(x) \neq y)$; and its (standard) excess error is defined as $\mathrm{excess}(h) := \mathrm{err}(h) - \mathrm{err}(h^\star)$. Our goal is to learn an accurate classifier with a small number of label querying.

**The nonparametric setting.** We consider the nonparametric setting where the conditional probability $\eta$ is characterized by a smooth function. Fix any $\alpha \in \mathbb{N}_+$, the *Sobolev norm* of a function $f : \mathcal{X} \to \mathbb{R}$ is defined as $\|f\|_{\mathcal{W}^{\alpha,\infty}} := \max_{\overline{\alpha}, |\overline{\alpha}| \leq \alpha} \mathrm{ess\,sup}_{x \in \mathcal{X}} |\mathsf{D}^\alpha f(x)|$, where $\alpha = (\alpha_1, \ldots, \alpha_d)$, $|\alpha| = \sum_{i=1}^d \alpha_i$ and $\mathsf{D}^\alpha f$ denotes the standard $\alpha$-th weak derivative of $f$. The unit ball in the Sobolev space is defined as $\mathcal{W}_1^{\alpha,\infty}(\mathcal{X}) := \{f : \|f\|_{\mathcal{W}^{\alpha,\infty}} \leq 1\}$. Following the convention of nonparametric active learning (Castro and Nowak, 2008; Minsker, 2012; Locatelli et al., 2017, 2018; Shekhar et al., 2021; Kpotufe et al., 2021), we assume $\mathcal{X} = [0,1]^d$ and $\eta \in \mathcal{W}_1^{\alpha,\infty}(\mathcal{X})$ (except in Section 4).

**Neural Networks.** We consider *feedforward neural networks* with Rectified Linear Unit (ReLU) activation function, which is defined as $\mathrm{ReLU}(x) := \max\{x, 0\}$. Each neural network $f_{\mathsf{dnn}} : \mathcal{X} \to \mathbb{R}$ consists of several input units (which corresponds to the covariates of $x \in \mathcal{X}$), one output unit (which corresponds to the prediction in $\mathbb{R}$), and multiple hidden computational units. Each hidden

computational unit takes inputs $\{\bar{x}_i\}_{i=1}^{N}$ (which are outputs from previous layers) and perform the computation $\mathsf{ReLU}(\sum_{i=1}^{N} w_i \bar{x}_i + b)$ with *adjustable* parameters $\{w_i\}_{i=1}^{N}$ and $b$; the output unit performs the same operation, but without the ReLU nonlinearity. We use $W$ to denote the total number of parameters of a neural network, and $L$ to denote the depth of the neural network.

## 1.2 Contributions and paper organization

Neural networks are known to be universal approximators (Cybenko, 1989; Hornik, 1991). In this paper, we argue that, in both passive and active regimes, the universal approximatability makes neural networks "universal classifiers" for classification problems: With an appropriately chosen set of neural networks, one can recover known minimax rates (up to disagreement coefficients in the active setting) in the rich nonparametric regimes.[1] We provide informal statements of our main results in the sequel, with detailed statements and associated definitions/algorithms deferred to later sections.

In Section 2, we analyze the label complexity of deep active learning under the standard Tsybakov noise condition with smoothness parameter $\beta \geq 0$ (Tsybakov, 2004). Let $\mathcal{H}_{\mathsf{dnn}}$ be an appropriately chosen set of neural network classifiers and denote $\theta_{\mathcal{H}_{\mathsf{dnn}}}(\varepsilon)$ as the disagreement coefficient (Hanneke, 2007, 2014) at level $\varepsilon$. We develop the following label complexity guarantees for deep active learning.

**Theorem 1** (Informal). *There exists an algorithm that returns a neural network classifier $\widehat{h} \in \mathcal{H}_{\mathsf{dnn}}$ with excess error $\widetilde{O}(\varepsilon)$ after querying $\widetilde{O}(\theta_{\mathcal{H}_{\mathsf{dnn}}}(\varepsilon^{\frac{\beta}{1+\beta}}) \cdot \varepsilon^{-\frac{d+2\alpha}{\alpha+\alpha\beta}})$ labels.*

The label complexity presented in Theorem 1 matches the active learning lower bound $\Omega(\varepsilon^{-\frac{d+2\alpha}{\alpha+\alpha\beta}})$ (Locatelli et al., 2017) up to the dependence on the disagreement coefficient (and other logarithmic factors). Since $\theta_{\mathcal{H}_{\mathsf{dnn}}}(\varepsilon) \leq \varepsilon^{-1}$ by definition, the label complexity presented in Theorem 1 is never worse than the passive learning rates $\widetilde{\Theta}(\varepsilon^{-\frac{d+2\alpha+\alpha\beta}{\alpha+\alpha\beta}})$ (Audibert and Tsybakov, 2007). We also discover conditions under which the disagreement coefficient with respect to a set of neural network classifiers can be properly bounded, i.e., $\theta_{\mathcal{H}_{\mathsf{dnn}}}(\varepsilon) = o(\varepsilon^{-1})$ (implying strict improvement over passive learning) and $\theta_{\mathcal{H}_{\mathsf{dnn}}}(\varepsilon) = o(1)$ (implying matching active learning lower bound).

In Section 3, we develop label complexity guarantees for deep active learning when an additional abstention option is allowed (Chow, 1970; Puchkin and Zhivotovskiy, 2021; Zhu and Nowak, 2022). Suppose a cost (e.g. $0.49$) that is marginally smaller than random guessing (which has expected cost $0.5$) is incurred whenever the classifier abstains from making a predication, we develop the following label complexity guarantees for deep active learning.

**Theorem 2** (Informal). *There exists an efficient algorithm that constructs a neural network classifier $\widehat{h}_{\mathsf{dnn}}$ with Chow's excess error $\widetilde{O}(\varepsilon)$ after querying $\mathrm{polylog}(\frac{1}{\varepsilon})$ labels.*

The above $\mathrm{polylog}(\frac{1}{\varepsilon})$ label complexity bound is achieved *without any low noise assumptions*. Such exponential label savings theoretically justify the great empirical performances of deep active learning observed in practice (e.g., in Sener and Savarese (2018)): It suffices to label a few data points to achieve a high accuracy level. Moreover, apart from an initialization step, our algorithm (Algorithm 4) developed for Theorem 2 can be *efficiently* implemented in $\widetilde{O}(\varepsilon^{-1})$ time, given a convex loss regression oracle over an appropriately chosen set of neural networks; in practice, the regression oracle can be approximated by running stochastic gradient descent.

**Technical contributions.** Besides identifying the "right tools" (ranging from approximation results (Yarotsky, 2017, 2018) to complexity analyses (Bartlett et al., 2019)) to analyze deep active learning, our theoretical guarantees are empowered by novel extensions of active learning algorithms *under neural network approximations*. In particular, we deal with approximation error in active learning under Tsybakov noise, and identify conditions that greatly relax the approximation requirement in the learning with abstention setup; we also analyze the disagreement coefficient, both classifier-based and value function-based, with a set of neural networks. These analyses together lead to our main results for deep active learning (e.g., Theorem 1 and Theorem 2). More generally, we establish a

---

[1]As a byproduct, our results also provide a new perspective on nonparametric active learning through the lens of neural network approximations. Nonparametric active learning was previously tackled through space partitioning and local estimations over exponentially many sub-regions (Castro and Nowak, 2008; Minsker, 2012; Locatelli et al., 2017, 2018; Shekhar et al., 2021; Kpotufe et al., 2021).

bridge between approximation theory and active learning; we provide these general guarantees in Appendix B (under Tsybakov noise) and Appendix D (with the abstention option), which can be of independent interests. Benefited from these generic algorithms and guarantees, in Section 4, we extend our results into learning smooth functions in the Radon $BV^2$ space (Ongie et al., 2020; Parhi and Nowak, 2021, 2022a,b; Unser, 2022), which is recently proposed as a natural space to analyze neural networks.

## 1.3 Related work

Active learning concerns about learning accurate classifiers without extensive human labeling. One of the earliest work of active learning dates back to the CAL algorithm proposed by Cohn et al. (1994), which set the cornerstone for *disagreement-based* active learning. Since then, a long line of work have been developed, either directly working with a set classifier (Balcan et al., 2006; Hanneke, 2007; Dasgupta et al., 2007; Beygelzimer et al., 2009, 2010; Huang et al., 2015; Cortes et al., 2019) or work with a set of regression functions (Krishnamurthy et al., 2017, 2019). These work mainly focus on the parametric regime (e.g., learning with a set of linear classifiers), and their label complexities rely on the boundedness of the so-called disagreement coefficient (Hanneke, 2007, 2014; Friedman, 2009). Active learning in the nonparametric regime has been analyzed in Castro and Nowak (2008); Minsker (2012); Locatelli et al. (2017, 2018); Kpotufe et al. (2021). These algorithms rely on partitioning of the input space $\mathcal{X} \subseteq [0,1]^d$ into exponentially (in dimension) many small cubes, and then conduct local mean (or some higher-order statistics) estimation within each small cube.

It is well known that, in the worst case, active learning exhibits no label complexity gains over the passive counterpart (Kääriäinen, 2006). To bypass these worst-case scenarios, active learning has been popularly analyzed under the so-called Tsybakov low noise conditions (Tsybakov, 2004). Under Tsybakov noise conditions, active learning has been shown to be strictly superior than passive learning in terms of label complexity (Castro and Nowak, 2008; Locatelli et al., 2017). Besides analyzing active learning under favorable low noise assumptions, more recently, researchers consider active learning with an abstention option and analyze its label complexity under Chow's error (Chow, 1970). In particular, Puchkin and Zhivotovskiy (2021); Zhu and Nowak (2022) develop active learning algorithms with $\mathrm{polylog}(\frac{1}{\varepsilon})$ label complexity when analyzed under Chow's excess error. Shekhar et al. (2021) study nonparametric active learning under a different notion of the Chow's excess error, and propose algorithms with $\mathrm{poly}(\frac{1}{\varepsilon})$ label complexity; their algorithms follow similar procedures of those partition-based nonparametric active learning algorithms (e.g., Minsker (2012); Locatelli et al. (2017)).

Inspired by the success of deep learning in the passive regime, active learning with neural networks has been extensively explored in recent years (Sener and Savarese, 2018; Ash et al., 2019; Citovsky et al., 2021; Ash et al., 2021; Kothawade et al., 2021; Emam et al., 2021; Ren et al., 2021). Great empirical performances are observed in these papers, however, rigorous label complexity guarantees have largely remains elusive (except in Karzand and Nowak (2020); Wang et al. (2021), with limitations discussed before). We bridge the gap between practice and theory by providing the first near-optimal label complexity guarantees for deep active learning. Our results are built upon approximation results of deep neural networks (Yarotsky, 2017, 2018; Parhi and Nowak, 2022b) and VC/pseudo dimension analyses of neural networks with given structures (Bartlett et al., 2019).

## 2 Label complexity of deep active learning

We analyze the label complexity of deep active learning in this section. We first introduce the Tsybakov noise condition in Section 2.1, and then identify the "right tools" to analyze classification problems with neural network classifiers in Section 2.2 (where we also provide passive learning guarantees). We establish our main active learning guarantees in Section 2.3.

## 2.1 Tsybakov noise condition

It is well known that active learning exhibits no label complexity gains over the passive counterpart without additional low noise assumptions (Kääriäinen, 2006). We next introduce the Tsybokov low noise condition (Tsybakov, 2004), which has been extensively analyzed in active learning literature.

**Definition 1** (Tsybakov noise). *A distribution $\mathcal{D}_{\mathcal{X}\mathcal{Y}}$ satisfies the Tsybakov noise condition with parameter $\beta \geq 0$ and a universal constant $c \geq 1$ if, $\forall \tau > 0$,*

$$\mathbb{P}_{x \sim \mathcal{D}_{\mathcal{X}}}(|\eta(x) - 1/2| \leq \tau) \leq c\tau^{\beta}.$$

The case with $\beta = 0$ corresponds to the general case *without* any low noise conditions, where no active learning algorithm can outperform the passive counterpart (Audibert and Tsybakov, 2007; Locatelli et al., 2017). We use $\mathcal{P}(\alpha, \beta)$ to denote the set of distributions satisfying: (i) the smoothness conditions introduced in Section 1.1 with parameter $\alpha > 0$; and (ii) the Tsybakov low noise condition (i.e., Definition 1) with parameter $\beta \geq 0$. We assume $\mathcal{D}_{\mathcal{X}\mathcal{Y}} \in \mathcal{P}(\alpha, \beta)$ in the rest of Section 2. As in Castro and Nowak (2008); Hanneke (2014), we assume the knowledge of noise/smoothness parameters.

## 2.2 Approximation and expressiveness of neural networks

Neural networks are known to be universal approximators (Cybenko, 1989; Hornik, 1991): For any continuous function $g : \mathcal{X} \to \mathbb{R}$ and any error tolerance $\kappa > 0$, there exists a large enough neural network $f_{\mathsf{dnn}}$ such that $\|f_{\mathsf{dnn}} - g\|_{\infty} := \sup_{x \in \mathcal{X}} |f_{\mathsf{dnn}}(x) - g(x)| \leq \kappa$. Recently, *non-asympototic* approximation rates by ReLU neural networks have been developed for smooth functions in the Sobolev space, which we restate in the following.[2]

**Theorem 3** (Yarotsky (2017)). *Fix any $\kappa > 0$. For any $f^{\star} = \eta \in \mathcal{W}_1^{\alpha,\infty}([0,1]^d)$, there exists a neural network $f_{\mathsf{dnn}}$ with $W = O(\kappa^{-\frac{d}{\alpha}} \log \frac{1}{\kappa})$ total number of parameters arranged in $L = O(\log \frac{1}{\kappa})$ layers such that $\|f_{\mathsf{dnn}} - f^{\star}\|_{\infty} \leq \kappa$.*

The architecture of the neural network $f_{\mathsf{dnn}}$ appearing in the above theorem only depends on the smooth function space $\mathcal{W}_1^{\alpha,\infty}([0,1]^d)$, but otherwise is independent of the true regression function $f^{\star}$; also see Yarotsky (2017) for details. Let $\mathcal{F}_{\mathsf{dnn}}$ denote the set of neural network *regression functions* with the same architecture. We construct a set of neural network *classifiers* by thresholding the regression function at $\frac{1}{2}$, i.e., $\mathcal{H}_{\mathsf{dnn}} := \{h_f := \mathrm{sign}(2f(x) - 1) : f \in \mathcal{F}_{\mathsf{dnn}}\}$. The next result concerns about the expressiveness of the neural network classifiers, in terms of a well-known complexity measure: the VC dimension (Vapnik and Chervonenkis, 1971).

**Theorem 4** (Bartlett et al. (2019)). *Let $\mathcal{H}_{\mathsf{dnn}}$ be a set of neural network classifiers of the same architecture and with $W$ parameters arranged in $L$ layers. We then have*

$$\Omega(WL \log(W/L)) \leq \mathrm{VCdim}(\mathcal{H}_{\mathsf{dnn}}) \leq O(WL \log(W)).$$

With these tools, we can construct a set of neural network classifiers $\mathcal{H}_{\mathsf{dnn}}$ such that (i) the best in-class classifier $\check{h} \in \mathcal{H}_{\mathsf{dnn}}$ has small excess error, and (ii) $\mathcal{H}_{\mathsf{dnn}}$ has a well-controlled VC dimension that is proportional to smooth/noise parameters. More specifically, we have the following proposition.

**Proposition 1.** *Suppose $\mathcal{D}_{\mathcal{X}\mathcal{Y}} \in \mathcal{P}(\alpha, \beta)$. One can construct a set of neural network classifier $\mathcal{H}_{\mathsf{dnn}}$ such that the following two properties hold simultaneously:*

$$\inf_{h \in \mathcal{H}_{\mathsf{dnn}}} \mathrm{err}(h) - \mathrm{err}(h^{\star}) = O(\varepsilon) \quad \text{and} \quad \mathrm{VCdim}(\mathcal{H}_{\mathsf{dnn}}) = \widetilde{O}(\varepsilon^{-\frac{d}{\alpha(1+\beta)}}).$$

With the approximation results obtained above, to learn a classifier with $O(\varepsilon)$ excess error, one only needs to focus on a set of neural networks $\mathcal{H}_{\mathsf{dnn}}$ with a well-controlled VC dimension. As a warm-up, we first analyze the label complexity of such procedure in the passive regime (with fast rates).

**Theorem 5.** *Suppose $\mathcal{D}_{\mathcal{X}\mathcal{Y}} \in \mathcal{P}(\alpha, \beta)$. Fix any $\varepsilon, \delta > 0$. Let $\mathcal{H}_{\mathsf{dnn}}$ be the set of neural network classifiers constructed in Proposition 1. With $n = \widetilde{O}(\varepsilon^{-\frac{d+2\alpha+\alpha\beta}{\alpha(1+\beta)}})$ i.i.d. sampled points, with probability at least $1 - \delta$, the empirical risk minimizer $\widehat{h} \in \mathcal{H}_{\mathsf{dnn}}$ achieves excess error $O(\varepsilon)$.*

The label complexity results obtained in Theorem 5 matches, up to logarithmic factors, the passive learning lower bound $\Omega(\varepsilon^{-\frac{d+2\alpha+\alpha\beta}{\alpha(1+\beta)}})$ established in Audibert and Tsybakov (2007), indicating that our proposed learning procedure *with a set of neural networks* is near minimax optimal.[3]

---

[2]As in Yarotsky (2017), we hide constants that are potentially $\alpha$-dependent and $d$-dependent into the Big-Oh notation.

[3]Similar passive learning guarantees have been developed with different tools and analyses, e.g., see results in Kim et al. (2021).

## 2.3 Deep active learning and guarantees

The passive learning procedure presented in the previous section treats every data point equally, i.e., it requests the label of every data point. Active learning reduces the label complexity by only querying labels of data points that are "more important". We present deep active learning results in this section. Our algorithm (Algorithm 1) is inspired by RobustCAL (Balcan et al., 2006; Hanneke, 2007, 2014) and the seminal CAL algorithm (Cohn et al., 1994); we call our algorithm NeuralCAL to emphasize that it works with a set of neural networks.

For any accuracy level $\varepsilon > 0$, NeuralCAL first initialize a set of neural network classifiers $\mathcal{H}_0 := \mathcal{H}_{\mathsf{dnn}}$ such that (i) the best in-class classifier $\check{h} := \arg\min_{h \in \mathcal{H}_{\mathsf{dnn}}} \mathrm{err}(h)$ has excess error at most $O(\varepsilon)$, and (ii) the VC dimension of $\mathcal{H}_{\mathsf{dnn}}$ is upper bounded by $\widetilde{O}(\varepsilon^{-\frac{d}{\alpha(1+\beta)}})$ (see Section 2.2 for more details). NeuralCAL then runs in epochs of geometrically increasing lengths. At the beginning of epoch $m$, based on previously *labeled* data points, NeuralCAL updates a set of active classifier $\mathcal{H}_m$ such that, with high probability, the best classifier $\check{h}$ remains *uneliminated*. Within each epoch $m$, NeuralCAL only queries the label $y$ of a data point $x$ if it lies in the *region of disagreement* with respect to the current active set of classifier $\mathcal{H}_m$, i.e., $\mathsf{DIS}(\mathcal{H}_m) := \{x \in \mathcal{X} : \exists h_1, h_2 \in \mathcal{H}_m \text{ s.t. } h_1(x) \neq h_2(x)\}$. NeuralCAL returns any classifier $\widehat{h} \in \mathcal{H}_m$ that remains uneliminated after $M - 1$ epoch.

---

**Algorithm 1** NeuralCAL

---

**Input:** Accuracy level $\varepsilon \in (0, 1)$, confidence level $\delta \in (0, 1)$.
1: Let $\mathcal{H}_{\mathsf{dnn}}$ be a set of neural networks classifiers constructed in Proposition 1.
2: Define $T := \varepsilon^{-\frac{2+\beta}{1+\beta}} \cdot \mathrm{VCdim}(\mathcal{H}_{\mathsf{dnn}})$, $M := \lceil \log_2 T \rceil$, $\tau_m := 2^m$ for $m \geq 1$ and $\tau_0 := 0$.
3: Define $\rho_m := O\left( \left( \frac{\mathrm{VCdim}(\mathcal{H}_{\mathsf{dnn}}) \cdot \log(\tau_{m-1}) \cdot \log(M/\delta)}{\tau_{m-1}} \right)^{\frac{1+\beta}{2+\beta}} \right)$ for $m \geq 2$ and $\rho_1 := 1$.
4: Define $\widehat{R}_m(h) := \sum_{t=1}^{\tau_{m-1}} Q_t \mathbb{1}(h(x_t) \neq y_t)$ with the convention that $\sum_{t=1}^0 \ldots = 0$.
5: Initialize $\mathcal{H}_0 := \mathcal{H}_{\mathsf{dnn}}$.
6: **for** epoch $m = 1, 2, \ldots, M$ **do**
7:      Update active set $\mathcal{H}_m := \left\{ h \in \mathcal{H}_{m-1} : \widehat{R}_m(h) \leq \inf_{h \in \mathcal{H}_{m-1}} \widehat{R}_m(h) + \tau_{m-1} \cdot \rho_m \right\}$
8:      **if** epoch $m = M$ **then**
9:          **Return** any classifier $\widehat{h} \in \mathcal{H}_M$.
10:      **for** time $t = \tau_{m-1} + 1, \ldots, \tau_m$ **do**
11:          Observe $x_t \sim \mathcal{D}_{\mathcal{X}}$. Set $Q_t := \mathbb{1}(x_t \in \mathsf{DIS}(\mathcal{H}_m))$.
12:          **if** $Q_t = 1$ **then**
13:              Query the label $y_t$ of $x_t$.

---

Since NeuralCAL only queries labels of data points lying in the region of disagreement, its label complexity should intuitively be related to how fast the region of disagreement shrinks. More formally, the rate of collapse of the (probability measure of) region of disagreement is captured by the *(classifier-based) disagreement coefficient* (Hanneke, 2007, 2014), which we introduce next.

**Definition 2** (Classifier-based disagreement coefficient). *For any $\varepsilon_0$ and classifier $h \in \mathcal{H}$, the classifier-based disagreement coefficient of $h$ is defined as*

$$\theta_{\mathcal{H},h}(\varepsilon_0) := \sup_{\varepsilon > \varepsilon_0} \frac{\mathbb{P}_{x \sim \mathcal{D}_{\mathcal{X}}}(\mathsf{DIS}(\mathcal{B}_{\mathcal{H}}(h, \varepsilon)))}{\varepsilon} \vee 1,$$

*where $\mathcal{B}_{\mathcal{H}}(h, \varepsilon) := \{g \in \mathcal{H} : \mathbb{P}(x \in \mathcal{X} : g(x) \neq h(x)) \leq \varepsilon\}$. We also define $\theta_{\mathcal{H}}(\varepsilon_0) := \sup_{h \in \mathcal{H}} \theta_{\mathcal{H},h}(\varepsilon_0)$.*

The guarantees of NeuralCAL follows from a more general analysis of RobustCAL under approximation. In particular, to achieve fast rates (under Tsybakov noise), previous analysis of RobustCAL requires that the Bayes classifier is in the class (or a Bernstein condition for every $h \in \mathcal{H}$) (Hanneke, 2014). These requirements are stronger compared to what we have in the case with neural network approximations. Our analysis extends the understanding of RobustCAL under approximation. We defer such general analysis to Appendix B, and present the following guarantees.

**Theorem 6.** *Suppose $\mathcal{D}_{\mathcal{X}\mathcal{Y}} \in \mathcal{P}(\alpha, \beta)$. Fix any $\varepsilon, \delta > 0$. With probability at least $1 - \delta$, Algorithm 1 returns a classifier $\widehat{h} \in \mathcal{H}_{\mathsf{dnn}}$ with excess error $\widetilde{O}(\varepsilon)$ after querying $\widetilde{O}(\theta_{\mathcal{H}_{\mathsf{dnn}}}(\varepsilon^{\frac{\beta}{1+\beta}}) \cdot \varepsilon^{-\frac{d+2\alpha}{\alpha+\alpha\beta}})$ labels.*

We next discuss in detail the label complexity of deep active learning proved in Theorem 6.

- Ignoring the dependence on disagreement coefficient, the label complexity appearing in Theorem 6 matches, up to logarithmic factors, the lower bound $\Omega(\varepsilon^{-\frac{d+2\alpha}{\alpha+\alpha\beta}})$ for active learning (Locatelli et al., 2017). At the same time, the label complexity appearing in Theorem 6 is *never worse* than the passive counterpart (i.e., $\widetilde{\Theta}(\varepsilon^{-\frac{d+2\alpha+\alpha\beta}{\alpha(1+\beta)}})$) since $\theta_{\mathcal{H}_{\mathsf{dnn}}}(\varepsilon^{\frac{\beta}{1+\beta}}) \leq \varepsilon^{-\frac{\beta}{1+\beta}}$.

- We also identify cases when $\theta_{\mathcal{H}_{\mathsf{dnn}}}(\varepsilon^{\frac{\beta}{1+\beta}}) = o(\varepsilon^{-\frac{\beta}{1+\beta}})$, indicating *strict* improvement over passive learning (e.g., when $\mathcal{D}_{\mathcal{X}}$ is supported on countably many data points), and when $\theta_{\mathcal{H}_{\mathsf{dnn}}}(\varepsilon^{\frac{\beta}{1+\beta}}) = O(1)$, indicating matching the minimax active lower bound (e.g., when $\mathcal{D}_{\mathcal{X}\mathcal{Y}}$ satisfies conditions such as *decomposibility* defined in Definition 4. See Appendix C.2 for detailed discussion).[4]

Our algorithm and theorems lead to the following results, which could benefit both deep active learning and nonparametric learning communities.

- **Near minimax optimal label complexity for deep active learning.** While empirical successes of deep active learning have been observed, rigorous label complexity analysis remains elusive except for two attempts made in Karzand and Nowak (2020); Wang et al. (2021). The guarantees provided in Karzand and Nowak (2020) only work in very special cases (i.e., data uniformly sampled from $[0, 1]$ and labeled by well-separated piece-constant functions in a noise-free way). Wang et al. (2021) study deep active learning in the NTK regime by linearizing the neural network at its random initialization and analyzing it as a linear function; moreover, as the authors agree, their error bounds and label complexity guarantees are *vacuous* in certain cases. On the other hand, our guarantees are minimax optimal, up to disagreement coefficient and other logarithmic factors, which bridge the gap between theory and practice in deep active learning.

- **New perspective on nonparametric learning.** Nonparametric learning of smooth functions have been mainly approached by partitioning-based methods (Tsybakov, 2004; Audibert and Tsybakov, 2007; Castro and Nowak, 2008; Minsker, 2012; Locatelli et al., 2017, 2018; Kpotufe et al., 2021) : Partition the unit cube $[0, 1]^d$ into exponentially (in dimension) many sub-cubes and conduct local mean estimation within each sub-cube (which additionally requires a strictly stronger membership querying oracle). Our results show that, in both passive and active settings, one can learn *globally* with a set of neural networks and achieve near minimax optimal label complexities.

## 3 Deep active learning with abstention: Exponential speedups

While the theoretical guarantees provided in Section 2 are near minimax optimal, the label complexity scales as $\mathrm{poly}(\frac{1}{\varepsilon})$, which doesn't match the great empirical performance observed in deep active learning. In this section, we fill in this gap by leveraging the idea of abstention and provide a deep active learning algorithm that achieves exponential label savings. We introduce the concepts of abstention and Chow's excess error in Section 3.1, and provide our label complexity guarantees in Section 3.2.

### 3.1 Active learning without low noise conditions

The previous section analyzes active learning under Tsybakov noise, which has been extensively studied in the literature since Castro and Nowak (2008). More recently, promising results are observed in active learning under Chow's excess error, but otherwise *without any low noise assumption* (Puchkin and Zhivotovskiy, 2021; Zhu and Nowak, 2022). We introduce this setting in the following.

**Abstention and Chow's error (Chow, 1970).** We consider classifier of the form $\widehat{h} : \mathcal{X} \to \mathcal{Y} \cup \{\bot\}$ where $\bot$ denotes the action of abstention. For any fixed $0 < \gamma < \frac{1}{2}$, the Chow's error is defined as

$$\mathrm{err}_\gamma(\widehat{h}) := \mathbb{P}_{(x,y)\sim\mathcal{D}_{\mathcal{X}\mathcal{Y}}}(\widehat{h}(x) \neq y, \widehat{h}(x) \neq \bot) + (1/2 - \gamma) \cdot \mathbb{P}_{(x,y)\sim\mathcal{D}_{\mathcal{X}\mathcal{Y}}}(\widehat{h}(x) = \bot).$$

---

[4]We remark that disagreement coefficient is usually bounded/analyzed under additional assumptions on $\mathcal{D}_{\mathcal{X}\mathcal{Y}}$, even for simple cases with a set of linear classifiers (Friedman, 2009; Hanneke, 2014). The label complexity guarantees of partition-based nonparametric active algorithms (e.g., Castro and Nowak (2008)) do not depend on the disagreement coefficient, but they are analyzed under stronger assumptions, e.g., they require the strictly stronger membership querying oracle. See Wang (2011) for a discussion. We left a comprehensive analysis of the disagreement coefficient with a set of neural network classifiers for future work.

The parameter $\gamma$ can be chosen as a small constant, e.g., $\gamma = 0.01$, to avoid excessive abstention: The price of abstention is only marginally smaller than random guess (which incurs cost $0.5$). The *Chow's excess error* is then defined as $\text{excess}_\gamma(\widehat{h}) := \text{err}_\gamma(\widehat{h}) - \text{err}(h^\star)$ (Puchkin and Zhivotovskiy, 2021).

At a high level, analyzing with Chow's excess error allows slackness in predications of hard examples (e.g., data points whose $\eta(x)$ is close to $\frac{1}{2}$) by leveraging the power of abstention. Puchkin and Zhivotovskiy (2021); Zhu and Nowak (2022) show that $\text{polylog}(\frac{1}{\varepsilon})$ is always achievable in the *parametric* settings. We generalize their results to the *nonparametric* setting and analyze active learning with a set of neural networks.

## 3.2 Exponential speedups with abstention

In this section, we work with a set of neural network *regression functions* $\mathcal{F}_{\text{dnn}} : \mathcal{X} \to [0,1]$ (that approximates $\eta$) and then *construct* classifiers $h : \mathcal{X} \to \mathcal{Y} \cup \{\bot\}$ *with an additional abstention action*. To work with a set of regression functions $\mathcal{F}_{\text{dnn}}$, we analyze its "complexity" from the lenses of *pseudo dimension* $\text{Pdim}(\mathcal{F}_{\text{dnn}})$ (Pollard, 1984; Haussler, 1989, 1995) and *value function disagreement coefficient* $\theta^{\text{val}}_{\mathcal{F}_{\text{dnn}}}(\iota)$ (for some $\iota > 0$) (Foster et al., 2020). We defer detailed definitions of these complexity measures to Appendix D.1.

---

**Algorithm 2** NeuralCAL++

---

**Input:** Accuracy level $\varepsilon \in (0,1)$, confidence level $\delta \in (0,1)$, abstention parameter $\gamma \in (0, 1/2)$.

1: Let $\mathcal{F}_{\text{dnn}}$ be a set of neural network regression functions obtained by (i) applying Theorem 3 with an appropriate approximation level $\kappa$ (which satisfies $\frac{1}{\kappa} = \text{poly}(\frac{1}{\gamma})\,\text{polylog}(\frac{1}{\varepsilon\gamma})$), and (ii) applying a preprocessing step on the set of neural networks obtained from step (i). See Appendix E for details.

2: Define $T := \frac{\theta^{\text{val}}_{\mathcal{F}_{\text{dnn}}}(\gamma/4)\cdot\text{Pdim}(\mathcal{F}_{\text{dnn}})}{\varepsilon\gamma}$, $M := \lceil \log_2 T \rceil$, and $C_\delta := O(\text{Pdim}(\mathcal{F}_{\text{dnn}}) \cdot \log(T/\delta))$.

3: Define $\tau_m := 2^m$ for $m \geq 1$, $\tau_0 := 0$, and $\beta_m := 3(M - m + 1)C_\delta$.

4: Define $\widehat{R}_m(f) := \sum_{t=1}^{\tau_{m-1}} Q_t(\widehat{f}(x_t) - y_t)^2$ with the convention that $\sum_{t=1}^{0} \ldots = 0$.

5: **for** epoch $m = 1, 2, \ldots, M$ **do**

6:     Get $\widehat{f}_m := \arg\min_{f \in \mathcal{F}_{\text{dnn}}} \sum_{t=1}^{\tau_{m-1}} Q_t(f(x_t) - y_t)^2$.

7:     (Implicitely) Construct active set $\mathcal{F}_m := \left\{ f \in \mathcal{F}_{\text{dnn}} : \widehat{R}_m(f) \leq \widehat{R}_m(\widehat{f}_m) + \beta_m \right\}$.

8:     Construct classifier $\widehat{h}_m : \mathcal{X} \to \{+1, -1, \bot\}$ as

$$\widehat{h}_m(x) := \begin{cases} \bot, & \text{if } [\text{lcb}(x; \mathcal{F}_m) - \frac{\gamma}{4}, \text{ucb}(x; \mathcal{F}_m) + \frac{\gamma}{4}] \subseteq \left[\frac{1}{2} - \gamma, \frac{1}{2} + \gamma\right]; \\ \text{sign}(2\widehat{f}_m(x) - 1), & \text{o.w.} \end{cases}$$

    and query function $g_m(x) := \mathbb{1}\left(\frac{1}{2} \in \left(\text{lcb}(x; \mathcal{F}_m) - \frac{\gamma}{4}, \text{ucb}(x; \mathcal{F}_m) + \frac{\gamma}{4}\right)\right) \cdot \mathbb{1}(\widehat{h}_m(x) \neq \bot)$.

9:     **if** epoch $m = M$ **then**

10:         **Return** classifier $\widehat{h}_M$.

11:     **for** time $t = \tau_{m-1} + 1, \ldots, \tau_m$ **do**

12:         Observe $x_t \sim \mathcal{D}_{\mathcal{X}}$. Set $Q_t := g_m(x_t)$.

13:         **if** $Q_t = 1$ **then**

14:             Query the label $y_t$ of $x_t$.

---

We now present NeuralCAL++ (Algorithm 2), a deep active learning algorithm that leverages the power of abstention. NeuralCAL++ first initialize a set of set of neural network regression functions $\mathcal{F}_{\text{dnn}}$ by applying a preprocessing step on top of the set of regression functions obtained from Theorem 3 with a carefully chosen approximation level $\kappa$. The preprocessing step mainly contains two actions: (1) clipping $f_{\text{dnn}} : \mathcal{X} \to \mathbb{R}$ into $\check{f}_{\text{dnn}} : \mathcal{X} \to [0,1]$ (since we obviously have $\eta(x) \in [0,1]$); and (2) filtering out $f_{\text{dnn}} \in \mathcal{F}_{\text{dnn}}$ that are clearly not a good approximation of $\eta$. After initialization, NeuralCAL++ runs in epochs of geometrically increasing lengths. At the beginning of epoch $m \in [M]$, NeuralCAL++ (implicitly) constructs an active set of regression functions $\mathcal{F}_m$ that are "close" to the true conditional probability $\eta$. For any $x \sim \mathcal{D}_{\mathcal{X}}$, NeuralCAL++ constructs a lower bound $\text{lcb}(x; \mathcal{F}_m) := \inf_{f \in \mathcal{F}_m} f(x)$ and an upper bound $\text{ucb}(x; \mathcal{F}_m) := \sup_{f \in \mathcal{F}_m} f(x)$ as a confidence range of $\eta(x)$ (based on $\mathcal{F}_m$). An empirical classifier with an abstention option

$\widehat{h}_m : \mathcal{X} \to \{+1, -1, \perp\}$ and a query function $g_m : \mathcal{X} \to \{0, 1\}$ are then constructed based on the confidence range (and the abstention parameter $\gamma$). For any time step $t$ within epoch $m$, NeuralCAL++ queries the label of the observed data point $x_t$ if and only if $Q_t := g_m(x_t) = 1$. NeuralCAL++ returns $\widehat{h}_M$ as the learned classifier.

NeuralCAL++ is adapted from the algorithm developed in Zhu and Nowak (2022), but with novel extensions. In particular, the algorithm presented in Zhu and Nowak (2022) requires the existence of a $\bar{f} \in \mathcal{F}$ such that $\|\bar{f} - \eta\|_\infty \le \varepsilon$ (to achieve $\varepsilon$ Chow's excess error), Such an approximation requirement directly leads to $\mathrm{poly}(\frac{1}{\varepsilon})$ label complexity *in the nonparametric setting*, which is unacceptable. The initialization step of NeuralCAL++ (line 1) is carefully chosen to ensure that $\mathrm{Pdim}(\mathcal{F}_{\mathsf{dnn}}), \theta^{\mathrm{val}}_{\mathcal{F}_{\mathsf{dnn}}}(\frac{\gamma}{4}) = \mathrm{poly}(\frac{1}{\varepsilon}) \cdot \mathrm{polylog}(\frac{1}{\varepsilon})$; together with a sharper analysis of concentration results, these conditions help us derive the following deep active learning guarantees (also see Appendix D for a more general guarantee).

**Theorem 7.** *Fix any $\varepsilon, \delta, \gamma > 0$. With probability at least $1 - \delta$, Algorithm 2 (with an appropriate initialization at line 1) returns a classifier $\widehat{h}$ with Chow's excess error $\widetilde{O}(\varepsilon)$ after querying $\mathrm{poly}(\frac{1}{\gamma}) \cdot \mathrm{polylog}(\frac{1}{\varepsilon\delta})$ labels.*

We discuss two important aspects of Algorithm 2/Theorem 7 in the following, i.e., exponential savings and computational efficiency. We defer more detailed discussions to Appendix F.1.

- **Exponential speedups.** Theorem 7 shows that, equipped with an abstention option, deep active learning enjoys $\mathrm{polylog}(\frac{1}{\varepsilon})$ label complexity. This provides theoretical justifications for great empirical results of deep active learning observed in practice. Moreover, Algorithm 2 outputs a classifier that abstains *properly*, i.e., it abstains only if abstention is the optimal choice; such a property further implies $\mathrm{polylog}(\frac{1}{\varepsilon})$ label complexity under *standard* excess error and Massart noise (Massart and Nédélec, 2006).

- **Computational efficiency.** Suppose one can efficiently implement a (weighted) square loss regression oracle over the *initialized* set of neural networks $\mathcal{F}_{\mathsf{dnn}}$: Given any set $\mathcal{S}$ of weighted examples $(w, x, y) \in \mathbb{R}_+ \times \mathcal{X} \times \mathcal{Y}$ as input, the regression oracle outputs $\widehat{f}_{\mathsf{dnn}} := \arg\min_{f \in \mathcal{F}_{\mathsf{dnn}}} \sum_{(w,x,y) \in \mathcal{S}} w(f(x) - y)^2$ .[5] Algorithm 2 can then be *efficiently* implemented with $\mathrm{poly}(\frac{1}{\gamma}) \cdot \frac{1}{\varepsilon}$ oracle calls.

While the label complexity obtained in Theorem 7 has desired dependence on $\mathrm{polylog}(\frac{1}{\varepsilon})$, its dependence on $\gamma$ can be of order $\gamma^{-\mathrm{poly}(d)}$. Our next result shows that, however, such dependence is unavoidable even in the case of learning a single ReLU function.

**Theorem 8.** *Fix any $\gamma \in (0, 1/8)$. For any accuracy level $\varepsilon$ sufficiently small, there exists a problem instance such that (1) $\eta \in \mathcal{W}_1^{1,\infty}(\mathcal{X})$ and is of the form $\eta(x) := \mathsf{ReLU}(\langle w, x \rangle + a) + b$; and (2) for any active learning algorithm, it takes at least $\gamma^{-\Omega(d)}$ labels to identify an $\varepsilon$-optimal classifier, for either standard excess error or Chow's excess error (with parameter $\gamma$).*

## 4 Extensions

Previous results are developed in the commonly studied Sobolev/Hölder spaces. Our techniques, however, are generic and can be adapted to other function spaces, given neural network approximation results. In this section, we provide extensions of our results to the Radon $\mathsf{BV}^2$ space, which was recently proposed as the natural function space associated with ReLU neural networks (Ongie et al., 2020; Parhi and Nowak, 2021, 2022a,b; Unser, 2022).[6]

**The Radon $\mathsf{BV}^2$ space.** The Radon $\mathsf{BV}^2$ unit ball over domain $\mathcal{X}$ is defined as $\mathscr{R}\,\mathsf{BV}_1^2(\mathcal{X}) := \{f : \|f\|_{\mathscr{R}\,\mathsf{BV}^2(\mathcal{X})} \le 1\}$, where $\|f\|_{\mathscr{R}\,\mathsf{BV}^2(\mathcal{X})}$ denotes the Radon $\mathsf{BV}^2$ norm of $f$ over domain $\mathcal{X}$.[7] Following Parhi and Nowak (2022b), we assume $\mathcal{X} = \{x \in \mathbb{R}^d : \|x\|_2 \le 1\}$ and $\eta \in \mathscr{R}\,\mathsf{BV}_1^2(\mathcal{X})$.

---

[5]In practice, one can approximate this oracle by running stochastic gradient descent.

[6]Other extensions are also possible given neural network approximation results, e.g., recent results established in Lu et al. (2021).

[7]We provide more mathematical backgrounds and associated definitions in Appendix G.

The Radon $\mathsf{BV}^2$ space naturally contains neural networks of the form $f_{\mathsf{dnn}}(x) = \sum_{k=1}^{K} v_i \cdot \mathsf{ReLU}(w_i^\top x + b_i)$. On the contrary, such $f_{\mathsf{dnn}}$ doesn't lie in any Sobolev space of order $\alpha \geq 2$ (since $f_{\mathsf{dnn}}$ doesn't have second order *weak* derivative). Thus, if $\eta$ takes the form of the aforementioned neural network (e.g., $\eta = f_{\mathsf{dnn}}$), approximating $\eta$ up to $\kappa$ from a Sobolev perspective requires $\widetilde{O}(\kappa^{-d})$ total parameters, which suffers from the curse of dimensionality. On the other side, however, such bad dependence on dimensionality goes away when approximating from a Radon $\mathsf{BV}^2$ perspective, as shown in the following theorem.

**Theorem 9** (Parhi and Nowak (2022b)). *Fix any $\kappa > 0$. For any $f^\star \in \mathscr{R}\,\mathsf{BV}_1^2(\mathcal{X})$, there exists a one-hidden layer neural network $f_{\mathsf{dnn}}$ of width $K = O(\kappa^{-\frac{2d}{d+3}})$ such that $\|f^\star - f_{\mathsf{dnn}}\|_\infty \leq \kappa$.*

Equipped with this approximation result, we provide the active learning guarantees for learning a smooth function within the Radon $\mathsf{BV}^2$ unit ball as follows.

**Theorem 10.** *Suppose $\eta \in \mathscr{R}\,\mathsf{BV}_1^2(\mathcal{X})$ and the Tsybakov noise condition is satisfied with parameter $\beta \geq 0$. Fix any $\varepsilon, \delta > 0$. There exists an algorithm such that, with probability at least $1 - \delta$, it learns a classifier $\widehat{h} \in \mathcal{H}_{\mathsf{dnn}}$ with excess error $\widetilde{O}(\varepsilon)$ after querying $\widetilde{O}(\theta_{\mathcal{H}_{\mathsf{dnn}}}(\varepsilon^{\frac{\beta}{1+\beta}}) \cdot \varepsilon^{-\frac{4d+6}{(1+\beta)(d+3)}})$ labels.*

Compared to the label complexity obtained in Theorem 6, the label complexity obtained in the above theorem doesn't suffer from the curse of dimensionality: For $d$ large enough, the above label complexity scales as $\varepsilon^{-O(1)}$ yet label complexity in Theorem 6 scales as $\varepsilon^{-O(d)}$. Active learning guarantees under Chow's excess error in the Radon $\mathsf{BV}^2$ space are similar to results presented in Theorem 7, and are thus deferred to Appendix G.

## 5    Discussion

We provide the first near-optimal deep active learning guarantees, under both standard excess error and Chow's excess error. Our results are powered by generic algorithms and analyses developed for active learning that bridge approximation guarantees into label complexity guarantees. We outline some natural directions for future research below.

- **Disagreement coefficients for neural networks.** While we have provided some results regarding the disagreement coefficients for neural networks, we believe a comprehensive investigation on this topic is needed. For instance, can we discover more general settings where the classifier-based disagreement coefficient can be upper bounded by $O(1)$? It is also interesting to explore sharper analyses on the value function disagreement coefficient.

- **Adaptivity in deep active learning.** Our current results are established with the knowledge of some problem-dependent parameters, e.g., the smoothness parameters regarding the function spaces and the noise levels. It will be interesting to see if one can develop algorithms that can automatically adapt to unknown parameters, e.g., by leveraging techniques developed in Locatelli et al. (2017, 2018).

## Acknowledgments and Disclosure of Funding

The authors would like to thank Rahul Parhi for many helpful discussions regarding his papers. We also would like to thank anonymous reviewers for their constructive comments. This work is partially supported by NSF grant 1934612 and AFOSR grant FA9550-18-1-0166.

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
