}}$ with $W = O(\varepsilon^{-\frac{d}{\alpha(1+\beta)}} \log \frac{1}{\varepsilon})$ total parameters arranged in $L = O(\log \frac{1}{\varepsilon})$ layers. According to Theorem 4, we know

$$\mathrm{VCdim}(\mathcal{H}_{\mathsf{dnn}}) = O(\varepsilon^{-\frac{d}{\alpha(1+\beta)}} \cdot \log^2(\varepsilon^{-1})) = \widetilde{O}(\varepsilon^{-\frac{d}{\alpha(1+\beta)}}).$$

We now show that there exists a classifier $\overline{h} \in \mathcal{H}_{\mathsf{dnn}}$ with small excess error. Let $\overline{h} = h_{\overline{f}}$ be the classifier such that $\|\overline{f} - \eta\|_\infty \leq \kappa$. We can see that

$$\begin{aligned}
\mathsf{excess}(\overline{h}) &= \mathbb{E}\big[\mathbb{1}(\overline{h}(x) \neq y) - \mathbb{1}(h^\star(x) \neq y)\big] \\
&= \mathbb{E}\big[|2\eta(x) - 1| \cdot \mathbb{1}(\overline{h}(x) \neq h^\star(x))\big] \\
&\leq 2\kappa \cdot \mathbb{P}_{x \sim \mathcal{D}_{\mathcal{X}}}(x \in \mathcal{X} : |\eta(x) - 1/2| \leq \kappa) \\
&= O(\kappa^{1+\beta}) \\
&= O(\varepsilon),
\end{aligned}$$

where the third line follows from the fact that $\overline{h}$ and $h^\star$ disagrees only within region $\{x \in \mathcal{X} : |\eta(x) - 1/2| \leq \kappa\}$ and the incurred error is at most $2\kappa$ on each disagreed data point. The fourth line follows from the Tsybakov noise condition and the last line follows from the selection of $\kappa$.   $\square$

Before proving Theorem 5, we first recall the excess error guarantee for empirical risk minimization under Tsybakov noise condition.

**Theorem 11** (Boucheron et al. (2005)). *Suppose $\mathcal{D}_{\mathcal{X}\mathcal{Y}}$ satisfies Tsybakov noise condition with parameter $\beta \geq 0$. Consider a datatset $D_n = \{(x_i, y_i)\}_{i=1}^n$ of $n$ points i.i.d. sampled from $\mathcal{D}_{\mathcal{X}\mathcal{Y}}$. Let $\widehat{h} \in \mathcal{H}$ be the empirical risk minimizer on $D_n$. For any constant $\rho > 0$, we have*

$$\mathsf{err}(\widehat{h}) - \min_{h \in \mathcal{H}} \mathsf{err}(h)$$

$$\leq \rho \cdot (\min_{h \in \mathcal{H}} \mathsf{err}(h) - \mathsf{err}(h^\star)) + O\left(\frac{(1+\rho)^2}{\rho} \cdot \left(\frac{\mathrm{VCdim}(\mathcal{H}) \cdot \log n}{n}\right)^{\frac{1+\beta}{2+\beta}} + \frac{\log \delta^{-1}}{n}\right),$$

*with probability at least $1 - \delta$.*

**Theorem 5.** *Suppose $\mathcal{D}_{\mathcal{X}\mathcal{Y}} \in \mathcal{P}(\alpha, \beta)$. Fix any $\varepsilon, \delta > 0$. Let $\mathcal{H}_{\mathsf{dnn}}$ be the set of neural network classifiers constructed in Proposition 1. With $n = \widetilde{O}(\varepsilon^{-\frac{d+2\alpha+\alpha\beta}{\alpha(1+\beta)}})$ i.i.d. sampled points, with probability at least $1 - \delta$, the empirical risk minimizer $\widehat{h} \in \mathcal{H}_{\mathsf{dnn}}$ achieves excess error $O(\varepsilon)$.*

*Proof.* Proposition 1 certifies $\min_{h \in \mathcal{H}_{\mathsf{dnn}}} \mathsf{err}(h) - \mathsf{err}(h^\star) = O(\varepsilon)$ and $\mathrm{VCdim}(\mathcal{H}_{\mathsf{dnn}}) = O\left(\varepsilon^{-\frac{d}{\alpha(1+\beta)}} \cdot \log^2(\varepsilon^{-1})\right)$. Take $\rho = 1$ in Theorem 11, leads to

$$\mathsf{err}(\widehat{h}) - \mathsf{err}(h^\star) \leq O\left(\varepsilon + \left(\varepsilon^{-\frac{d}{\alpha(1+\beta)}} \cdot \log^2(\varepsilon^{-1}) \cdot \frac{\log n}{n}\right)^{\frac{1+\beta}{2+\beta}} + \frac{\log \delta^{-1}}{n}\right),$$

Taking $n = O(\varepsilon^{-\frac{d+2\alpha+\alpha\beta}{\alpha(1+\beta)}} \cdot \log(\varepsilon^{-1}) + \varepsilon^{-1} \cdot \log(\delta^{-1})) = \widetilde{O}(\varepsilon^{-\frac{d+2\alpha+\alpha\beta}{\alpha(1+\beta)}})$ thus ensures that $\mathsf{err}(\widehat{h}) - \mathsf{err}(h^\star) = O(\varepsilon)$.   $\square$

# B  Generic version of Algorithm 1 and its guarantees

We present Algorithm 3 below, a generic version of Algorithm 1 that doesn't require the approximating classifiers to be neural networks. The guarantees of Algorithm 3 are provided in Theorem 12, which is proved in Appendix B.2 based on supporting lemmas provided in Appendix B.1.

---

**Algorithm 3** RobustCAL with Approximation

---

**Input:** Accuracy level $\varepsilon \in (0, 1)$, confidence level $\delta \in (0, 1)$.

1: Let $\mathcal{H}$ be a set of approximating classifiers such that $\inf_{h \in \mathcal{H}} \mathrm{err}(h) - \mathrm{err}(h^\star) = O(\varepsilon)$.

2: Define $T := \varepsilon^{-\frac{2+\beta}{1+\beta}} \cdot \mathrm{VCdim}(\mathcal{H})$, $M := \lceil \log_2 T \rceil$, $\tau_m := 2^m$ for $m \geq 1$ and $\tau_0 := 0$.

3: Define $\rho_m := O\left( \left( \frac{\mathrm{VCdim}(\mathcal{H}) \cdot \log(\tau_{m-1}) \cdot \log(M/\delta)}{\tau_{m-1}} \right)^{\frac{1+\beta}{2+\beta}} \right)$ for $m \geq 2$ and $\rho_1 := 1$.

4: Define $\widehat{R}_m(h) := \sum_{t=1}^{\tau_{m-1}} Q_t \mathbb{1}(h(x_t) \neq y_t)$ with the convention that $\sum_{t=1}^{0} \ldots = 0$.

5: Initialize $\mathcal{H}_0 := \mathcal{H}$.

6: **for** epoch $m = 1, 2, \ldots, M$ **do**

7:    Update active set $\mathcal{H}_m := \left\{ h \in \mathcal{H}_{m-1} : \widehat{R}_m(h) \leq \inf_{h \in \mathcal{H}_{m-1}} \widehat{R}_m(h) + \tau_{m-1} \cdot \rho_m \right\}$

8:    **if** epoch $m = M$ **then**

9:       **Return** any classifier $\widehat{h} \in \mathcal{H}_M$.

10:    **for** time $t = \tau_{m-1} + 1, \ldots, \tau_m$ **do**

11:       Observe $x_t \sim \mathcal{D}_{\mathcal{X}}$. Set $Q_t := \mathbb{1}(x_t \in \mathsf{DIS}(\mathcal{H}_m))$.

12:       **if** $Q_t = 1$ **then**

13:          Query the label $y_t$ of $x_t$.

---

We provide guarantees for Algorithm 3, and then specialize them to the settings with neural network approximation, i.e., in Theorem 6 and Theorem 10. As discussed before, our analysis is based on the analysis RobustCAL, but with novel extensions in removing the requirements that the Bayes classifier is in the class (or a Bernstein condition for every $h \in \mathcal{H}$).

**Theorem 12.** *Fix $\varepsilon, \delta > 0$. With probability at least $1 - \delta$, Algorithm 3 returns a classifier $\widehat{h} \in \mathcal{H}$ with excess error $\widetilde{O}(\varepsilon)$ after querying*

$$\widetilde{O}\left( \theta_{\mathcal{H}}(\varepsilon^{\frac{\beta}{1+\beta}}) \cdot \varepsilon^{-\frac{2}{1+\beta}} \cdot \mathrm{VCdim}(\mathcal{H}) \right)$$

*labels.*

## B.1  Supporting lemmas

We first recall that Tsybakov noise condition leads to the so-called Bernstein condition (with respect to Bayes classifier $h^\star$).

**Lemma 1** (Tsybakov (2004))**.** *Suppose $\mathcal{D}_{\mathcal{X}\mathcal{Y}}$ satisfies the Tsybakov noise condition with parameter $\beta \geq 0$, then there exists an universal constant $c' > 0$ such that we have*

$$\mathbb{P}_{x \sim \mathcal{D}_{\mathcal{X}}}(h(x) \neq h^\star(x)) \leq c'(\mathrm{err}(h) - \mathrm{err}(h^\star))^{\frac{\beta}{1+\beta}}$$

*for any $h : \mathcal{X} \to \mathcal{Y}$.*

We next present a lemma in the passive learning setting, which will later be incorporated into the active learning setting. We first define some notations. Suppose $D_n = \{(x_i, y_i)\}_{i=1}^{n}$ are $n$ i.i.d. data points drawn from $\mathcal{D}_{\mathcal{X}\mathcal{Y}}$. For any $h : \mathcal{X} \to \mathcal{Y}$, we denote $\overline{R}_n(h) := \sum_{i=1}^{n} \mathbb{1}(h(x_i) \neq y_i)$ as the empirical error of $h$ over dataset $D_n$. We clearly have $\mathbb{E}[\overline{R}_n(h)] = n \cdot \mathrm{err}(h)$ by i.i.d. assumption.

**Lemma 2.** *Fix $\varepsilon, \overline{\delta} > 0$. Suppose $\mathcal{D}_{\mathcal{X}\mathcal{Y}}$ satisfies Tsybakov noise condition with parameter $\beta \geq 0$ and $\mathrm{err}(\check{h}) - \mathrm{err}(h^\star) = O(\varepsilon)$, where $\check{h} = \arg \max_{h \in \mathcal{H}} \mathrm{err}(h)$ and $h^\star$ is the Bayes classifier. Let $D_n = \{(x_i, y_i)\}_{i=1}^{n}$ be a set of $n$ i.i.d. data points drawn from $\mathcal{D}_{\mathcal{X}\mathcal{Y}}$. If $\beta > 0$, suppose $n$ satisfies*

$$n \leq \varepsilon^{-\frac{2+\beta}{1+\beta}} \cdot \mathrm{VCdim}(\mathcal{H})^{\frac{2+2\beta}{\beta}} \cdot \log(\overline{\delta}^{-1}) \cdot (\log n)^{\frac{2+2\beta}{\beta}}.$$

*With probability at least $1 - \bar{\delta}$, we have the following inequalities hold:*

$$n \cdot (\mathrm{err}(h) - \mathrm{err}(h^\star)) \leq 2 \cdot (\overline{R}_n(h) - \overline{R}_n(\check{h})) + n \cdot \rho(n, \bar{\delta}), \quad \forall h \in \mathcal{H}, \tag{1}$$

$$\overline{R}_n(\check{h}) - \min_{h \in \mathcal{H}} \overline{R}_n(h) \leq n \cdot \rho(n, \bar{\delta}), \tag{2}$$

*where $\rho(n, \bar{\delta}) = C \cdot \left( \left( \frac{\mathrm{VCdim}(\mathcal{H}) \cdot \log n \cdot \log \bar{\delta}^{-1}}{n} \right)^{\frac{1+\beta}{2+\beta}} + \varepsilon \right)$ with a universal constant $C > 0$.*[8]

*Proof.* Denote $\overline{\mathcal{H}} := \mathcal{H} \cup \{h^\star\}$. We know that $\mathrm{VCdim}(\overline{\mathcal{H}}) \leq \mathrm{VCdim}(\mathcal{H}) + 1 = O(\mathrm{VCdim}(\mathcal{H}))$. From Lemma 1, we know Bernstein condition is satisfied with respect to $\overline{\mathcal{H}}$ and $h^\star \in \overline{\mathcal{H}}$. Invoking Lemma 3.1 in Hanneke (2014), with probability at least $1 - \frac{\bar{\delta}}{2}$, $\forall h \in \overline{\mathcal{H}}$, we have

$$n \cdot (\mathrm{err}(h) - \mathrm{err}(h^\star)) \leq \max\big\{ 2 \cdot (\overline{R}_n(h) - \overline{R}_n(h^\star)), n \cdot \bar{\rho}(n, \bar{\delta}) \big\}, \tag{3}$$

$$\overline{R}_n(h) - \min_{h \in \overline{\mathcal{H}}} \overline{R}_n(h) \leq \max\big\{ 2n \cdot (\mathrm{err}(h) - \mathrm{err}(h^\star)), n \cdot \bar{\rho}(n, \bar{\delta}) \big\}, \tag{4}$$

where $\bar{\rho}(n, \bar{\delta}) = O\left( \left( \frac{\mathrm{VCdim}(\overline{\mathcal{H}}) \cdot \log n + \log \bar{\delta}^{-1}}{n} \right)^{\frac{1+\beta}{2+\beta}} \right) = O\left( \left( \frac{\mathrm{VCdim}(\mathcal{H}) \cdot \log n \cdot \log \bar{\delta}^{-1}}{n} \right)^{\frac{1+\beta}{2+\beta}} \right)$.

Eq. (2) follows by taking $h = \check{h}$ in Eq. (4) and noticing that

$$\overline{R}_h(\check{h}) - \min_{h \in \mathcal{H}} \overline{R}_n(h) \leq \overline{R}_n(\check{h}) - \min_{h \in \overline{\mathcal{H}}} \overline{R}_n(h)$$

$$\leq \max\big\{ 2n \cdot O(\varepsilon), n \cdot \bar{\rho}(n, \bar{\delta}) \big\},$$

where we use the assumption that $\mathrm{err}(\check{h}) - \mathrm{err}(h^\star) = O(\varepsilon)$.

To derive Eq. (1), we first notice that applying Eq. (3) for any $h \in \mathcal{H}$, we have

$$n \cdot (\mathrm{err}(h) - \mathrm{err}(h^\star)) \leq 2 \cdot (\overline{R}_n(h) - \overline{R}_n(\check{h}) + \overline{R}_n(\check{h}) - \overline{R}_n(h^\star)) + n \cdot \bar{\rho}(n, \bar{\delta}).$$

We next only need to upper bound $\overline{R}_n(\check{h}) - \overline{R}_n(h^\star)$, and show that it is order-wise smaller than $n \cdot \rho(n, \bar{\delta})$. We consider random variable $g_i := \mathbb{1}(\check{h}(x_i) \neq y_i) - \mathbb{1}(h^\star(x_i) \neq y_i)$. We have

$$\mathbb{V}(g_i) \leq \mathbb{E}[g_i^2]$$

$$= \mathbb{E}[\mathbb{1}(\check{h}(x_i) \neq h^\star(x_i)]$$

$$= O\left( \varepsilon^{\frac{\beta}{1+\beta}} \right),$$

where the last line follows from Lemma 1 and the assumption that $\mathrm{err}(\check{h}) - \mathrm{err}(h^\star) = O(\varepsilon)$. Denote $g = \frac{1}{n} \sum_{i=1}^{n} g_i = \frac{1}{n} (\overline{R}_n(\check{h}) - \overline{R}_n(h^\star))$, and notice that $\mathbb{E}[g] = \mathrm{err}(\check{h}) - \mathrm{err}(h^\star)$. Applying Bernstein inequality on $-g$, with probability at least $1 - \frac{\bar{\delta}}{2}$, we have

$$g - \mathbb{E}[g] \leq O\left( \left( \frac{\varepsilon^{\frac{\beta}{1+\beta}} \log \bar{\delta}^{-1}}{n} \right)^{\frac{1}{2}} + \frac{\log \bar{\delta}^{-1}}{n} \right),$$

which further leads to

$$\overline{R}_n(\check{h}) - \overline{R}_n(h^\star) \leq n \cdot O\left( \varepsilon + \left( \frac{\varepsilon^{\frac{\beta}{1+\beta}} \log \bar{\delta}^{-1}}{n} \right)^{\frac{1}{2}} + \frac{\log \bar{\delta}^{-1}}{n} \right).$$

The RHS is order-wise smaller than $\rho_n$ when $\beta = 0$. We consider the case when $\beta > 0$ next. Since $\log(\bar{\delta}^{-1})/n$ is clearly a lower-order term compared to $\rho_n$, we only need to show that $\left( \frac{\varepsilon^{\frac{\beta}{1+\beta}} \log \bar{\delta}^{-1}}{n} \right)^{\frac{1}{2}}$

---

is order-wise smaller than $\rho_n$. We can easily check that

[8]The logarithmic factors in this bound might be further optimized. We don't focus on optimizing logarithmic factors.

$$\left(\frac{\varepsilon^{\frac{\beta}{1+\beta}} \log \bar{\delta}^{-1}}{n}\right)^{\frac{1}{2}} \leq \left(\frac{\text{VCdim}(\mathcal{H}) \cdot \log n \cdot \log \bar{\delta}^{-1}}{n}\right)^{\frac{1+\beta}{2+\beta}}$$

whenever $n$ satisfies the following condition

$$n \leq \varepsilon^{-\frac{2+\beta}{1+\beta}} \cdot \text{VCdim}(\mathcal{H})^{\frac{2+2\beta}{\beta}} \cdot \log(\bar{\delta}^{-1}) \cdot (\log n)^{\frac{2+2\beta}{\beta}}.$$

$\square$

We denote $\check{h} = \arg\min_{h \in \mathcal{H}} \text{err}(h)$. By assumption of Theorem 12, we have $\text{err}(\check{h}) - \text{err}(h^\star) = O(\varepsilon)$. For any $h \in \mathcal{H}$, we also use the shorthand $\bar{R}_m(h) = \bar{R}_{\tau_{m-1}}(h) := \sum_{t=1}^{\tau_{m-1}} \mathbb{1}(h(x_t) \neq y_t)$. Note that $\bar{R}_m$ is only used in analysis since some $y_t$ are not observable.

**Lemma 3.** *With probability at least* $1 - \frac{\delta}{2}$*, the following holds true for all epochs* $m \in [M]$:

1. $\check{h} \in \mathcal{H}_m$.

2. $\text{err}(h) - \text{err}(h^\star) \leq 3\rho_m, \forall h \in \mathcal{H}_m$.

*Proof.* For each $m = 2, 3, \ldots, M$, we invoke Lemma 2 with $n = \tau_{m-1}$ and $\bar{\delta} = \delta/2M$, which guarantees that

$$\tau_{m-1} \cdot (\text{err}(h) - \text{err}(h^\star)) \leq 2 \cdot (\bar{R}_m(h) - \bar{R}_m(\check{h})) + \tau_{m-1} \cdot \rho_m, \quad \forall h \in \mathcal{H}, \tag{5}$$

$$\bar{R}_m(\check{h}) - \min_{h \in \mathcal{H}} \bar{R}_m(h) \leq \tau_{m-1} \cdot \rho_m. \tag{6}$$

Note that the choice $T$ chosen in Algorithm 3 clearly satisfies the requirement needed (for $n = \tau_{m-1}$) in Lemma 2 when $\beta > 0$; and ensures that the second term in $\rho(\tau_{m-1}, \delta/2M)$ (i.e., $\varepsilon$, see Lemma 2 for definition of $\rho(\tau_{m-1}, \delta/2M)$) is a lower-order term compared to the first term.

We use $\mathcal{E}$ to denote the good event where Eq. (5) and Eq. (6) hold true across $m = 2, 3, \ldots, M$. This good event happens with probability at least $1 - \frac{\delta}{2}$. We analyze under $\mathcal{E}$ in the following.

We prove Lemma 3 through induction. The statements clearly hold true for $m = 1$. Suppose the statements hold true up to epoch $m$, we next prove the correctness for epoch $m + 1$.

We know that $\check{h} \in \mathcal{H}_m$ by assumption. Based on the querying criteria of Algorithm 3, we know that

$$\widehat{R}_{m+1}(\check{h}) - \widehat{R}_{m+1}(h) = \bar{R}_{m+1}(\check{h}) - \bar{R}_{m+1}(h), \quad \forall h \in \mathcal{H}_m \tag{7}$$

From Eq. (6), we also have

$$\bar{R}_{m+1}(\check{h}) - \min_{h \in \mathcal{H}_m} R_{m+1}(h) \leq \bar{R}_{m+1}(\check{h}) - \min_{h \in \mathcal{H}} R_{m+1}(h)$$

$$\leq \tau_m \cdot \rho_{m+1}.$$

Combining the above two inequalities shows that

$$\widehat{R}_{m+1}(\check{h}) - \widehat{R}_{m+1}(h) \leq \tau_m \cdot \rho_{m+1},$$

implying that $\check{h} \in \mathcal{H}_{m+1}$ (due to the construction of $\mathcal{H}_{m+1}$ in Algorithm 3).

Based on Eq. (7), the construction $\mathcal{H}_{m+1}$ and the fact that $\check{h} \in \mathcal{H}_m$, we know that, for any $h \in \mathcal{H}_{m+1} \subseteq \mathcal{H}_m$,

$$\bar{R}_{m+1}(h) - \bar{R}_{m+1}(\check{h}) = \widehat{R}_{m+1}(h) - \widehat{R}_{m+1}(\check{h})$$

$$\leq \widehat{R}_{m+1}(h) - \min_{h \in \mathcal{H}_m} \widehat{R}_{m+1}(h)$$

$$\leq \tau_m \cdot \rho_{m+1}.$$

Plugging the above inequality into Eq. (5) (at epoch $m + 1$) leads to $\text{err}(h) - \text{err}(h^\star) \leq 3\rho_{m+1}$ for any $h \in \mathcal{H}_{m+1}$. We thus prove the desired statements at epoch $m + 1$. $\square$

## B.2   Proof of Theorem 12

**Theorem 12.** *Fix $\varepsilon, \delta > 0$. With probability at least $1 - \delta$, Algorithm 3 returns a classifier $\widehat{h} \in \mathcal{H}$ with excess error $\widetilde{O}(\varepsilon)$ after querying*

$$\widetilde{O}\left(\theta_{\mathcal{H}}(\varepsilon^{\frac{\beta}{1+\beta}}) \cdot \varepsilon^{-\frac{2}{1+\beta}} \cdot \mathrm{VCdim}(\mathcal{H})\right)$$

*labels.*

*Proof.* Based on Lemma 3, we know that, with probability at least $1 - \frac{\delta}{2}$, we have

$$
\begin{aligned}
\mathrm{err}(\widehat{h}) - \mathrm{err}(h^\star) &\leq 3\rho_M \\
&= O\left(\left(\frac{\mathrm{VCdim}(\mathcal{H}) \cdot \log(\tau_{M-1}) \cdot \log(M/\delta)}{\tau_{M-1}}\right)^{\frac{1+\beta}{2+\beta}}\right) \\
&= \widetilde{O}(\varepsilon),
\end{aligned}
$$

where we use the definition of $T$ and $\tau_M$.

We next analyze the label complexity of Algorithm 3. Since Algorithm 3 stops and the beginning at epoch $M$, we only need to calculated the label complexity in the first $M - 1$ epochs. We have

$$
\begin{aligned}
\sum_{t=1}^{\tau_{M-1}} Q_t &= \sum_{m=1}^{M-1} (\tau_m - \tau_{m-1}) \cdot \mathbb{1}(x_t \in \mathsf{DIS}(\mathcal{H}_m)) \\
&\leq \sum_{m=1}^{M-1} (\tau_m - \tau_{m-1}) \cdot \mathbb{1}\left(x_t \in \mathsf{DIS}(\mathcal{B}_{\mathcal{H}}(h^\star, c'(3\rho_m)^{\frac{\beta}{1+\beta}}))\right),
\end{aligned}
$$

where on the last line we use the facts (1) $\mathrm{err}(h) - \mathrm{err}(h^\star) \leq 3\rho_m, \forall h \in \mathcal{H}_m$ from Lemma 3; and (2) $\mathbb{P}(x : h(x) \neq h^\star(x)) \leq c'(\mathrm{err}(h) - \mathrm{err}(h^\star))^{\frac{\beta}{1+\beta}}$ from Lemma 2 (with the same constant $c'$). Suppose $\mathrm{err}(\check{h}) - \mathrm{err}(h^\star) = c''\varepsilon$ (with another universal constant $c''$ by assumption). Applying Lemma 2 on $\check{h}$ leads to the fact that $h^\star \in \mathcal{B}_{\mathcal{H}}(\check{h}, c'(c''\varepsilon)^{\frac{\beta}{1+\beta}})$. Since $\mathbb{P}(x : h(x) \neq \check{h}(x)) \leq \mathbb{P}(x : h(x) \neq h^\star(x)) + \mathbb{P}(x : h^\star(x) \neq \check{h}(x))$, we further have

$$
\sum_{t=1}^{\tau_{M-1}} Q_t \leq \sum_{m=1}^{M-1} (\tau_m - \tau_{m-1}) \cdot \mathbb{1}\left(x_t \in \mathsf{DIS}(\mathcal{B}_{\mathcal{H}}(\check{h}, \bar{c} \cdot \rho_m^{\frac{\beta}{1+\beta}}))\right),
$$

with a universal constant $\bar{c} > 0$. Noticing that the RHS is a sum of independent Bernoulli random variables, applying a Bernstein-type bound (e.g., Lemma 5), on a good event $\mathcal{E}'$ that happens with probability at least $1 - \frac{\delta}{2}$, we have

$$
\begin{aligned}
\sum_{t=1}^{\tau_{M-1}} Q_t &\leq 2 \sum_{m=1}^{M-1} (\tau_m - \tau_{m-1}) \cdot \mathbb{P}\left(x \in \mathsf{DIS}(\mathcal{B}_{\mathcal{H}}(\check{h}, \bar{c} \cdot \rho_m^{\frac{\beta}{1+\beta}}))\right) + 4\log(4/\delta) \\
&\leq 2 \sum_{m=2}^{M-1} \tau_{m-1} \cdot \theta_{\mathcal{H}, \check{h}}\left(\bar{c} \cdot \rho_m^{\frac{\beta}{1+\beta}}\right) \cdot \bar{c} \cdot \rho_m^{\frac{\beta}{1+\beta}} + 4\log(4/\delta) + 4 \\
&\leq 2M \cdot \theta_{\mathcal{H}, \check{h}}\left(\bar{c} \cdot \rho_M^{\frac{\beta}{1+\beta}}\right) \cdot \left(\bar{c} \cdot \tau_{M-1} \cdot \rho_M^{\frac{\beta}{1+\beta}}\right) + 4\log(4/\delta) + 4,
\end{aligned}
$$

where the second using the definition of disagreement coefficient; and the last line follows from the fact that $\rho_m$ is non-increasing and $\tau_{m-1} \cdot \rho_m$ is increasing. Basic algebra and basic properties of the disagreement coefficient (i.e., Theorem 7.1 and Corollary 7.2 in Hanneke (2014)) shows that

$$
\sum_{t=1}^{\tau_{M-1}} Q_t \leq \widetilde{O}\left(\theta_{\mathcal{H}}(\varepsilon^{\frac{\beta}{1+\beta}}) \cdot \varepsilon^{-\frac{2}{1+\beta}} \cdot \mathrm{VCdim}(\mathcal{H})\right),
$$

under event $\mathcal{E} \cap \mathcal{E}'$, which happen with probability at least $1 - \delta$. $\qquad\square$

# C  Omitted details for Section 2.3

We prove Theorem 6 in Appendix C.1 and discuss the disagreement coefficient in Appendix C.2.

## C.1  Proof of Theorem 6

**Theorem 6.** *Suppose $\mathcal{D}_{\mathcal{X}\mathcal{Y}} \in \mathcal{P}(\alpha, \beta)$. Fix any $\varepsilon, \delta > 0$. With probability at least $1 - \delta$, Algorithm 1 returns a classifier $\widehat{h} \in \mathcal{H}_{\mathsf{dnn}}$ with excess error $\widetilde{O}(\varepsilon)$ after querying $\widetilde{O}(\theta_{\mathcal{H}_{\mathsf{dnn}}}(\varepsilon^{\frac{\beta}{1+\beta}}) \cdot \varepsilon^{-\frac{d+2\alpha}{\alpha+\alpha\beta}})$ labels.*

*Proof.* Construct $\mathcal{H}_{\mathsf{dnn}}$ based on Proposition 1 such that $\min_{h \in \mathcal{H}_{\mathsf{dnn}}} \mathrm{err}(h) - \mathrm{err}(h^{\star}) = O(\varepsilon)$ and $\mathrm{VCdim}(\mathcal{H}_{\mathsf{dnn}}) = \widetilde{O}(\varepsilon^{-\frac{d}{\alpha(1+\beta)}})$. Taking such $\mathcal{H}_{\mathsf{dnn}}$ into Theorem 12 leads to the desired result. $\qquad\square$

## C.2  Discussion on disagreement coefficient in Theorem 6

We discuss cases when the (classifier-based) disagreement coefficient with respect to a set of neural networks is well-bounded. As mentioned before, even for simple classifiers such as linear functions, the disagreement coefficient has been analyzed under additional assumptions (Friedman, 2009; Hanneke, 2014). In this section, we analyze the disagreement coefficient for a set of neural networks under additional assumptions on $\mathcal{D}_{\mathcal{X}\mathcal{Y}}$ and $\mathcal{H}_{\mathsf{dnn}}$ (assumptions on $\mathcal{H}_{\mathsf{dnn}}$ can be implemented via proper preprocessing steps). We leave a more comprehensive investigation of the disagreement coefficient for future work.

The first case is when $\mathcal{D}_{\mathcal{X}}$ is supported on countably many data points. The following result show strict improvement over passive learning.

**Definition 3** (Disagreement core). *For any hypothesis class $\mathcal{H}$ and classifier $h$, the disagreement core of $h$ with respect to $\mathcal{H}$ under $\mathcal{D}_{\mathcal{X}\mathcal{Y}}$ is defined as*

$$\partial_{\mathcal{H}} h := \lim_{r \to 0} \mathsf{DIS}(\mathcal{B}_{\mathcal{H}}(h, r)). \tag{8}$$

**Proposition 2** (Lemma 7.12 and Theorem 7.14 in Hanneke (2014)). *For any hypothesis class $\mathcal{H}$ and classifier $h$, we have $\theta_h(\varepsilon) = o(1/\varepsilon)$ if and only if $\mathcal{D}_{\mathcal{X}}(\partial_{\mathcal{H}} h) = 0$. In particular, this implies that $\theta_{\mathcal{H}}(\varepsilon) = o(1/\varepsilon)$ whenever $\mathcal{D}_{\mathcal{X}}$ is supported on countably many data points.*

We now discuss conditions under which we can upper bound the disagreement coefficient by $O(1)$, which ensures results in Theorem 6 matching the minimax lower bound for active learning, up to logarithmic factors. We introduce the following *decomposable* condition.

**Definition 4.** *A marginal distribution $\mathcal{D}_{\mathcal{X}}$ is $\varepsilon$-decomposable if its (known) support $\mathrm{supp}(\mathcal{D}_{\mathcal{X}})$ can be decomposed into connected subsets, i.e., $\mathrm{supp}(\mathcal{D}_{\mathcal{X}}) = \cup_{i \in \mathcal{I}} \mathcal{X}_i$, such that*

$$\mathcal{D}_{\mathcal{X}}(\cup_{i \in \mathcal{I}'} \mathcal{X}_i) = O(\varepsilon),$$

*where $\mathcal{I}' := \{i \in \mathcal{I} : \mathcal{D}_{\mathcal{X}}(\mathcal{X}_i) \leq \varepsilon\}$.*

**Remark 1.** *Note that Definition 4 permits a decomposition such that $|\overline{\mathcal{I}}| = \Omega(\frac{1}{\varepsilon})$ where $\overline{\mathcal{I}} = \mathcal{I} \setminus \mathcal{I}'$. Definition 4 requires no knowledge of the index set $\mathcal{I}$ or any $\mathcal{X}_i$; it also places no restrictions on the conditional probability on each $\mathcal{X}_i$.*

We first give results for a general hypothesis class $\mathcal{H}$ as follows, and then discuss how to bound the disagreement coefficient for a set of neural networks.

**Proposition 3.** *Suppose $\mathcal{D}_{\mathcal{X}}$ is decomposable (into $\cup_{i \in \mathcal{I}} \mathcal{X}_i$) and the hypothesis class $\mathcal{H}$ consists of classifiers whose predication on each $\mathcal{X}_i$ is the same, i.e., $|\{h(x) : x \in \mathcal{X}_i\}| = 1$ for any $h \in \mathcal{H}$ and $i \in \mathcal{I}$. We then have $\theta_{\mathcal{H}}(\varepsilon) = O(1)$ for $\varepsilon$ sufficiently small.*

*Proof.* Fix any $h \in \mathcal{H}$. we know that for any $h' \in \mathcal{B}_{\mathcal{H}}(h, \varepsilon)$, we must have $\mathsf{DIS}(\{h, h'\}) \subseteq \cup_{i \in \mathcal{I}'} \mathcal{X}_i$ since $\mathcal{D}_{\mathcal{X}}(x \in \mathcal{X} : h(x) \neq h'(x)) \leq \varepsilon$, and $|\{h(x) : x \in \mathcal{X}_i\}| = 1$ for any $h \in \mathcal{H}$ and any $\mathcal{X}_i$. This further implies that $\mathbb{P}(\mathsf{DIS}(\mathcal{B}_{\mathcal{H}}(h, \varepsilon)) = O(\varepsilon)$, and thus $\theta_{\mathcal{H}}(\varepsilon) = O(1)$. $\qquad\square$

We next discuss conditions under which we can satisfy the prerequisites of Proposition 3. Suppose $\mathcal{D}_{\mathcal{X}\mathcal{Y}} \in \mathcal{P}(\alpha, \beta)$. We assume that $\mathcal{D}_{\mathcal{X}}$ is $(\varepsilon^{\frac{\beta}{1+\beta}})$-decomposable, and, for the desired accuracy level $\varepsilon$, we have

$$|\eta(x) - 1/2| \geq 2\varepsilon^{\frac{1}{1+\beta}}, \quad \forall x \in \mathrm{supp}(\mathcal{D}_{\mathcal{X}}). \tag{9}$$

With the above conditions satisfied, we can filter out neural networks that are clearly not "close" to $\eta$. Specifically, with $\kappa = \varepsilon^{\frac{1}{1+\beta}}$ and $\mathcal{F}_{\mathsf{dnn}}$ be the set of neural networks constructed from Proposition 1, we consider

$$\widetilde{\mathcal{F}}_{\mathsf{dnn}} := \{ f \in \mathcal{F}_{\mathsf{dnn}} : |f(x) - 1/2| \geq \varepsilon^{\frac{1}{1+\beta}}, \forall x \in \operatorname{supp}(\mathcal{D}_{\mathcal{X}}) \}, \tag{10}$$

which is guaranteed to contain $\bar{f} \in \mathcal{F}_{\mathsf{dnn}}$ such that $\|\bar{f} - \eta\|_\infty \leq \varepsilon^{\frac{1}{1+\beta}}$. Now focus on the subset

$$\widetilde{\mathcal{H}}_{\mathsf{dnn}} := \{ h_f : f \in \widetilde{\mathcal{F}}_{\mathsf{dnn}} \}. \tag{11}$$

We clearly have $h_{\bar{f}} \in \widetilde{\mathcal{H}}_{\mathsf{dnn}}$ (which ensures an $O(\varepsilon)$-optimal classifier) and $\operatorname{VCdim}(\widetilde{\mathcal{H}}_{\mathsf{dnn}}) \leq \operatorname{VCdim}(\mathcal{H}_{\mathsf{dnn}})$ (since $\widetilde{\mathcal{H}}_{\mathsf{dnn}} \subseteq \mathcal{H}_{\mathsf{dnn}}$). We upper bound the disagreement coefficient $\theta_{\widetilde{\mathcal{H}}_{\mathsf{dnn}}}(\varepsilon^{\frac{\beta}{1+\beta}})$ next.

**Proposition 4.** *Suppose $\mathcal{D}_{\mathcal{X}\mathcal{Y}} \in \mathcal{P}(\alpha, \beta)$ such that $\mathcal{D}_{\mathcal{X}}$ is $(\varepsilon^{\frac{\beta}{1+\beta}})$-decomposable and Eq. (9) is satisfied (with the desired accuracy level $\varepsilon$). We then have $\theta_{\widetilde{\mathcal{H}}_{\mathsf{dnn}}}(\varepsilon^{\frac{\beta}{1+\beta}}) = O(1)$.*

*Proof.* The proof is similar to the proof of Proposition 3. Fix any $h = h_f \in \widetilde{\mathcal{H}}_{\mathsf{dnn}}$. We first argue that, for any $i \in \mathcal{I}$, under Eq. (9), $|\{h_f(x) : x \in \mathcal{X}_i\}| = 1$, i.e., for $x \in \mathcal{X}_i$, $h_f(x)$ equals either 1 or 0, but not both: This can be seen from the fact that any $f \in \widetilde{\mathcal{F}}_{\mathsf{dnn}}$ is continuous and satisfies $|f(x) - 1/2| \geq \varepsilon^{\frac{1}{1+\beta}}$ for any $x \in \mathcal{X}_i$.

Fix any $h \in \widetilde{\mathcal{H}}_{\mathsf{dnn}}$. We know that for any $h' \in \mathcal{H}_{\widetilde{\mathcal{H}}_{\mathsf{dnn}}}(h, \varepsilon^{\frac{\beta}{1+\beta}})$, we must have $\operatorname{DIS}(\{h, h'\}) \subseteq \cup_{i \in \mathcal{I}'} \mathcal{X}_i$ due to similar reasons argued in the proof of Proposition 3. This further implies that $\mathbb{P}(\operatorname{DIS}(\mathcal{B}_{\widetilde{\mathcal{H}}_{\mathsf{dnn}}}(h, \varepsilon^{\frac{\beta}{1+\beta}}))) = O(\varepsilon^{\frac{\beta}{1+\beta}})$, and thus $\theta_{\widetilde{\mathcal{H}}_{\mathsf{dnn}}}(\varepsilon^{\frac{\beta}{1+\beta}}) = O(1)$. $\square$

We next argue that Eq. (9) is only needed in an approximate sense. We define the approximate decomposable condition in the following.

**Definition 5.** *A marginal distribution $\mathcal{D}_{\mathcal{X}}$ is $(\varepsilon, \delta)$-decomposable if there exists a known subset $\overline{\mathcal{X}} \subseteq \operatorname{supp}(\mathcal{D}_{\mathcal{X}})$ such that*

$$\mathcal{D}_{\mathcal{X}}(\overline{\mathcal{X}}) \geq 1 - \delta, \tag{12}$$

*and it can be decomposed into connected subsets, i.e., $\overline{\mathcal{X}} = \cup_{i \in \mathcal{I}} \mathcal{X}_i$, such that*

$$\mathcal{D}_{\mathcal{X}}(\cup_{i \in \mathcal{I}'} \mathcal{X}_i) = O(\varepsilon),$$

*where $\mathcal{I}' := \{ i \in \mathcal{I} : \mathcal{D}_{\mathcal{X}}(\mathcal{X}_i) \leq \varepsilon \}$.*

Suppose $\mathcal{D}_{\mathcal{X}\mathcal{Y}} \in \mathcal{P}(\alpha, \beta)$. We assume that $\mathcal{D}_{\mathcal{X}}$ is $(\varepsilon^{\frac{\beta}{1+\beta}}, \varepsilon^{\frac{\beta}{1+\beta}})$-decomposable (wrt $\overline{\mathcal{X}} \subseteq \mathcal{D}_{\mathcal{X}}$), and, for the desired accuracy level $\varepsilon$, we have

$$|\eta(x) - 1/2| \geq 2\varepsilon^{\frac{1}{1+\beta}}, \quad \forall x \in \overline{\mathcal{X}}. \tag{13}$$

With the above conditions satisfied, we can filter out neural networks that are clearly not "close" to $\eta$. Specifically, with $\kappa = \varepsilon^{\frac{1}{1+\beta}}$ and $\mathcal{F}_{\mathsf{dnn}}$ be the set of neural networks constructed from Proposition 1, we consider

$$\overline{\mathcal{F}}_{\mathsf{dnn}} := \{ f \in \mathcal{F}_{\mathsf{dnn}} : |f(x) - 1/2| \geq \varepsilon^{\frac{1}{1+\beta}}, \forall x \in \overline{\mathcal{X}} \}, \tag{14}$$

which is guaranteed to contain $\bar{f} \in \mathcal{F}_{\mathsf{dnn}}$ such that $\|\bar{f} - \eta\|_\infty \leq \varepsilon^{\frac{1}{1+\beta}}$. Now focus on the subset

$$\overline{\mathcal{H}}_{\mathsf{dnn}} := \{ h_f : f \in \overline{\mathcal{F}}_{\mathsf{dnn}} \}. \tag{15}$$

We clearly have $h_{\bar{f}} \in \overline{\mathcal{H}}_{\mathsf{dnn}}$ (which ensures an $O(\varepsilon)$-optimal classifier) and $\operatorname{VCdim}(\overline{\mathcal{H}}_{\mathsf{dnn}}) \leq \operatorname{VCdim}(\mathcal{H}_{\mathsf{dnn}})$ (since $\overline{\mathcal{H}}_{\mathsf{dnn}} \subseteq \mathcal{H}_{\mathsf{dnn}}$). We upper bound the disagreement coefficient $\theta_{\overline{\mathcal{H}}_{\mathsf{dnn}}}(\varepsilon^{\frac{\beta}{1+\beta}})$ next.

**Proposition 5.** *Suppose $\mathcal{D}_{\mathcal{X}\mathcal{Y}} \in \mathcal{P}(\alpha, \beta)$ such that $\mathcal{D}_{\mathcal{X}}$ is $(\varepsilon^{\frac{1}{1+\beta}}, \varepsilon)$-decomposable (wrt known $\overline{\mathcal{X}} \subseteq \operatorname{supp}(\mathcal{D}_{\mathcal{X}})$) and Eq. (13) is satisfied (with the desired accuracy level $\varepsilon$). We then have $\theta_{\overline{\mathcal{H}}_{\mathsf{dnn}}}(\varepsilon^{\frac{\beta}{1+\beta}}) = O(1)$.*

*Proof.* The proof is the same as the proof of Proposition 5 except for any $h' \in \mathcal{H}_{\overline{\mathcal{H}}_{\text{dnn}}}(h, \varepsilon^{\frac{\beta}{1+\beta}})$, we must have $\mathsf{DIS}(\{h, h'\}) \subseteq (\cup_{i \in \mathcal{I}'} \mathcal{X}_i) \cup (\text{supp}(\mathcal{D}_{\mathcal{X}}) \setminus \overline{\mathcal{X}})$. Based on the assumption that $\mathcal{D}_{\mathcal{X}}$ is $(\varepsilon^{\frac{1}{1+\beta}}, \varepsilon)$-decomposable, this also leads to $\theta_{\overline{\mathcal{H}}_{\text{dnn}}}(\varepsilon^{\frac{\beta}{1+\beta}}) = O(1)$ ☐

# D    Generic version of Algorithm 2 and its guarantees

This section is organized as follows. We first introduce some complexity measures in Appendix D.1. We then provide the generic algorithm (Algorithm 4) and state its theoretical guarantees (Theorem 14) in Appendix D.2.

## D.1    Complexity measures

We first introduce *pseudo dimension* (Pollard, 1984; Haussler, 1989, 1995), a complexity measure used to analyze real-valued functions.

**Definition 6** (Pseudo dimension). *Consider a set of real-valued function $\mathcal{F} : \mathcal{X} \to \mathbb{R}$. The pseudo dimension $\text{Pdim}(\mathcal{F})$ of $\mathcal{F}$ is defined as the VC dimension of the set of threshold functions $\{(x, \zeta) \mapsto \mathbb{1}(f(x) > \zeta) : f \in \mathcal{F}\}$.*

As discussed in Bartlett et al. (2019), similar results as in Theorem 4 holds true for $\text{Pdim}(\mathcal{F})$ as well.

**Theorem 13** (Bartlett et al. (2019)). *Let $\mathcal{F}_{\text{dnn}}$ be a set of neural network regression functions of the same architecture and with $W$ parameters arranged in $L$ layers. We then have*

$$\Omega(WL \log(W/L)) \leq \text{Pdim}(\mathcal{F}_{\text{dnn}}) \leq O(WL \log(W)).$$

We now introduce *value function disagreement coefficient*, which is proposed by Foster et al. (2020) in contextual bandits and then adapted to active learning by Zhu and Nowak (2022) with additional supreme over the marginal distribution $\mathcal{D}_{\mathcal{X}}$ to deal with distributional shifts caused by selective sampling.

**Definition 7** (Value function disagreement coefficient). *For any $f^\star \in \mathcal{F}$ and $\gamma_0, \varepsilon_0 > 0$, the value function disagreement coefficient $\theta_{f^\star}^{\text{val}}(\mathcal{F}, \gamma_0, \varepsilon_0)$ is defined as*

$$\sup_{\mathcal{D}_{\mathcal{X}}} \sup_{\gamma > \gamma_0, \varepsilon > \varepsilon_0} \left\{ \frac{\gamma^2}{\varepsilon^2} \cdot \mathbb{P}_{\mathcal{D}_{\mathcal{X}}} \left( \exists f \in \mathcal{F} : |f(x) - f^\star(x)| > \gamma, \|f - f^\star\|_{\mathcal{D}_{\mathcal{X}}} \leq \varepsilon \right) \right\} \vee 1,$$

*where $\|f\|_{\mathcal{D}_{\mathcal{X}}}^2 := \mathbb{E}_{x \sim \mathcal{D}_{\mathcal{X}}}[f^2(x)]$. We also define $\theta_{\mathcal{F}}^{\text{val}}(\gamma_0) := \sup_{f^\star \in \mathcal{F}, \varepsilon_0 > 0} \theta_{f^\star}^{\text{val}}(\mathcal{F}, \gamma_0, \varepsilon_0)$.*

## D.2    The generic algorithm and its guarantees

We present Algorithm 4, a generic version of Algorithm 2 that doesn't require the approximating classifiers to be neural networks.

**Algorithm 4** NeuralCAL++ (Generic Version)

---

**Input:** Accuracy level $\varepsilon \in (0,1)$, confidence level $\delta \in (0,1)$, abstention parameter $\gamma \in (0, 1/2)$.

1: Let $\mathcal{F} : \mathcal{X} \to [0,1]$ be a set of regression functions such that there exists a regression function $\bar{f} \in \mathcal{F}$ with $\|\bar{f} - \eta\|_\infty \leq \kappa \leq \gamma/4$.

2: Define $T := \frac{\theta_{\mathcal{F}}^{\mathrm{val}}(\gamma/4) \cdot \mathrm{Pdim}(\mathcal{F})}{\varepsilon \gamma}$, $M := \lceil \log_2 T \rceil$, and $C_\delta := O(\mathrm{Pdim}(\mathcal{F}) \cdot \log(T/\delta))$.

3: Define $\tau_m := 2^m$ for $m \geq 1$, $\tau_0 := 0$, and $\beta_m := 3(M - m + 1)C_\delta$.

4: Define $\widehat{R}_m(f) := \sum_{t=1}^{\tau_{m-1}} Q_t(\widehat{f}(x_t) - y_t)^2$ with the convention that $\sum_{t=1}^0 \ldots = 0$.

5: **for** epoch $m = 1, 2, \ldots, M$ **do**

6:     Get $\widehat{f}_m := \arg\min_{f \in \mathcal{F}} \sum_{t=1}^{\tau_{m-1}} Q_t(f(x_t) - y_t)^2$.

7:     (Implicitly) Construct active set $\mathcal{F}_m := \left\{ f \in \mathcal{F} : \widehat{R}_m(f) \leq \widehat{R}_m(\widehat{f}_m) + \beta_m \right\}$.

8:     Construct classifier $\widehat{h}_m : \mathcal{X} \to \{+1, -1, \bot\}$ as

$$\widehat{h}_m(x) := \begin{cases} \bot, & \text{if } [\mathsf{lcb}(x; \mathcal{F}_m) - \tfrac{\gamma}{4}, \mathsf{ucb}(x; \mathcal{F}_m) + \tfrac{\gamma}{4}] \subseteq [\tfrac{1}{2} - \gamma, \tfrac{1}{2} + \gamma]; \\ \mathrm{sign}(2\widehat{f}_m(x) - 1), & \text{o.w.} \end{cases}$$

    and query function $g_m(x) := \mathbb{1}\big( \tfrac{1}{2} \in \big( \mathsf{lcb}(x; \mathcal{F}_m) - \tfrac{\gamma}{4}, \mathsf{ucb}(x; \mathcal{F}_m) + \tfrac{\gamma}{4} \big) \big) \cdot \mathbb{1}(\widehat{h}_m(x) \neq \bot)$.

9:     **if** epoch $m = M$ **then**

10:         **Return** classifier $\widehat{h}_M$.

11:     **for** time $t = \tau_{m-1} + 1, \ldots, \tau_m$ **do**

12:         Observe $x_t \sim \mathcal{D}_{\mathcal{X}}$. Set $Q_t := g_m(x_t)$.

13:         **if** $Q_t = 1$ **then**

14:         Query the label $y_t$ of $x_t$.

---

We next state the theoretical guarantees for Algorithm 4.

**Theorem 14.** *Suppose $\theta_{\mathcal{F}}^{\mathrm{val}}(\gamma/4) \leq \bar{\theta}$ and the approximation level $\kappa \in (0, \gamma/4]$ satisfies*

$$\left( \frac{432\bar{\theta} \cdot M^2}{\gamma^2} \right) \cdot \kappa^2 \leq \frac{1}{10}. \tag{16}$$

*With probability at least $1 - \delta$, Algorithm 4 returns a classifier $\widehat{h} : \mathcal{X} \to \{+1, -1, \bot\}$ with Chow's excess error*

$$\mathsf{excess}_\gamma(\widehat{h}) = O\left( \varepsilon \cdot \log\left( \frac{\bar{\theta} \cdot \mathrm{Pdim}(\mathcal{F})}{\varepsilon \gamma \delta} \right) \right),$$

*after querying at most*

$$O\left( \frac{M^2 \cdot \mathrm{Pdim}(\mathcal{F}) \cdot \log(T/\delta) \cdot \bar{\theta}}{\gamma^2} \right)$$

*labels.*

Theorem 14 is proved in Appendix D.3, based on supporting lemmas and theorems established in Appendix D.2.1 and Appendix D.2.2. The general result (Theorem 14) will be used to prove results in specific settings (e.g., Theorem 7 and Theorem 18).

### D.2.1 Concentration results

The Freedman's inequality is commonly used in the field of active learning and contextual bandits, e.g., (Freedman, 1975; Agarwal et al., 2014; Krishnamurthy et al., 2019; Foster et al., 2020). We thus state the result without proof.

**Lemma 4** (Freedman's inequality). *Let $(X_t)_{t \leq T}$ be a real-valued martingale difference sequence adapted to a filtration $\mathfrak{F}_t$, and let $\mathbb{E}_t[\cdot] := \mathbb{E}[\cdot \mid \mathfrak{F}_{t-1}]$. If $|X_t| \leq B$ almost surely, then for any $\eta \in (0, 1/B)$ it holds with probability at least $1 - \delta$,*

$$\sum_{t=1}^T X_t \leq \eta \sum_{t=1}^T \mathbb{E}_t[X_t^2] + \frac{\log \delta^{-1}}{\eta}.$$

**Lemma 5.** *Let $(X_t)_{t \leq T}$ to be real-valued sequence of random variables adapted to a filtration $\mathfrak{F}_t$. If $|X_t| \leq B$ almost surely, then with probability at least $1 - \delta$,*

$$\sum_{t=1}^{T} X_t \leq \frac{3}{2} \sum_{t=1}^{T} \mathbb{E}_t[X_t] + 4B \log(2\delta^{-1}),$$

*and*

$$\sum_{t=1}^{T} \mathbb{E}_t[X_t] \leq 2 \sum_{t=1}^{T} X_t + 8B \log(2\delta^{-1}).$$

*Proof.* This is a direct consequence of Lemma 4. $\qquad\qquad\square$

We now define/recall some notations. Denote $n_m := \tau_m - \tau_{m-1}$. Fix any epoch $m \in [M]$ and any time step $t$ within epoch $m$. We have $f^\star = \eta$. For any $f \in \mathcal{F}$, we denote $M_t(f) := Q_t((f(x_t) - y_t)^2 - (f^\star(x_t) - y_t)^2)$, and $\widehat{R}_m(f) := \sum_{t=1}^{\tau_{m-1}} Q_t(f(x_t) - y_t)^2$. Recall that we have $Q_t = g_m(x_t)$. We define filtration $\mathfrak{F}_t := \sigma((x_1, y_1), \ldots, (x_t, y_t))$,[9] and denote $\mathbb{E}_t[\cdot] := \mathbb{E}[\cdot \mid \mathfrak{F}_{t-1}]$. We next present concentration results with respect to a general set of regression function $\mathcal{F}$ with finite pseudo dimension.

**Lemma 6** (Krishnamurthy et al. (2019)). *Consider an infinite set of regression function $\mathcal{F}$. Fix any $\delta \in (0, 1)$. For any $\tau, \tau' \in [T]$ such that $\tau < \tau'$, with probability at least $1 - \frac{\delta}{2}$, we have*

$$\sum_{t=\tau}^{\tau'} M_t(f) \leq \sum_{t=\tau}^{\tau'} \frac{3}{2} \mathbb{E}_t[M_t(f)] + C_\delta,$$

*and*

$$\sum_{t=\tau}^{\tau'} \mathbb{E}_t[M_t(f)] \leq 2 \sum_{t=\tau}^{\tau'} M_t(f) + C_\delta,$$

*where $C_\delta = C \cdot \left( \mathrm{Pdim}(\mathcal{F}) \cdot \log T + \log\left( \frac{\mathrm{Pdim}(\mathcal{F}) \cdot T}{\delta} \right) \right)$ with a universal constant $C > 0$.*

### D.2.2 Supporting lemmas for Theorem 14

Fix any classifier $\widehat{h} : \mathcal{X} \to \{+1, -1, \bot\}$. For any $x \in \mathcal{X}$, we use the notion

$\mathsf{excess}_\gamma(\widehat{h}; x) :=$
$\mathbb{P}_{y|x}\big(y \neq \mathrm{sign}(\widehat{h}(x))\big) \cdot \mathbb{1}\big(\widehat{h}(x) \neq \bot\big) + (1/2 - \gamma) \cdot \mathbb{1}\big(\widehat{h}(x) = \bot\big) - \mathbb{P}_{y|x}\big(y \neq \mathrm{sign}(h^\star(x))\big)$
$= \mathbb{1}\big(\widehat{h}(x) \neq \bot\big) \cdot \big(\mathbb{P}_{y|x}\big(y \neq \mathrm{sign}(\widehat{h}(x))\big) - \mathbb{P}_{y|x}\big(y \neq \mathrm{sign}(h^\star(x))\big)\big)$
$\qquad + \mathbb{1}\big(\widehat{h}(x) = \bot\big) \cdot \big((1/2 - \gamma) - \mathbb{P}_{y|x}\big(y \neq \mathrm{sign}(h^\star(x))\big)\big)$  (17)

to represent the excess error of $\widehat{h}$ at point $x \in \mathcal{X}$. Excess error of classifier $\widehat{h}$ can be then written as $\mathsf{excess}_\gamma(\widehat{h}) := \mathrm{err}_\gamma(\widehat{h}) - \mathrm{err}(h^\star) = \mathbb{E}_{x \sim \mathcal{D}_\mathcal{X}}[\mathsf{excess}_\gamma(\widehat{h}; x)]$.

We let $\mathcal{E}$ denote the good event considered in Lemma 6, we analyze under this event through out the rest of this section. Most lemmas presented in this section are inspired by results provided Zhu and Nowak (2022). Our main innovation is an inductive analysis of lemmas that eventually relaxes the requirements for approximation error for Theorem 14.

**General lemmas.** We introduce some general lemmas for Theorem 14.

**Lemma 7.** *For any $m \in [M]$, we have $g_m(x) = 1 \implies w(x; \mathcal{F}_m) > \frac{\gamma}{2}$.*

---

[9]$y_t$ is not observed (and thus not included in the filtration) when $Q_t = 0$. Note that $Q_t$ is measurable with respect to $\sigma((\mathfrak{F}_{t-1}, x_t))$.

*Proof.* We only need to show that $\mathsf{ucb}(x; \mathcal{F}_m) - \mathsf{lcb}(x; \mathcal{F}_m) \leq \frac{\gamma}{2} \implies g_m(x) = 0$. Suppose otherwise $g_m(x) = 1$, which implies that both

$$\frac{1}{2} \in \left( \mathsf{lcb}(x; \mathcal{F}_m) - \frac{\gamma}{4}, \mathsf{ucb}(x; \mathcal{F}_m) + \frac{\gamma}{4} \right) \quad \text{and}$$

$$\left[ \mathsf{lcb}(x; \mathcal{F}_m) - \frac{\gamma}{4}, \mathsf{ucb}(x; \mathcal{F}_m) + \frac{\gamma}{4} \right] \not\subseteq \left[ \frac{1}{2} - \gamma, \frac{1}{2} + \gamma \right]. \tag{18}$$

If $\frac{1}{2} \in \left( \mathsf{lcb}(x; \mathcal{F}_m) - \frac{\gamma}{4}, \mathsf{ucb}(x; \mathcal{F}_m) + \frac{\gamma}{4} \right)$ and $\mathsf{ucb}(x; \mathcal{F}_m) - \mathsf{lcb}(x; \mathcal{F}_m) \leq \frac{\gamma}{2}$, we must have $\mathsf{lcb}(x; \mathcal{F}_m) \geq \frac{1}{2} - \frac{3}{4}\gamma$ and $\mathsf{ucb}(x; \mathcal{F}_m) \leq \frac{1}{2} + \frac{3}{4}\gamma$, which contradicts with Eq. (18). $\qquad\square$

**Lemma 8.** *Fix any $m \in [M]$. Suppose $\bar{f} \in \mathcal{F}_m$, we have $\mathsf{excess}_\gamma(\widehat{h}_m; x) \leq 0$ if $g_m(x) = 0$.*

*Proof.* Recall that

$$\mathsf{excess}_\gamma(\widehat{h}; x) = \mathbb{1}\big(\widehat{h}(x) \neq \perp\big) \cdot \big( \mathbb{P}_{y|x}\big(y \neq \mathsf{sign}(\widehat{h}(x))\big) - \mathbb{P}_{y|x}\big(y \neq \mathsf{sign}(h^\star(x))\big) \big)$$
$$+ \mathbb{1}\big(\widehat{h}(x) = \perp\big) \cdot \big( (1/2 - \gamma) - \mathbb{P}_{y|x}\big(y \neq \mathsf{sign}(h^\star(x))\big) \big).$$

We now analyze the event $\{g_m(x) = 0\}$ in two cases.

**Case 1:** $\widehat{h}_m(x) = \perp$.

Since $\bar{f}(x) \in [\mathsf{lcb}(x; \mathcal{F}_m), \mathsf{ucb}(x; \mathcal{F}_m)]$ and $\kappa \leq \frac{\gamma}{4}$ by assumption, we know that $\eta(x) = f^\star(x) \in [\frac{1}{2} - \gamma, \frac{1}{2} + \gamma]$ and thus $\mathbb{P}_y\big(y \neq \mathsf{sign}(h^\star(x))\big) \geq \frac{1}{2} - \gamma$. As a result, we have $\mathsf{excess}_\gamma(\widehat{h}_m; x) \leq 0$.

**Case 2:** $\widehat{h}_m(x) \neq \perp$ **but** $\frac{1}{2} \notin (\mathsf{lcb}(x; \mathcal{F}_m) - \frac{\gamma}{4}, \mathsf{ucb}(x; \mathcal{F}_m) + \frac{\gamma}{4})$.

Since $\bar{f}(x) \in [\mathsf{lcb}(x; \mathcal{F}_m), \mathsf{ucb}(x; \mathcal{F}_m)]$ and $\kappa \leq \frac{\gamma}{4}$ by assumption, we clearly have $\mathsf{sign}(\widehat{h}_m(x)) = \mathsf{sign}(h^\star(x))$ when $\frac{1}{2} \notin (\mathsf{lcb}(x; \mathcal{F}_m) - \frac{\gamma}{4}, \mathsf{ucb}(x; \mathcal{F}_m) + \frac{\gamma}{4})$. We thus have $\mathsf{excess}_\gamma(\widehat{h}_m; x) \leq 0$. $\quad\square$

**Inductive lemmas.** We prove a set of statements for Theorem 14 in an inductive way. Fix any epoch $m \in [M]$, we consider

$$
\begin{cases}
\widehat{R}_m(\bar{f}) - \widehat{R}_m(f^\star) \leq \mathbb{E}_t\left[ Q_t\big(\bar{f}(x_t) - f^\star(x_t)\big)^2 \right] + C_\delta \leq \dfrac{3}{2} C_\delta \\[2mm]
\bar{f} \in \mathcal{F}_m \\[2mm]
\displaystyle\sum_{t=1}^{\tau_{m-1}} \mathbb{E}_t[M_t(f)] \leq 4\beta_m, \forall f \in \mathcal{F}_m \\[2mm]
\displaystyle\sum_{t=1}^{\tau_{m-1}} \mathbb{E}[Q_t(x_t)(f(x_t) - \bar{f}(x_t))^2] \leq 9\beta_m, \forall f \in \mathcal{F}_m \\[2mm]
\mathcal{F}_m \subseteq \mathcal{F}_{m-1}
\end{cases}
, \tag{19}
$$

$$\mathbb{E}_{x \sim \mathcal{D}_\mathcal{X}}[\mathbb{1}(g_m(x) = 1)] \leq \frac{144\beta_m}{\tau_{m-1}\gamma^2} \cdot \theta_{\bar{f}}^{\mathrm{val}}\left( \mathcal{F}, \gamma/4, \sqrt{\beta_m/\tau_{m-1}} \right) \leq \frac{144\beta_m}{\tau_{m-1}\gamma^2} \cdot \bar{\theta}, \tag{20}$$

and

$$\mathbb{E}_{x \sim \mathcal{D}_\mathcal{X}}[\mathbb{1}(g_m(x) = 1) \cdot w(x; \mathcal{F}_m)] \leq \frac{72\beta_m}{\tau_{m-1}\gamma} \cdot \theta_{\bar{f}}^{\mathrm{val}}\left( \mathcal{F}, \gamma/4, \sqrt{\beta_m/\tau_{m-1}} \right) \leq \frac{72\beta_m}{\tau_{m-1}\gamma} \cdot \bar{\theta}. \tag{21}$$

**Lemma 9.** *Fix any $\overline{m} = [M]$. When $\overline{m} = 1, 2$ or when Eq. (20) holds true for epochs $m = 2, 3, \ldots, \overline{m} - 1$, then Eq. (19) holds true for epoch $m = \overline{m}$.*

*Proof.* The statements in Eq. (19) clearly hold true for $m = \overline{m} = 1$ since, by definition, $\mathcal{F}_0 = \mathcal{F}$ and $\sum_{t=1}^{0} \ldots = 0$. We thus only need to consider the case when $\overline{m} \geq 2$. We next prove each of the five statements in Eq. (19) for epoch $m = \overline{m}$.

1. In the case when $\overline{m} = 2$, from Lemma 6, we know that

$$\widehat{R}_{\overline{m}}(\overline{f}) - \widehat{R}_{\overline{m}}(f^\star) \leq \sum_{t=1}^{\tau_{\overline{m}-1}} \frac{3}{2} \cdot \mathbb{E}_t\left[Q_t\left(\overline{f}(x_t) - f^\star(x_t)\right)^2\right] + C_\delta$$
$$\leq 3 + C_\delta \leq \frac{3}{2}C_\delta,$$

where the second line follows from the fact that $\tau_1 = 2$ (without loss of generality, we assume $C_\delta \geq 6$ here).

We now focus on the case when $\overline{m} \geq 3$. We have

$$\widehat{R}_{\overline{m}}(\overline{f}) - \widehat{R}_{\overline{m}}(f^\star) \leq \sum_{t=1}^{\tau_{\overline{m}-1}} \frac{3}{2} \cdot \mathbb{E}_t\left[Q_t\left(\overline{f}(x_t) - f^\star(x_t)\right)^2\right] + C_\delta$$
$$\leq \frac{3}{2}\sum_{\check{m}=1}^{\overline{m}-1} n_{\check{m}}\mathbb{E}_{x\sim\mathcal{D}_\mathcal{X}}\left[\mathbb{1}(g_{\check{m}}(x) = 1)\right] \cdot \kappa^2 + C_\delta$$
$$\leq \frac{3}{2}\left(2 + \sum_{\check{m}=2}^{\overline{m}-1} n_{\check{m}}\frac{144\beta_{\check{m}} \cdot \overline{\theta}}{\tau_{\check{m}-1}\gamma^2}\right) \cdot \kappa^2 + C_\delta$$
$$\leq \left(3 + \frac{144\overline{\theta}}{\gamma^2} \cdot \left(\sum_{\check{m}=2}^{\overline{m}-1} \beta_{\check{m}}\right)\right) \cdot \kappa^2 + C_\delta$$
$$\leq \left(3 + \frac{432\overline{\theta} \cdot M^2}{\gamma^2} \cdot C_\delta\right) \cdot \kappa^2 + C_\delta$$
$$\leq \frac{3}{2}C_\delta,$$

where the first line follows from Lemma 6; the second line follows from the fact that $\|\overline{f} - f^\star\|_\infty \leq \kappa$; the third line follows from Eq. (20); the forth line follows from $n_{\check{m}} = \tau_{\check{m}-1}$; the fifth line follows from the definition of $\beta_{\check{m}}$; and the last line follows from the choice of $\kappa$ in Eq. (16)

2. Since $\mathbb{E}_t[M_t(f)] = \mathbb{E}_t[Q_t(f(x_t) - f^\star(x_t))^2]$, by Lemma 6, we have $\widehat{R}_{\overline{m}}(f^\star) \leq \widehat{R}_{\overline{m}}(f) + C_\delta/2$ for any $f \in \mathcal{F}$. Combining this with statement 1 leads to

$$\widehat{R}_{\overline{m}}(\overline{f}) \leq \widehat{R}_{\overline{m}}(f) + 2C_\delta$$
$$\leq \widehat{R}_{\overline{m}}(f) + \beta_{\overline{m}}$$

for any $f \in \mathcal{F}$, where the second line follows from the definition of $\beta_{\overline{m}}$. We thus have $\overline{f} \in \mathcal{F}_{\overline{m}}$ based on the elimination rule.

3. Fix any $f \in \mathcal{F}_{\overline{m}}$. We have

$$\sum_{t=1}^{\tau_{\overline{m}-1}} \mathbb{E}_t[M_t(f)] \leq 2\sum_{t=1}^{\tau_{\overline{m}-1}} M_t(f) + C_\delta$$
$$= 2\widehat{R}_{\overline{m}}(f) - 2\widehat{R}_{\overline{m}}(f^\star) + C_\delta$$
$$\leq 2\widehat{R}_{\overline{m}}(f) - 2\widehat{R}_{\overline{m}}(\overline{f}) + 4C_\delta$$
$$\leq 2\widehat{R}_{\overline{m}}(f) - 2\widehat{R}_{\overline{m}}(\widehat{f}_{\overline{m}}) + 4C_\delta$$
$$\leq 2\beta_{\overline{m}} + 4C_\delta$$
$$\leq 4\beta_{\overline{m}},$$

where the first line follows from Lemma 6; the third line follows from statement 1; the fourth line follows from the fact that $\widehat{f}_{\overline{m}}$ is the minimizer of $\widehat{R}_{\overline{m}}(\cdot)$; and the fifth line follows from the fact that $f \in \mathcal{F}_{\overline{m}}$.

4. Fix any $f \in \mathcal{F}_{\overline{m}}$. We have

$$
\begin{aligned}
\sum_{t=1}^{\tau_{\overline{m}-1}} \mathbb{E}_t[Q_t(x_t)(f(x_t) - \bar{f}(x_t))^2] &= \sum_{t=1}^{\tau_{\overline{m}-1}} \mathbb{E}_t[Q_t(x_t)((f(x_t) - f^\star(x_t)) + (f^\star(x_t) - \bar{f}(x_t)))^2] \\
&\leq 2 \sum_{t=1}^{\tau_{\overline{m}-1}} \mathbb{E}_t[Q_t(x_t)(f(x_t) - f^\star(x_t))^2] + 2C_\delta \\
&= 2 \sum_{t=1}^{\tau_{\overline{m}-1}} \mathbb{E}_t[M_t(f)] + 2C_\delta \\
&\leq 8\beta_{\overline{m}} + 2C_\delta \\
&\leq 9\beta_{\overline{m}},
\end{aligned}
$$

where the second line follows from $(a+b)^2 \leq 2(a^2+b^2)$ and (the proof of) statement 1 on the second line; and the fourth line follows from statement 3.

5. Fix any $f \in \mathcal{F}_{\overline{m}}$. We have

$$
\begin{aligned}
\widehat{R}_{\overline{m}-1}(f) - \widehat{R}_{\overline{m}-1}(\widehat{f}_{\overline{m}-1}) &\leq \widehat{R}_{\overline{m}-1}(f) - \widehat{R}_{\overline{m}-1}(f^\star) + \frac{C_\delta}{2} \\
&= \widehat{R}_{\overline{m}}(f) - \widehat{R}_{\overline{m}}(f^\star) - \sum_{t=\tau_{\overline{m}-2}+1}^{\tau_{\overline{m}-1}} M_t(f) + \frac{C_\delta}{2} \\
&\leq \widehat{R}_{\overline{m}}(f) - \widehat{R}_{\overline{m}}(\bar{f}) + \frac{3}{2}C_\delta - \sum_{t=\tau_{\overline{m}-2}+1}^{\tau_{\overline{m}-1}} \mathbb{E}_t[M_t(f)]/2 + C_\delta \\
&\leq \widehat{R}_{\overline{m}}(f) - \widehat{R}_{\overline{m}}(\widehat{f}_{\overline{m}}) + \frac{5}{2}C_\delta \\
&\leq \beta_{\overline{m}} + 3C_\delta \\
&\leq \beta_{\overline{m}-1},
\end{aligned}
$$

where the first line follows from Lemma 6; the third line follows from statement 1 and Lemma 6; the fourth line follows from the fact that $\widehat{f}_{\overline{m}}$ is the minimizer with respect to $\widehat{R}_{\overline{m}}$ and Lemma 6; the last line follows from the construction of $\beta_{\overline{m}}$.

$\square$

We introduce more notations. Denote $(\mathcal{X}, \Sigma, \mathcal{D}_{\mathcal{X}})$ as the (marginal) probability space, and denote $\overline{\mathcal{X}}_m := \{x \in \mathcal{X} : g_m(x) = 1\} \in \Sigma$ be the region where query *is* requested within epoch $m$. Under the prerequisites of Lemma 10 and Lemma 11 (i.e., Eq. (19) holds true for epochs $m = 1, 2, \ldots, \overline{m}$), we have $\mathcal{F}_m \subseteq \mathcal{F}_{m-1}$ for $m = 1, 2, \ldots, \overline{m}$, which leads to $\overline{\mathcal{X}}_m \subseteq \overline{\mathcal{X}}_{m-1}$ for $m = 1, 2, \ldots, \overline{m}$. We now define a sub probability measure $\bar{\mu}_m := (\mathcal{D}_{\mathcal{X}})_{|\overline{\mathcal{X}}_m}$ such that $\bar{\mu}_m(\omega) = \mathcal{D}_{\mathcal{X}}(\omega \cap \overline{\mathcal{X}}_m)$ for any $\omega \in \Sigma$. Fix any epoch $m \leq \overline{m}$ and consider any measurable function $F$ (that is $\mathcal{D}_{\mathcal{X}}$ integrable), we have

$$
\begin{aligned}
\mathbb{E}_{x \sim \mathcal{D}_{\mathcal{X}}}[\mathbb{1}(g_{\overline{m}}(x) = 1) \cdot F(x)] &= \int_{x \in \overline{\mathcal{X}}_{\overline{m}}} F(x) \, d\mathcal{D}_{\mathcal{X}}(x) \\
&\leq \int_{x \in \overline{\mathcal{X}}_m} F(x) \, d\mathcal{D}_{\mathcal{X}}(x) \\
&= \int_{x \in \mathcal{X}} F(x) \, d\bar{\mu}_m(x) \\
&=: \mathbb{E}_{x \sim \bar{\mu}_m}[F(x)], \quad\quad\quad (22)
\end{aligned}
$$

where, by a slightly abuse of notations, we use $\mathbb{E}_{x \sim \mu}[\cdot]$ to denote the integration with any sub probability measure $\mu$. In particular, Eq. (22) holds with equality when $m = \overline{m}$.

**Lemma 10.** *Fix any epoch $\overline{m} \geq 2$. Suppose Eq. (19) holds true for epochs $m = 1, 2, \ldots, \overline{m}$, we then have Eq. (20) holds true for epoch $m = \overline{m}$.*

*Proof.* We prove Eq. (20) for epoch $m = \overline{m}$. We know that $\mathbb{1}(g_{\overline{m}}(x) = 1) = \mathbb{1}(g_{\overline{m}}(x) = 1) \cdot \mathbb{1}(w(x; \mathcal{F}_{\overline{m}}) > \gamma/2)$ from Lemma 7. Thus, for any $\check{m} \leq \overline{m}$, we have

$$
\begin{aligned}
\mathbb{E}_{x \sim \mathcal{D}_{\mathcal{X}}}[\mathbb{1}(g_{\overline{m}}(x) = 1)] &= \mathbb{E}_{x \sim \mathcal{D}_{\mathcal{X}}}[\mathbb{1}(g_{\overline{m}}(x) = 1) \cdot \mathbb{1}(w(x; \mathcal{F}_{\overline{m}}) > \gamma/2)] \\
&\leq \mathbb{E}_{x \sim \bar{\mu}_{\check{m}}}[\mathbb{1}(w(x; \mathcal{F}_{\overline{m}}) > \gamma/2)] \\
&\leq \mathbb{E}_{x \sim \bar{\mu}_{\check{m}}}\Big(\mathbb{1}\big(\sup_{f \in \mathcal{F}_{\overline{m}}} |f(x) - \bar{f}(x)| > \gamma/4\big)\Big), \quad (23)
\end{aligned}
$$

where the second line uses Eq. (22) and the last line follows from the facts that $\bar{f} \in \mathcal{F}_{\overline{m}}$ (by Eq. (19)) and $w(x; \mathcal{F}_{\overline{m}}) > \gamma/2 \implies \exists f \in \mathcal{F}_{\overline{m}}, |f(x) - \bar{f}(x)| > \gamma/4$.

For any time step $t$, let $m(t)$ denote the epoch where $t$ belongs to. From Eq. (19), we know that, $\forall f \in \mathcal{F}_{\overline{m}}$,

$$
\begin{aligned}
9\beta_{\overline{m}} &\geq \sum_{t=1}^{\tau_{\overline{m}-1}} \mathbb{E}_t\Big[Q_t\big(f(x_t) - \bar{f}(x_t)\big)^2\Big] \\
&= \sum_{t=1}^{\tau_{\overline{m}-1}} \mathbb{E}_{x \sim \mathcal{D}_{\mathcal{X}}}\Big[\mathbb{1}(g_{m(t)}(x) = 1) \cdot \big(f(x) - \bar{f}(x)\big)^2\Big] \\
&= \sum_{\check{m}=1}^{\overline{m}-1} n_{\check{m}} \cdot \mathbb{E}_{x \sim \bar{\mu}_{\check{m}}}\Big[\big(f(x) - \bar{f}(x)\big)^2\Big] \\
&= \tau_{\overline{m}-1} \mathbb{E}_{x \sim \bar{\nu}_{\overline{m}}}\Big[\big(f(x) - \bar{f}(x)\big)^2\Big], \quad (24)
\end{aligned}
$$

where we use $Q_t = g_{m(t)}(x_t) = \mathbb{1}(g_{m(t)}(x) = 1)$ and Eq. (22) on the second line, and define a new sub probability measure

$$
\bar{\nu}_{\overline{m}} := \frac{1}{\tau_{\overline{m}-1}} \sum_{\check{m}=1}^{\overline{m}-1} n_{\check{m}} \cdot \bar{\mu}_{\check{m}}
$$

on the third line.

Plugging Eq. (24) into Eq. (23) leads to the bound

$$
\begin{aligned}
&\mathbb{E}_{x \sim \mathcal{D}_{\mathcal{X}}}[\mathbb{1}(g_{\overline{m}}(x) = 1)] \\
&\leq \mathbb{E}_{x \sim \bar{\nu}_{\overline{m}}}\bigg[\mathbb{1}\Big(\exists f \in \mathcal{F}, |f(x) - \bar{f}(x)| > \gamma/4, \mathbb{E}_{x \sim \bar{\nu}_{\overline{m}}}\big[\big(f(x) - \bar{f}(x)\big)^2\big] \leq \frac{9\beta_{\overline{m}}}{\tau_{\overline{m}-1}}\Big)\bigg],
\end{aligned}
$$

where we use the definition of $\bar{\nu}_{\overline{m}}$ again (note that Eq. (23) works with any $\check{m} \leq \overline{m}$). Based on the Definition 7,[10] we then have

$$
\begin{aligned}
&\mathbb{E}_{x \sim \mathcal{D}_{\mathcal{X}}}[\mathbb{1}(g_{\overline{m}}(x) = 1)] \\
&\leq \frac{144\beta_{\overline{m}}}{\tau_{\overline{m}-1}\gamma^2} \cdot \theta_{\bar{f}}^{\mathrm{val}}\Big(\mathcal{F}, \gamma/4, \sqrt{9\beta_{\overline{m}}/2\tau_{\overline{m}-1}}\Big) \\
&\leq \frac{144\beta_{\overline{m}}}{\tau_{\overline{m}-1}\gamma^2} \cdot \theta_{\bar{f}}^{\mathrm{val}}\Big(\mathcal{F}, \gamma/4, \sqrt{\beta_{\overline{m}}/\tau_{\overline{m}-1}}\Big) \\
&\leq \frac{144\beta_{\overline{m}}}{\tau_{\overline{m}-1}\gamma^2} \cdot \bar{\theta}.
\end{aligned}
$$

$\square$

**Lemma 11.** *Fix any epoch $\overline{m} \geq 2$. Suppose Eq. (19) holds true for epochs $m = 1, 2, \ldots, \overline{m}$, we then have Eq. (21) holds true for epoch $m = \overline{m}$.*

*Proof.* We prove Eq. (21) for epoch $m = \overline{m}$. Similar to the proof of Lemma 10, we have

$$
\begin{aligned}
\mathbb{E}_{x \sim \mathcal{D}_{\mathcal{X}}}[\mathbb{1}(g_{\overline{m}}(x) = 1) \cdot w(x; \mathcal{F}_{\overline{m}})] &= \mathbb{E}_{x \sim \mathcal{D}_{\mathcal{X}}}[\mathbb{1}(g_{\overline{m}}(x) = 1) \cdot \mathbb{1}(w(x; \mathcal{F}_{\overline{m}}) > \gamma/2) \cdot w(x; \mathcal{F}_{\overline{m}})] \\
&\leq \mathbb{E}_{x \sim \bar{\mu}_{\check{m}}}[\mathbb{1}(w(x; \mathcal{F}_{\overline{m}}) > \gamma/2) \cdot w(x; \mathcal{F}_{\overline{m}})]
\end{aligned}
$$

---

[10]Note that analyzing with a sub probability measure $\bar{\nu}$ does not cause any problem. See Zhu and Nowak (2022) for detailed discussion.

for any $\check{m} \leq \overline{m}$. With $\overline{\nu}_{\overline{m}} := \frac{1}{\tau_{\overline{m}-1}} \sum_{\check{m}=1}^{\overline{m}-1} n_{\check{m}} \cdot \overline{\mu}_{\check{m}}$, we then have

$$
\begin{aligned}
&\mathbb{E}_{x \sim \mathcal{D}_{\mathcal{X}}}[\mathbb{1}(g_{\overline{m}}(x) = 1) \cdot w(x; \mathcal{F}_{\overline{m}})] \\
&\leq \mathbb{E}_{x \sim \overline{\nu}_{\overline{m}}}[\mathbb{1}(w(x; \mathcal{F}_{\overline{m}}) > \gamma/2) \cdot w(x; \mathcal{F}_{\overline{m}})] \\
&\leq \mathbb{E}_{x \sim \overline{\nu}_{\overline{m}}}\left[\mathbb{1}\left(\sup_{f \in \mathcal{F}_{\overline{m}}} |f(x) - \overline{f}(x)| > \gamma/4\right) \cdot \left(\sup_{f,f' \in \mathcal{F}_{\overline{m}}} |f(x) - f'(x)|\right)\right] \\
&\leq 2\mathbb{E}_{x \sim \overline{\nu}_{\overline{m}}}\left[\mathbb{1}\left(\sup_{f \in \mathcal{F}_{\overline{m}}} |f(x) - \overline{f}(x)| > \gamma/4\right) \cdot \left(\sup_{f \in \mathcal{F}_{\overline{m}}} |f(x) - \overline{f}(x)|\right)\right] \\
&\leq 2 \int_{\gamma/4}^{1} \mathbb{E}_{x \sim \overline{\nu}_{\overline{m}}}\left[\mathbb{1}\left(\sup_{f \in \mathcal{F}_{\overline{m}}} |f(x) - \overline{f}(x)| \geq \omega\right)\right] d\omega \\
&\leq 2 \int_{\gamma/4}^{1} \frac{1}{\omega^2} d\omega \cdot \left(\frac{9\beta_{\overline{m}}}{\tau_{\overline{m}-1}} \cdot \theta_{\overline{f}}^{\text{val}}\left(\mathcal{F}, \gamma/4, \sqrt{9\beta_{\overline{m}}/2\tau_{\overline{m}-1}}\right)\right) \\
&\leq \frac{72\beta_{\overline{m}}}{\tau_{\overline{m}-1}\,\gamma} \cdot \theta_{\overline{f}}^{\text{val}}\left(\mathcal{F}, \gamma/4, \sqrt{\beta_{\overline{m}}/\tau_{\overline{m}-1}}\right) \\
&\leq \frac{72\beta_{\overline{m}}}{\tau_{\overline{m}-1}\,\gamma} \cdot \overline{\theta},
\end{aligned}
$$

where we follow similar steps as in the proof of Lemma 10 and use some basic arithmetic facts. □

**Lemma 12.** *Eq.* (19), *Eq.* (20) *and Eq.* (21) *hold true for all* $m \in [M]$.

*Proof.* We first notice that, by Lemma 9, Eq. (19) holds true for epochs $\overline{m} = 1, 2$ unconditionally. We also know that, by Lemma 10 and Lemma 11, once Eq. (19) holds true for epochs $m = 1, 2, \ldots, \overline{m}$, Eq. (20) and Eq. (21) hold true for epochs $m = \overline{m}$ as well; at the same time, by Lemma 9, once Eq. (20) holds true for epochs $m = 2, 3, \ldots, \overline{m}$, Eq. (19) will hold true for epoch $m = \overline{m} + 1$.

We thus can start the induction procedure from $\overline{m} = 2$, and make sure that Eq. (19), Eq. (20) and Eq. (21) hold true for all $m \in [M]$. □

### D.3 Proof of Theorem 14

**Theorem 14.** *Suppose* $\theta_{\mathcal{F}}^{\text{val}}(\gamma/4) \leq \overline{\theta}$ *and the approximation level* $\kappa \in (0, \gamma/4]$ *satisfies*

$$\left(\frac{432\overline{\theta} \cdot M^2}{\gamma^2}\right) \cdot \kappa^2 \leq \frac{1}{10}. \tag{16}$$

*With probability at least* $1 - \delta$, *Algorithm 4 returns a classifier* $\widehat{h} : \mathcal{X} \to \{+1, -1, \bot\}$ *with Chow's excess error*

$$\text{excess}_{\gamma}(\widehat{h}) = O\left(\varepsilon \cdot \log\left(\frac{\overline{\theta} \cdot \text{Pdim}(\mathcal{F})}{\varepsilon\,\gamma\,\delta}\right)\right),$$

*after querying at most*

$$O\left(\frac{M^2 \cdot \text{Pdim}(\mathcal{F}) \cdot \log(T/\delta) \cdot \overline{\theta}}{\gamma^2}\right)$$

*labels.*

*Proof.* We analyze under the good event $\mathcal{E}$ defined in Lemma 6, which holds with probability at least $1 - \frac{\delta}{2}$. Note that all supporting lemmas stated in Appendix D.2.2 hold true under this event.

Fix any $m \in [M]$. We analyze the Chow's excess error of $\widehat{h}_m$, which is measurable with respect to $\mathfrak{F}_{\tau_{m-1}}$. For any $x \in \mathcal{X}$, if $g_m(x) = 0$, Lemma 8 implies that $\text{excess}_{\gamma}(\widehat{h}_m; x) \leq 0$. If $g_m(x) = 1$, we know that $\widehat{h}_m(x) \neq \bot$ and $\frac{1}{2} \in (\text{lcb}(x; \mathcal{F}_m) - \frac{\gamma}{4}, \text{ucb}(x; \mathcal{F}_m) + \frac{\gamma}{4})$. Since $\overline{f} \in \mathcal{F}_m$ by Lemma 12

(with Eq. (19)) and $\sup_{x \in \mathcal{X}} |\bar{f}(x) - f^\star(x)| \le \kappa \le \gamma/4$ by construction. The error incurred in this case is upper bounded by

$$
\begin{aligned}
\text{excess}(\widehat{h}_m; x) &\le 2|f^\star(x) - 1/2| \\
&\le 2\kappa + 2|\bar{f}(x) - 1/2| \\
&\le 2\kappa + 2w(x; \mathcal{F}_m) + \frac{\gamma}{2} \\
&\le 4w(x; \mathcal{F}_m),
\end{aligned}
$$

where we use Lemma 7 in the last line.

Combining these two cases together, we have

$$
\text{excess}(\widehat{h}_m) \le 4\,\mathbb{E}_{x \sim \mathcal{D}_\mathcal{X}}[\mathbb{1}(g_m(x) = 1) \cdot w(x; \mathcal{F}_m)].
$$

Take $m = M$ and apply Lemma 12 (and Eq. (21)) leads to the following guarantee.

$$
\begin{aligned}
\text{excess}(\widehat{h}_M) &\le \frac{576\beta_M}{\tau_{M-1}\gamma} \cdot \theta_{\bar{f}}^{\text{val}}\left(\mathcal{F}, \gamma/4, \sqrt{\beta_M/\tau_{M-1}}\right) \\
&\le O\left(\frac{\text{Pdim}(\mathcal{F})\log(T/\delta)}{T\,\gamma} \cdot \bar{\theta}\right) \\
&= O\left(\varepsilon \cdot \log\left(\frac{\bar{\theta} \cdot \text{Pdim}(\mathcal{F})}{\varepsilon\,\gamma\,\delta}\right)\right),
\end{aligned}
$$

where we use the fact that $T = \frac{\bar{\theta} \cdot \text{Pdim}(\mathcal{F})}{\varepsilon\,\gamma}$.

We now analyze the label complexity (note that the sampling process of Algorithm 4 stops at time $t = \tau_{M-1}$). Note that $\mathbb{E}[\mathbb{1}(Q_t = 1) \mid \mathfrak{F}_{t-1}] = \mathbb{E}_{x \sim \mathcal{D}_\mathcal{X}}[\mathbb{1}(g_m(x) = 1)]$ for any epoch $m \ge 2$ and time step $t$ within epoch $m$. Combine Lemma 12 with Eq. (20) (and Lemma 12) leads to

$$
\begin{aligned}
\sum_{t=1}^{\tau_{M-1}} \mathbb{1}(Q_t = 1) &\le \frac{3}{2} \sum_{t=1}^{\tau_{M-1}} \mathbb{E}[\mathbb{1}(Q_t = 1) \mid \mathfrak{F}_{t-1}] + 4\log(2/\delta) \\
&\le 3 + \frac{3}{2} \sum_{m=2}^{M-1} \frac{(\tau_m - \tau_{m-1}) \cdot 144\beta_m}{\tau_{m-1}\,\gamma^2} \cdot \bar{\theta} + 4\log(2/\delta) \\
&\le 3 + 4\log(2/\delta) + O\left(\frac{M^2 \cdot \text{Pdim}(\mathcal{F}) \cdot \log(T/\delta) \cdot \bar{\theta}}{\gamma^2}\right) \\
&= O\left(\frac{M^2 \cdot \text{Pdim}(\mathcal{F}) \cdot \log(T/\delta) \cdot \bar{\theta}}{\gamma^2}\right)
\end{aligned}
$$

with probability at least $1 - \delta$ (due to another application of Lemma 5 with confidence level $\delta/2$); where we use the fact that $T = \frac{\bar{\theta} \cdot \text{Pdim}(\mathcal{F})}{\varepsilon\,\gamma}$.  $\square$

# E   Proof of Theorem 7

We provide prerequisites in Appendix E.1 and the preprocessing procedures in Appendix E.2. We give the proof of Theorem 7 in Appendix E.3.

## E.1   Prerequisites

### E.1.1   Upper bound on the pseudo dimension

We present a result regarding the approximation and an upper bound on the pseudo dimension (i.e., Definition 6).

**Proposition 6.** *Suppose $\mathcal{D}_{\mathcal{X}\mathcal{Y}} \in \mathcal{P}(\alpha, \beta)$. One can construct a set of neural network regression functions $\mathcal{F}_{\text{dnn}}$ such that the following two properties hold simultaneously:*

$$
\exists f \in \mathcal{F}_{\text{dnn}} \text{ s.t. } \|f - f^\star\|_\infty \le \kappa, \quad \text{and} \quad \text{Pdim}(\mathcal{F}_{\text{dnn}}) \le c \cdot \kappa^{-\frac{d}{\alpha}} \log^2(\kappa^{-1}),
$$

*where $c > 0$ is a universal constant.*

*Proof.* The result follows by combining Theorem 3 and Theorem 13.  $\square$

### E.1.2 Upper bounds on the value function disagreement coefficient

We derive upper bounds on the value function disagreement coefficient (i.e., Definition 7). We first introduce the (value function) eluder dimension, a complexity measure that is closely related to the value function disagreement coefficient Russo and Van Roy (2013); Foster et al. (2020).

**Definition 8** (Value function eluder dimension). *For any $f^\star \in \mathcal{F}$ and $\gamma_0 > 0$, let $\check{\mathfrak{e}}_{f^\star}(\mathcal{F}, \gamma)$ be the length of the longest sequence of data points $x^1, \ldots, x^m$ such that for all $i$, there exists $f^i \in \mathcal{F}$ such that*

$$|f^i(x^i) - f^\star(x^i)| > \gamma, \quad \text{and} \quad \sum_{j < i}(f^i(x^j) - f^\star(x^j))^2 \le \gamma^2.$$

*The value function eluder dimension is defined as $\mathfrak{e}_{f^\star}(\mathcal{F}, \gamma_0) := \sup_{\gamma > \gamma_0} \check{\mathfrak{e}}_{f^\star}(\mathcal{F}, \gamma)$.*

The next result shows that the value function disagreement coefficient can be upper bounded by eluder dimension.

**Proposition 7** (Foster et al. (2020)). *Suppose $\mathcal{F}$ is a uniform Glivenko-Cantelli class. For any $f^\star : \mathcal{X} \to [0,1]$ and $\gamma, \varepsilon > 0$, we have $\theta_{f^\star}^{\text{val}}(\mathcal{F}, \gamma, \varepsilon) \le 4\,\mathfrak{e}_{f^\star}(\mathcal{F}, \gamma)$.*

We remark here that the requirement that $\mathcal{F}$ is a uniform Glivenko-Cantelli class is rather weak: It is satisfied as long as $\mathcal{F}$ has finite pseudo dimension (Anthony, 2002).

In the following, we only need to derive upper bounds on the value function eluder dimension, which upper bounds on the value function disagreement coefficient.[11] We first define two definitions: (i) the standard definition of covering number (e.g., see Wainwright (2019)), and (ii) a newly-proposed definition of approximate Lipschitzness.

**Definition 9.** *An $\iota$-covering of a set $\mathcal{X}$ with respect to a metric $\rho$ is a set $\{x_1, \ldots, x_N\} \subseteq \mathcal{X}$ such that for each $x \in \mathcal{X}$, there exists some $i \in [N]$ such that $\rho(x, x_i) \le \iota$. The $\iota$-covering number $\mathcal{N}(\iota; \mathcal{X}, \rho)$ is the cardinality of the smallest $\iota$-cover.*

**Definition 10.** *We call a function $f : \mathcal{X} \to \mathbb{R}$ $(L, \kappa)$-approximate Lipschitz if*

$$|f(x) - f(x')| \le L \cdot \|x - x'\|_2 + \kappa$$

*for any $x, x' \in \mathcal{X}$.*

We next provide upper bounds on value function eluder dimension and value function disagreement coefficient.

**Theorem 15.** *Suppose $\mathcal{F}$ is a set of $(L, \kappa/4)$-approximate Lipschitz functions. For any $\kappa' \ge \kappa$, we have $\sup_{f \in \mathcal{F}} \mathfrak{e}_f(\mathcal{F}, \kappa') \le 17 \cdot \mathcal{N}(\frac{\kappa'}{8L}; \mathcal{X}, \|\cdot\|_2)$.*

*Proof.* Fix any $f \in \mathcal{F}$ and $\bar{\kappa} \ge \kappa'$. We first give upper bounds on $\check{\mathfrak{e}}_f(\mathcal{F}, \bar{\kappa})$.

We construct $\mathcal{G} := \mathcal{F} - f$, which is a set of $(2L, \kappa/2)$-Lipschitz functions. Fix any eluder sequence $x^1, \ldots, x^m$ at scale $\bar{\kappa}$ and any $\check{x} \in \mathcal{X}$. We claim that $|\{x_j\}_{j \le m} \cap \mathcal{S}| \le 17$ where $\mathcal{S} := \{x \in \mathcal{X} : \|x - \check{x}\|_2 \le \frac{\bar{\kappa}}{8L}\}$. Suppose $\{x_j\}_{j \le m} \cap \mathcal{S} = x_{j_1}, \ldots, x_{j_k}$ ($j_i$ is ordered based on the ordering of $\{x_j\}_{j \le m}$). Since $x^{j_k}$ is added into the eluder sequence, there must exists a $g^{j_k} \in \mathcal{G}$ such that

$$|g^{j_k}(x^{j_k})| > \bar{\kappa}, \quad \text{and} \quad \sum_{j < j_k}(g^{j_k}(x^j))^2 \le \bar{\kappa}^2. \tag{25}$$

Since $g^{j_k}$ is $(2L, \kappa/2)$-Lipschitz, $\bar{\kappa} \ge \kappa' \ge \kappa$ and $x^{j_k} \in \mathcal{S}$, we must have $g^{j_k}(x) \ge \frac{\bar{\kappa}}{4}$ for any $x \in \mathcal{S}$. As a result, we must have $|\{x_j\}_{j < j_k} \cap \mathcal{S}^i| \le 16$ as otherwise the second constraint in Eq. (25) will be violated. We cover the space $\mathcal{X}$ with $\mathcal{N}(\frac{\bar{\kappa}}{8L}; \mathcal{X}, \|\cdot\|_2)$ balls of radius $\frac{\bar{\kappa}}{8L}$. Since the eluder sequence contains at most 17 data points within each ball, we know that $\check{\mathfrak{e}}_f(\mathcal{F}, \bar{\kappa}) \le 17 \cdot \mathcal{N}(\frac{\bar{\kappa}}{8L}; \mathcal{X}, \|\cdot\|_2)$.

The desired result follows by noticing that $17 \cdot \mathcal{N}(\frac{\bar{\kappa}}{8L}; \mathcal{X}, \|\cdot\|_2)$ is non-increasing in $\bar{\kappa}$. $\qquad\square$

---

[11]We focus on Euclidean geometry on $\mathcal{X}$ (i.e., using $\|\cdot\|_2$ norm) in deriving the upper bound. Slightly tighter bounds might be possible with other norms.

**Corollary 1.** *Suppose $\mathcal{X} \subseteq \mathbb{B}_r^d := \{x \in \mathbb{R}^d : \|x\|_2 \leq r\}$ and $\mathcal{F}$ is a set of $(L, \kappa/4)$-approximate Lipschitz functions. For any $\kappa' \geq \kappa$, we have $\theta_{\mathcal{F}}^{\mathrm{val}}(\kappa') := \sup_{f \in \mathcal{F}, \iota > 0} \theta_f^{\mathrm{val}}(\mathcal{F}, \kappa', \iota) \leq c \cdot (\frac{Lr}{\kappa'})^d$ with a universal constant $c > 0$.*

*Proof.* It is well-known that $\mathcal{N}(\iota; \mathbb{B}_r^d, \|\cdot\|_2) \leq (1 + 2r/\iota)^d$ (Wainwright, 2019). The desired result thus follows from combining Theorem 15 with Proposition 7. $\qquad\square$

### E.2 The preprocessing step: Clipping and filtering

Let $\eta : \mathcal{X} \to [0, 1]$ denote the true conditional probability and $\mathcal{F}_{\mathsf{dnn}}$ denote a set of neural network regression functions (e.g., constructed based on Theorem 3). We assume that (i) $\eta$ is $L$-Lipschitz, and (ii) there exists a $f \in \mathcal{F}$ such that $\|f - \eta\|_\infty \leq \kappa$ for some approximation factor $\kappa > 0$. We present the preprocessing step below in Algorithm 5.

---

**Algorithm 5** The Preprocessing Step: Clipping and Filtering

**Input:** A set of regression functions $\mathcal{F}$, Lipschitz parameter $L > 0$, approximation factor $\kappa > 0$.
1: **Clipping.** Set $\check{\mathcal{F}} := \{\check{f} : f \in \mathcal{F}\}$, where, for any $f \in \mathcal{F}$, we denote

$$\check{f}(x) := \begin{cases} 1, & \text{if } f(x) \geq 1; \\ 0, & \text{if } f(x) \leq 0; \\ f(x), & \text{o.w.} \end{cases}$$

2: **Filtering.** Set $\widetilde{\mathcal{F}} := \{\check{f} \in \check{\mathcal{F}} : \check{f} \text{ is } (L, 2\kappa)\text{-approximate Lipschitz}\}$
3: **Return** $\widetilde{\mathcal{F}}$.

---

**Proposition 8.** *Suppose $\eta$ is $L$-Lipschitz and $\mathcal{F}_{\mathsf{dnn}}$ is a set of neural networks (of the same architecture) with $W$ parameters arranged in $L$ layers such that there exists a $f \in \mathcal{F}_{\mathsf{dnn}}$ with $\|f - \eta\|_\infty \leq \kappa$. Let $\widetilde{\mathcal{F}}_{\mathsf{dnn}}$ be the set of functions obtained by applying Algorithm 5 on $\mathcal{F}_{\mathsf{dnn}}$, we then have (i) $\mathrm{Pdim}(\widetilde{\mathcal{F}}_{\mathsf{dnn}}) = O(WL \log(W))$, and (ii) there exists a $\widetilde{f} \in \widetilde{\mathcal{F}}_{\mathsf{dnn}}$ such that $\|\widetilde{f} - \eta\|_\infty \leq \kappa$.*

*Proof.* Suppose $f$ is a neural network function, we first notice that the "clipping" step can be implemented by adding one additional layer with $O(1)$ additional parameters for each neural network function. More specifically, fix any $f : \mathcal{X} \to \mathbb{R}$, we can set $\check{f}(x) := \mathsf{ReLU}(f(x)) - \mathsf{ReLU}(f(x) - 1)$. Set $\check{\mathcal{F}}_{\mathsf{dnn}} := \{\check{f} : f \in \mathcal{F}_{\mathsf{dnn}}\}$, we then have $\mathrm{Pdim}(\check{\mathcal{F}}_{\mathsf{dnn}}) = O(WL \log(W))$ based on Theorem 13. Let $\widetilde{\mathcal{F}}_{\mathsf{dnn}}$ be the filtered version of $\check{\mathcal{F}}_{\mathsf{dnn}}$. Since $\widetilde{\mathcal{F}}_{\mathsf{dnn}} \subseteq \check{\mathcal{F}}_{\mathsf{dnn}}$, we have $\mathrm{Pdim}(\widetilde{\mathcal{F}}_{\mathsf{dnn}}) = O(WL \log(W))$.

Since $\eta : \mathcal{X} \to [0, 1]$, we have $\|\check{f} - \eta\|_\infty \leq \|f - \eta\|_\infty$, which implies that there must exists a $\check{f} \in \check{\mathcal{F}}_{\mathsf{dnn}}$ such $\|\check{f} - \eta\|_\infty \leq \kappa$. To prove the second statement, it suffices to show that the $\check{f} \in \check{\mathcal{F}}$ that achieves $\kappa$ approximation error is not removed in the "filtering" step, i.e., $\check{f}$ is $(L, 2\kappa)$-approximate Lipschitz. For any $x, x' \in \mathcal{X}$, we have

$$|\check{f}(x) - \check{f}(x')| = |\check{f}(x) - \eta(x) + \eta(x) - \eta(x') + \eta(x') - \check{f}(x')|$$
$$\leq L\|x - x'\|_2 + 2\kappa,$$

where we use the $L$-Lipschitzness of $\eta$ and the fact that $\|\check{f} - \eta\|_\infty \leq \kappa$. $\qquad\square$

**Proposition 9.** *Suppose $\eta$ is $L$-Lipschitz and $\mathcal{X} \subseteq \mathbb{B}_r^d$. Fix any $\kappa \in (0, \gamma/32]$. There exists a set of neural network regression functions $\mathcal{F}_{\mathsf{dnn}}$ such that the followings hold simultaneously.*

1. *$\mathrm{Pdim}(\mathcal{F}_{\mathsf{dnn}}) \leq c \cdot \kappa^{-\frac{d}{\alpha}} \log^2(\kappa^{-1})$ with a universal constant $c > 0$.*

2. *There exists a $\overline{f} \in \mathcal{F}_{\mathsf{dnn}}$ such that $\|\overline{f} - \eta\|_\infty \leq \kappa$.*

3. *$\theta_{\mathcal{F}_{\mathsf{dnn}}}^{\mathrm{val}}(\gamma/4) := \sup_{f \in \mathcal{F}_{\mathsf{dnn}}, \iota > 0} \theta_f^{\mathrm{val}}(\mathcal{F}_{\mathsf{dnn}}, \gamma/4, \iota) \leq c' \cdot (\frac{Lr}{\gamma})^d$ with a universal constant $c' > 0$.*

*Proof.* Let $\mathcal{F}_{\mathsf{dnn}}$ be obtained by (i) invoking Theorem 3 with approximation level $\kappa$, and (ii) invoking Algorithm 5 on the set of functions obtained in step (i). The first two statements follow from

Proposition 8, and the third statement follows from Corollary 1 (note that to achieve guarantees for disagreement coefficient at level $\gamma/4$, we need to have $\kappa \leq \gamma/32$ when invoking Theorem 3). $\qquad\square$

### E.3 Proof of Theorem 7

**Theorem 7.** *Fix any $\varepsilon, \delta, \gamma > 0$. With probability at least $1 - \delta$, Algorithm 2 (with an appropriate initialization at line 1) returns a classifier $\widehat{h}$ with Chow's excess error $\widetilde{O}(\varepsilon)$ after querying $\mathrm{poly}(\frac{1}{\gamma}) \cdot \mathrm{polylog}(\frac{1}{\varepsilon \delta})$ labels.*

*Proof.* Let line 1 of Algorithm 2 be the set of neural networks $\mathcal{F}_{\mathsf{dnn}}$ generated from Proposition 9 with approximation level $\kappa \in (0, \gamma/32]$ (and constants $c, c'$ specified therein). To apply results derived in Theorem 14, we need to satisfying Eq. (16), i.e., specifying an approximation level $\kappa \in (0, \gamma/32]$ such that the following holds true

$$\frac{1}{\kappa^2} \geq \frac{4320 \cdot c' \cdot (\frac{Lr}{\gamma})^d \cdot \left( \left\lceil \log_2 \left( \frac{c' \cdot (\frac{Lr}{\gamma})^d \cdot c \cdot (\kappa^{-\frac{d}{\alpha}} \log^2(\kappa^{-1}))}{\varepsilon \gamma} \right) \right\rceil \right)^2}{\gamma^2}$$

For the setting we considered, i.e., $\mathcal{X} = [0,1]^d$ and $\eta \in \mathcal{W}_1^{\alpha, \infty}(\mathcal{X})$, we have $r = \sqrt{d} = O(1)$ and $L \leq \sqrt{d} = O(1)$ (e.g., see Theorem 4.1 in Heinonen (2005)).[12] We thus only need to select a $\kappa \in (0, \gamma/32]$ such that

$$\frac{1}{\kappa} \geq \bar{c} \cdot \left( \frac{1}{\gamma} \right)^{\frac{d}{2}+1} \cdot \left( \log \frac{1}{\varepsilon \gamma} + \log \frac{1}{\kappa} \right),$$

with a universal constant $\bar{c} > 0$ (that is possibly $d$-dependent and $\alpha$-dependent). Since $x \geq 2a \log a \implies x \geq a \log x$ for any $a > 0$, we can select a $\kappa > 0$ such that

$$\frac{1}{\kappa} = \check{c} \cdot \left( \frac{1}{\gamma} \right)^{\frac{d}{2}+1} \cdot \log \frac{1}{\varepsilon \gamma}$$

with a universal constant $\check{c} > 0$. With such choice of $\kappa$, from Proposition 9, we have

$$\mathrm{Pdim}(\mathcal{F}_{\mathsf{dnn}}) = O\left( \left( \frac{1}{\gamma} \right)^{\frac{d^2+d}{2\alpha}} \cdot \mathrm{polylog}\left( \frac{1}{\varepsilon \gamma} \right) \right).$$

Plugging this bound on $\mathrm{Pdim}(\mathcal{F}_{\mathsf{dnn}})$ and the upper bound on $\theta_{\mathcal{F}_{\mathsf{dnn}}}^{\mathrm{val}}(\gamma/4)$ from Proposition 9 into the guarantee of Theorem 14 leads to $\mathrm{excess}_\gamma(\widehat{h}) = O(\varepsilon \cdot \log(\frac{1}{\varepsilon \gamma \delta}))$ after querying

$$O\left( \left( \frac{1}{\gamma} \right)^{d+2+\frac{d^2+d}{2\alpha}} \cdot \mathrm{polylog}\left( \frac{1}{\varepsilon \gamma \delta} \right) \right)$$

labels. $\qquad\square$

## F  Other omitted details for Section 3

We discuss the proper abstention property of classifier learned in Algorithm 2 the computational efficiency of Algorithm 2 in Appendix F.1. We provide the proof of Theorem 8 in Appendix F.2.

### F.1  Proper abstention and computational efficiency

#### F.1.1  Proper abstention

We first recall the definition of *proper abstention* proposed in Zhu and Nowak (2022).

---

[12]Recall that we ignore constants that can be potentially $\alpha$-dependent and $d$-dependent.

**Definition 11** (Proper abstention). *A classifier* $\widehat{h} : \mathcal{X} \to \mathcal{Y} \cup \{\perp\}$ *enjoys proper abstention if and only if it abstains in regions where abstention is indeed the optimal choice, i.e.,* $\{x \in \mathcal{X} : \widehat{h}(x) = \perp\} \subseteq \{x \in \mathcal{X} : \eta(x) \in [\frac{1}{2} - \gamma, \frac{1}{2} + \gamma]\} =: \mathcal{X}_\gamma$.

We next show that the classifier $\widehat{h}$ returned by Algorithm 4 enjoys the proper abstention property. We also convert the abstaining classifier $\widehat{h} : \mathcal{X} \to \mathcal{Y} \cup \{\perp\}$ into a standard classifier $\check{h} : \mathcal{X} \to \mathcal{Y}$ and quantify its standard excess error. The conversion is through randomizing the prediction of $\widehat{h}$ over its abstention region, i.e., if $\widehat{h}(x) = \perp$, then its randomized version $\check{h}(x)$ predicts $+1/-1$ with equal probability (Puchkin and Zhivotovskiy, 2021).

**Proposition 10.** *The classifier $\widehat{h}$ returned by Algorithm 4 enjoys proper abstention. With randomization over the abstention region, we have the following upper bound on its standard excess error*

$$\text{err}(\check{h}) - \text{err}(h^\star) = \text{err}_\gamma(\widehat{h}) - \text{err}(h^\star) + \gamma \cdot \mathbb{P}_{x \sim \mathcal{D}_\mathcal{X}}(x \in \mathcal{X}_\gamma). \tag{26}$$

*Proof.* The proper abstention property of $\widehat{h}$ returned by Algorithm 4 is achieved via conservation: $\widehat{h}$ will avoid abstention unless it is absolutely sure that abstention is the optimal choice (also see the proof of Lemma 8.

Let $\check{h} : \mathcal{X} \to \mathcal{Y}$ be the randomized version of $\bar{h} : \mathcal{X} \to \{+1, -1, \perp\}$ (over the abstention region $\{x \in \mathcal{X} : \widehat{h}(x) = \perp\} \subseteq \mathcal{X}_\gamma$). We can see that, compared to the Chow's abstention error $1/2 - \gamma$, the additional error incurred over the abstention region is exactly $\gamma \cdot \mathbb{P}_{x \sim \mathcal{D}_\mathcal{X}}(x \in \mathcal{X}_\gamma)$. We thus have

$$\text{err}(\widehat{h}) - \text{err}(h^\star) \leq \text{err}_\gamma(\widehat{h}) - \text{err}(h^\star) + \gamma \cdot \mathbb{P}_{x \sim \mathcal{D}_\mathcal{X}}(x \in \mathcal{X}_\gamma).$$

$\square$

To characterize the standard excess error of classifier with proper abstention, we only need to upper bound the term $\mathbb{P}_{x \sim \mathcal{D}_\mathcal{X}}(x \in \mathcal{X}_\gamma)$, which does *not* depends on the (random) classifier $\widehat{h}$. Instead, it only depends on the marginal distribution.

We next introduce the Massart (Massart and Nédélec, 2006), which can be viewed as the extreme version of the Tsybakov noise by sending $\beta \to \infty$.

**Definition 12** (Massart noise). *The marginal distribution $\mathcal{D}_\mathcal{X}$ satisfies the Massart noise condition with parameter $\tau_0 > 0$ if $\mathbb{P}_{x \sim \mathcal{D}_\mathcal{X}}(|\eta(x) - 1/2| \leq \tau_0) = 0$.*

**Proposition 11.** *Suppose Massart noise holds. By setting the abstention parameter $\gamma = \tau_0$ in Algorithm 4 (and randomization over the abstention region), with probability at least $1 - \delta$, we obtain a classifier with standard excess error $\widetilde{O}(\varepsilon)$ after querying $\text{poly}(\frac{1}{\tau_0}) \cdot \text{polylog}(\frac{1}{\varepsilon \delta})$ labels.*

*Proof.* This is a direct consequence of Theorem 7 and Proposition 10. $\square$

### F.1.2 Computational efficiency

We discuss the efficient implementation of Algorithm 4 and its computational complexity in the section. The computational efficiency of Algorithm 4 mainly follows from the analysis in Zhu and Nowak (2022). We provide the discussion here for completeness.

**Regression orcale.** We introduce the regression oracle over the set of initialized neural networks $\mathcal{F}_{\text{dnn}}$ (line 1 at Algorithm 2). Given any set $\mathcal{S}$ of weighted examples $(w, x, y) \in \mathbb{R}_+ \times \mathcal{X} \times \mathcal{Y}$ as input, the regression oracle outputs

$$\widehat{f}_{\text{dnn}} := \arg\min_{f \in \mathcal{F}_{\text{dnn}}} \sum_{(w,x,y) \in \mathcal{S}} w(f(x) - y)^2.$$

While the exact computational complexity of such oracle with a set of neural networks remains elusive, in practice, running stochastic gradient descent often leads to great approximations. We quantify the computational complexity in terms of the number of calls to the regression oracle. Any future analysis on such oracle can be incorporated into our guarantees.

We first state some known results in computing the confidence intervals with respect to a general set of regression functions $\mathcal{F}$.

**Proposition 12** (Krishnamurthy et al. (2017); Foster et al. (2018, 2020)). *Consider the setting studied in Algorithm 4. Fix any epoch $m \in [M]$ and denote $\mathcal{B}_m := \{(x_t, Q_t, y_t)\}_{t=1}^{\tau_m - 1}$. Fix any $\iota > 0$. For any data point $x \in \mathcal{X}$, there exists algorithms $\mathbf{Alg}_{\mathsf{lcb}}$ and $\mathbf{Alg}_{\mathsf{ucb}}$ that certify*

$$\mathsf{lcb}(x; \mathcal{F}_m) - \iota \leq \mathbf{Alg}_{\mathsf{lcb}}(x; \mathcal{B}_m, \beta_m, \iota) \leq \mathsf{lcb}(x; \mathcal{F}_m) \quad and$$
$$\mathsf{ucb}(x; \mathcal{F}_m) \leq \mathbf{Alg}_{\mathsf{ucb}}(x; \mathcal{B}_m, \beta_m, \iota) \leq \mathsf{ucb}(x; \mathcal{F}_m) + \iota.$$

*The algorithms take $O(\frac{1}{\iota^2} \log \frac{1}{\iota})$ calls of the regression oracle for general $\mathcal{F}$ and take $O(\log \frac{1}{\iota})$ calls of the regression oracle if $\mathcal{F}$ is convex and closed under pointwise convergence.*

*Proof.* See Algorithm 2 in Krishnamurthy et al. (2017) for the general case; and Algorithm 3 in Foster et al. (2018) for the case when $\mathcal{F}$ is convex and closed under pointwise convergence. $\quad\square$

We now state the computational guarantee of Algorithm 4, given the regression oracle introduced above.

**Theorem 16.** *Algorithm 4 can be efficiently implemented via the regression oracle and enjoys the same theoretical guarantees stated in Theorem 7. The number of oracle calls needed is $\mathrm{poly}(\frac{1}{\gamma}) \cdot \frac{1}{\varepsilon}$; the per-example inference time of the learned $\widehat{h}_M$ is $\widetilde{O}(\frac{1}{\gamma^2} \cdot \mathrm{polylog}(\frac{1}{\varepsilon\gamma}))$ for general $\mathcal{F}$, and $\widetilde{O}(\mathrm{polylog}(\frac{1}{\varepsilon\gamma}))$ when $\mathcal{F}$ is convex.*

*Proof.* Fix any epoch $m \in [M]$. Denote $\bar{\iota} := \frac{\gamma}{8M}$ and $\iota_m := \frac{(M-m)\gamma}{8M}$. With any observed $x \in \mathcal{X}$, we construct the approximated confidence intervals $\widehat{\mathsf{lcb}}(x; \mathcal{F}_m)$ and $\widehat{\mathsf{ucb}}(x; \mathcal{F}_m)$ as follows.

$$\widehat{\mathsf{lcb}}(x; \mathcal{F}_m) := \mathbf{Alg}_{\mathsf{lcb}}(x; \mathcal{B}_m, \beta_m, \bar{\iota}) - \iota_m \quad \text{and} \quad \widehat{\mathsf{ucb}}(x; \mathcal{F}_m) := \mathbf{Alg}_{\mathsf{ucb}}(x; \mathcal{B}_m, \beta_m, \bar{\iota}) + \iota_m.$$

For efficient implementation of Algorithm 4, we replace $\mathsf{lcb}(x; \mathcal{F}_m)$ and $\mathsf{ucb}(x; \mathcal{F}_m)$ with $\widehat{\mathsf{lcb}}(x; \mathcal{F}_m)$ and $\widehat{\mathsf{ucb}}(x; \mathcal{F}_m)$ in the construction of $\widehat{h}_m$ and $g_m$.

Based on Proposition 12, we know that

$$\mathsf{lcb}(x; \mathcal{F}_m) - \iota_m - \bar{\iota} \leq \widehat{\mathsf{lcb}}(x; \mathcal{F}_m) \leq \mathsf{lcb}(x; \mathcal{F}_m) - \iota_m \quad \text{and}$$
$$\mathsf{ucb}(x; \mathcal{F}_m) + \iota_m \leq \widehat{\mathsf{ucb}}(x; \mathcal{F}_m) \leq \mathsf{ucb}(x; \mathcal{F}_m) + \iota_m + \bar{\iota}.$$

Since $\iota_m + \bar{\iota} \leq \frac{\gamma}{8}$ for any $m \in [M]$, the guarantee stated in Lemma 7 can be modified as $g_m(x) = 1 \implies w(x; \mathcal{F}_m) \geq \frac{\gamma}{4}$. The guarantee stated in Lemma 8 also holds true since we have $\widehat{\mathsf{lcb}}(x; \mathcal{F}_m) \leq \mathsf{lcb}(x; \mathcal{F}_m)$ and $\widehat{\mathsf{ucb}}(x; \mathcal{F}_m) \geq \mathsf{ucb}(x; \mathcal{F}_m)$ by construction. Suppose $\mathcal{F}_m \subseteq \mathcal{F}_{m-1}$ (as in Lemma 9), we have

$$\widehat{\mathsf{lcb}}(x; \mathcal{F}_m) \geq \mathsf{lcb}(x; \mathcal{F}_m) - \iota_m - \bar{\iota} \geq \mathsf{lcb}(x; \mathcal{F}_{m-1}) - \iota_{m-1} \geq \widehat{\mathsf{lcb}}(x; \mathcal{F}_{m-1}) \quad \text{and}$$
$$\widehat{\mathsf{ucb}}(x; \mathcal{F}_m) \leq \mathsf{ucb}(x; \mathcal{F}_m) + \iota_m + \bar{\iota} \leq \mathsf{ucb}(x; \mathcal{F}_{m-1}) + \iota_{m-1} \leq \widehat{\mathsf{ucb}}(x; \mathcal{F}_{m-1}),$$

which ensures that $\mathbb{1}(g_m(x) = 1) \leq \mathbb{1}(g_{m-1}(x) = 1)$. Thus, the inductive lemmas appearing in Appendix D.2.2 can be proved similarly with changes only in constant terms (also change the constant terms in the definition of $\bar{\theta}$ and in Eq. (16), since $\frac{\gamma}{2}$ is replaced by $\frac{\gamma}{4}$ in Lemma 7). As a result, the guarantees stated in Theorem 14 (and Theorem 7) hold true with changes only in constant terms.

We now discuss the computational complexity of the efficient implementation. At the beginning of each epoch $m$. We use one oracle call to compute $\widehat{f}_m := \arg\min_{f \in \mathcal{F}} \sum_{t=1}^{\tau_m - 1} Q_t(f(x_t) - y_t)^2$. The main computational cost comes from computing $\widehat{\mathsf{lcb}}$ and $\widehat{\mathsf{ucb}}$ at each time step. We take $\iota = \bar{\iota} := \frac{\gamma}{8M}$ into Proposition 12, which leads to $O(\frac{(\log T)^2}{\gamma^2} \cdot \log(\frac{\log T}{\gamma}))$ calls of the regression oracle for general $\mathcal{F}$ and $O(\log(\frac{\log T}{\gamma}))$ calls of the regression oracle for any convex $\mathcal{F}$ that is closed under pointwise convergence. This also serves as the per-example inference time for $\widehat{h}_M$. The total computational cost of Algorithm 4 is then derived by multiplying the per-round cost by $T$ and plugging $T = \frac{\theta \operatorname{Pdim}(\mathcal{F})}{\varepsilon\gamma} = \widetilde{O}(\mathrm{poly}(\frac{1}{\gamma}) \cdot \frac{1}{\varepsilon})$ into the bound . $\quad\square$

## F.2 Proof of Theorem 8

For ease of construction, we suppose the instance space is $\mathcal{X} = \mathbb{B}_1^d := \{x \in \mathbb{R}^d : \|x\|_2 \leq 1\}$. Part of our construction is inspired by Li et al. (2021).

**Theorem 8.** *Fix any $\gamma \in (0, 1/8)$. For any accuracy level $\varepsilon$ sufficiently small, there exists a problem instance such that (1) $\eta \in \mathcal{W}_1^{1,\infty}(\mathcal{X})$ and is of the form $\eta(x) := \mathsf{ReLU}(\langle w, x \rangle + a) + b$; and (2) for any active learning algorithm, it takes at least $\gamma^{-\Omega(d)}$ labels to identify an $\varepsilon$-optimal classifier, for either standard excess error or Chow's excess error (with parameter $\gamma$).*

*Proof.* Fix any $\gamma \in (0, 1/8)$. We first claim that we can find a discrete subset $\overline{\mathcal{X}} \subseteq \mathcal{X}$ with cardinality $|\overline{\mathcal{X}}| \geq (1/8\gamma)^{d/2}$ such that $\|x_i\|_2 = 1$ and $\langle x_1, x_2 \rangle \leq 1 - 4\gamma$ for any $x_i \in \overline{\mathcal{X}}$. To prove this, we first notice that $\|x_1 - x_2\|_2 \geq \tau \iff \langle x_1, x_2 \rangle \leq 1 - \tau^2/2$. Since the $\tau$-packing number on the unit sphere is at least $(1/\tau)^d$, setting $\tau = \sqrt{8\gamma}$ leads to the desired claim.

We set $\mathcal{D}_{\mathcal{X}} := \mathrm{unif}(\overline{\mathcal{X}})$ and $\mathcal{F}_{\mathsf{dnn}} := \{\mathsf{ReLU}(\langle w, \cdot \rangle - (1 - 4\gamma)) + (1/2 - 2\gamma) : w \in \overline{\mathcal{X}}\}$. We have $\mathcal{F}_{\mathsf{dnn}} \subseteq \mathcal{W}_1^{1,\infty}(\mathcal{X})$ since $\|w\|_2 \leq$ for any $w \in \overline{\mathcal{X}}$. We randomly select a $w^\star \in \mathcal{X}$ and set $f^\star(\cdot) = \eta(\cdot) = \mathsf{ReLU}(\langle w^\star, \cdot \rangle - (1 - 4\gamma)) + (1/2 - 2\gamma)$. We assume that the labeling feedback is the conditional expectation, i.e., $\eta(x)$ is provided if $x$ is queried. We see that $f^\star(x) = 1/2 - 2\gamma$ for any $x \in \mathcal{X}$ but $x \neq w^\star$, and $f^\star(w^\star) = 1/2 + 2\gamma$. We can see that mistakenly select the wrong $\widehat{f} \neq f^\star$ leads to $\frac{\gamma}{4} \cdot \frac{2}{|\overline{\mathcal{X}}|} = \frac{\gamma}{2|\overline{\mathcal{X}}|}$ excess error. Note that the excess error holds true in both standard excess error and Chow's excess error (with parameter $\gamma$) since $\mathcal{D}_{\mathcal{X}}(x \in \mathcal{X} : \eta(x) \in [1/2 - \gamma, 1/2 + \gamma]) = 0$ by construction.

We suppose the desired access error $\varepsilon$ is sufficiently small (e.g., $\varepsilon \leq \frac{\gamma}{8|\overline{\mathcal{X}}|}$). We now show that, with label complexity at most $K := \lfloor |\overline{\mathcal{X}}|/2 \rfloor = \Omega(\gamma^{-d/2})$, any active learning algorithm will, in expectation, pick a classifier that has $\Omega(\varepsilon)$ excess error. Since the worst case error of any randomized algorithm is lower bounded by the expected error of the best deterministic algorithm against a input distribution (Yao, 1977), we only need to analyze a deterministic learner. We set the input distribution as the uniform distribution over instances with parameter $w^\star \in \overline{\mathcal{X}}$. For any deterministic algorithm, we use $s := (x_{i_1}, \ldots, x_{i_K})$ to denote the data points queried under the constraint that at most $K$ labels can be queried. We denote $\widehat{f} \in \mathcal{F}$ as the learned classifier conditioned on $s$. Since $w^\star \sim \mathrm{unif}(\overline{\mathcal{X}})$, we know that, with probability at least $\frac{1}{2}$, $w^\star \notin s$. Conditioned on that event, we know that, with probability at least $\frac{1}{2}$, the learner will output $\widehat{f} \neq f^\star$ since more than half of the data points remains unqueried. The deterministic algorithm thus outputs the wrong $\widehat{f} \neq f^\star$ with probability at least $\frac{1}{2} \cdot \frac{1}{2} = \frac{1}{4}$, which has $\frac{\gamma}{2|\overline{\mathcal{X}}|}$ excess error as previously discussed. When $\varepsilon \leq \frac{\gamma}{8|\overline{\mathcal{X}}|}$, this leads to $\Omega(\varepsilon)$ excess error in expectation. $\qquad\square$

# G Omitted details for Section 4

We provide mathematical backgrounds for the Radon $\mathsf{BV}^2$ space in Appendix G.1, derive approximation results and passive learning results in Appendix G.2, and derive active learning results in Appendix G.3.

## G.1 The Radon $\mathsf{BV}^2$ space

We provide explicit definition of the $\|f\|_{\mathscr{R}\mathsf{BV}^2(\mathcal{X})}$ and associated mathematical backgrounds in this section. Also see Ongie et al. (2020); Parhi and Nowak (2021, 2022a,b); Unser (2022) for more discussions.

We first introduce the *Radon transform* of a function $f : \mathbb{R}^d \to \mathbb{R}$ as

$$\mathscr{R}\{f\}(\gamma, t) := \int_{\{x : \gamma^\top x = t\}} f(x) \, \mathsf{d}s(x), \quad (\gamma, t) \in \mathbb{S}^{d-1} \times \mathbb{R},$$

where $s$ denotes the surface measure on the hyperplane $\{x : \gamma^\top x = t\}$. The Radon domain is parameterized by a *direction* $\gamma \in \mathbb{S}^{d-1}$ and an *offset* $t \in \mathbb{R}$. We also introduce the *ramp filter* as

$$\Lambda^{d-1} := (-\partial_t^2)^{\frac{d-1}{2}},$$

where $\partial_t$ denotes the partial derivative with respect to the offset variable, $t$, of the Radon domain, and the fractional powers are defined in terms of Riesz potentials.

With the above preparations, we can define the $\mathscr{R}\,\mathsf{TV}^2$-seminorm as

$$\mathscr{R}\,\mathsf{TV}^2(f) := c_d \|\partial_t^2 \Lambda^{d-1} \mathscr{R}f\|_{\mathcal{M}(\mathbb{S}^{d-1} \times \mathbb{R})},$$

where $c_d = 1/(2(2\pi)^{d-1})$ is a dimension-dependent constant, and $\|\cdot\|_{\mathcal{M}(\mathbb{S}^{d-1} \times \mathbb{R})}$ denotes the *total variation norm* (in terms of measures) over the bounded domain $\mathbb{S}^{d-1} \times \mathbb{R}$. The $\mathscr{R}\,\mathsf{BV}^2$ norm of $f$ over $\mathbb{R}^d$ is defined as

$$\|f\|_{\mathscr{R}\,\mathsf{BV}^2(\mathbb{R}^d)} := \mathscr{R}\,\mathsf{TV}^2(f) + |f(0)| + \sum_{k=1}^{d} |f(e_k) - f(0)|,$$

where $\{e_k\}_{k=1}^d$ denotes the canonical basis of $\mathbb{R}^d$. The $\mathscr{R}\,\mathsf{BV}^2(\mathbb{R}^d)$ space is then defined as

$$\mathscr{R}\,\mathsf{BV}^2(\mathbb{R}^d) := \{f \in L^{\infty,1}(\mathbb{R}^d) : \mathscr{R}\,\mathsf{BV}^2(f) < \infty\},$$

where $L^{\infty,1}(\mathbb{R}^d)$ is the Banach space of functions mapping $\mathbb{R}^d \to \mathbb{R}$ of at most linear growth. To define the $\mathscr{R}\,\mathsf{BV}^2$ norm of $f$ over a bounded domain $\mathcal{X} \subseteq \mathbb{R}^d$, we use the standard approach of considering restrictions of functions in $\mathscr{R}\,\mathsf{BV}^2(\mathbb{R}^d)$, i.e.,

$$\|f\|_{\mathscr{R}\,\mathsf{BV}^2(\mathcal{X})} := \inf_{g \in \mathscr{R}\,\mathsf{BV}^2(\mathbb{R}^d)} \|g\|_{\mathscr{R}\,\mathsf{BV}^2(\mathbb{R}^d)} \quad \text{s.t.} \quad g|_{\mathcal{X}} = f.$$

In the rest of Appendix G, we use $\mathcal{P}(\beta)$ to denote the set of distributions that satisfy (1) Tsybakov noise condition with parameter $\beta \geq 0$; and (2) $\eta \in \mathscr{R}\,\mathsf{BV}_1^2(\mathcal{X})$.

## G.2 Approximation and passive learning results

**Proposition 13.** *Suppose $\mathcal{D}_{\mathcal{XY}} \in \mathcal{P}(\beta)$. One can construct a set of neural network classifier $\mathcal{H}_{\mathsf{dnn}}$ such that the following two properties hold simultaneously:*

$$\min_{h \in \mathcal{H}_{\mathsf{dnn}}} \mathrm{err}(h) - \mathrm{err}(h^\star) = O(\varepsilon) \quad \text{and} \quad \mathrm{VCdim}(\mathcal{H}_{\mathsf{dnn}}) = \widetilde{O}(\varepsilon^{-\frac{2d}{(1+\beta)(d+3)}}).$$

*Proof.* We take $\kappa = \varepsilon^{\frac{1}{1+\beta}}$ in Theorem 9 to construct a set of neural network classifiers $\mathcal{H}_{\mathsf{dnn}}$ with $W = O(\varepsilon^{-\frac{2d}{(1+\beta)(d+3)}})$ total parameters arranged in $L = O(1)$ layers. According to Theorem 4, we know

$$\mathrm{VCdim}(\mathcal{H}_{\mathsf{dnn}}) = O(\varepsilon^{-\frac{2d}{(1+\beta)(d+3)}} \cdot \log(\varepsilon^{-1})) = \widetilde{O}(\varepsilon^{-\frac{2d}{(1+\beta)(d+3)}}).$$

We now show that there exists a classifier $\bar{h} \in \mathcal{H}_{\mathsf{dnn}}$ with small excess error. Let $\bar{h} = h_{\bar{f}}$ be the classifier such that $\|\bar{f} - \eta\|_\infty \leq \kappa$. We can see that

$$\begin{aligned}
\mathrm{excess}(\bar{h}) &= \mathbb{E}\big[\mathbb{1}(\bar{h}(x) \neq y) - \mathbb{1}(h^\star(x) \neq y)\big] \\
&= \mathbb{E}\big[|2\eta(x) - 1| \cdot \mathbb{1}(\bar{h}(x) \neq h^\star(x))\big] \\
&\leq 2\kappa \cdot \mathbb{P}_{x \sim \mathcal{D}_{\mathcal{X}}}(x \in \mathcal{X} : |\eta(x) - 1/2| \leq \kappa) \\
&= O(\kappa^{1+\beta}) \\
&= O(\varepsilon),
\end{aligned}$$

where the third line follows from the fact that $\bar{h}$ and $h^\star$ disagrees only within region $\{x \in \mathcal{X} : |\eta(x) - 1/2| \leq \kappa\}$ and the incurred error is at most $2\kappa$ on each disagreed data point. The fourth line follows from the Tsybakov noise condition and the last line follows from the selection of $\kappa$. $\square$

**Theorem 17.** *Suppose $\mathcal{D}_{\mathcal{XY}} \in \mathcal{P}(\beta)$. Fix any $\varepsilon, \delta > 0$. Let $\mathcal{H}_{\mathsf{dnn}}$ be the set of neural network classifiers constructed in Proposition 13. With $n = \widetilde{O}(\varepsilon^{-\frac{4d+6+\beta(d+3)}{(1+\beta)(d+3)}})$ i.i.d. sampled data points, with probability at least $1 - \delta$, the empirical risk minimizer $\widehat{h} \in \mathcal{H}_{\mathsf{dnn}}$ achieves excess error $O(\varepsilon)$.*

*Proof.* Proposition 13 certifies $\min_{h\in\mathcal{H}_{\mathsf{dnn}}}\mathrm{err}(h) - \mathrm{err}(h^\star) = O(\varepsilon)$ and $\mathrm{VCdim}(\mathcal{H}_{\mathsf{dnn}}) = O\left(\varepsilon^{-\frac{2d}{(1+\beta)(d+3)}} \cdot \log(\varepsilon^{-1})\right)$. Take $\rho = 1$ in Theorem 11, leads to

$$\mathrm{err}(\widehat{h}) - \mathrm{err}(h^\star) \leq O\left(\varepsilon + \left(\varepsilon^{-\frac{2d}{(1+\beta)(d+3)}} \cdot \log(\varepsilon^{-1}) \cdot \frac{\log n}{n}\right)^{\frac{1+\beta}{2+\beta}} + \frac{\log\delta^{-1}}{n}\right),$$

Taking $n = O(\varepsilon^{-\frac{4d+6+\beta(d+3)}{(1+\beta)(d+3)}} \cdot \log(\varepsilon^{-1}) + \varepsilon^{-1} \cdot \log(\delta^{-1})) = \widetilde{O}(\varepsilon^{-\frac{4d+6+\beta(d+3)}{(1+\beta)(d+3)}})$ thus ensures that $\mathrm{err}(\widehat{h}) - \mathrm{err}(h^\star) = O(\varepsilon)$. $\qquad\square$

### G.3   Active learning results

**Theorem 10.** *Suppose $\eta \in \mathscr{R}\,\mathsf{BV}_1^2(\mathcal{X})$ and the Tsybakov noise condition is satisfied with parameter $\beta \geq 0$. Fix any $\varepsilon, \delta > 0$. There exists an algorithm such that, with probability at least $1 - \delta$, it learns a classifier $\widehat{h} \in \mathcal{H}_{\mathsf{dnn}}$ with excess error $\widetilde{O}(\varepsilon)$ after querying $\widetilde{O}(\theta_{\mathcal{H}_{\mathsf{dnn}}}(\varepsilon^{\frac{\beta}{1+\beta}}) \cdot \varepsilon^{-\frac{4d+6}{(1+\beta)(d+3)}})$ labels.*

*Proof.* Construct $\mathcal{H}_{\mathsf{dnn}}$ based on Proposition 13 such that $\min_{h\in\mathcal{H}_{\mathsf{dnn}}}\mathrm{err}(h) - \mathrm{err}(h^\star) = O(\varepsilon)$ and $\mathrm{VCdim}(\mathcal{H}_{\mathsf{dnn}}) = \widetilde{O}(\varepsilon^{-\frac{2d}{(1+\beta)(d+3)}})$. Taking such $\mathcal{H}_{\mathsf{dnn}}$ as the initialization of Algorithm 3 (line 1) and applying Theorem 12 leads to the desired result. $\qquad\square$

To derive deep active learning guarantee with abstention in the Radon $\mathsf{BV}^2$ space, we first present two supporting results below.

**Proposition 14.** *Suppose $\mathcal{D}_{\mathcal{X}\mathcal{Y}} \in \mathcal{P}(\beta)$. One can construct a set of neural network regression functions $\mathcal{F}_{\mathsf{dnn}}$ such that the following two properties hold simultaneously:*

$$\exists f \in \mathcal{F}_{\mathsf{dnn}}\ s.t.\ \|f - f^\star\|_\infty \leq \kappa, \quad and \quad \mathrm{Pdim}(\mathcal{F}_{\mathsf{dnn}}) \leq c \cdot \kappa^{-\frac{2d}{d+3}}\log^2(\kappa^{-1}),$$

*where $c > 0$ is a universal constant.*

*Proof.* The result follows by combining Theorem 9 and Theorem 13. $\qquad\square$

**Proposition 15.** *Suppose $\eta$ is $L$-Lipschitz and $\mathcal{X} \subseteq \mathbb{B}_r^d$. Fix any $\kappa \in (0, \gamma/32]$. There exists a set of neural network regression functions $\mathcal{F}_{\mathsf{dnn}}$ such that the followings hold simultaneously.*

*1. $\mathrm{Pdim}(\mathcal{F}_{\mathsf{dnn}}) \leq c \cdot \kappa^{-\frac{2d}{d+3}}\log^2(\kappa^{-1})$ with a universal constant $c > 0$.*

*2. There exists a $\overline{f} \in \mathcal{F}_{\mathsf{dnn}}$ such that $\|\overline{f} - \eta\|_\infty \leq \kappa$.*

*3. $\theta_{\mathcal{F}_{\mathsf{dnn}}}^{\mathrm{val}}(\gamma/4) := \sup_{f\in\mathcal{F}_{\mathsf{dnn}}, \iota>0} \theta_f^{\mathrm{val}}(\mathcal{F}_{\mathsf{dnn}}, \gamma/4, \iota) \leq c' \cdot (\frac{Lr}{\gamma})^d$ with a universal constant $c' > 0$.*

*Proof.* The implementation and proof are similar to those in Proposition 9, except we use Proposition 14 instead of Proposition 6. $\qquad\square$

We now state and prove deep active learning guarantees in the Radon $\mathsf{BV}^2$ space.

**Theorem 18.** *Suppose $\eta \in \mathscr{R}\,\mathsf{BV}_1^2(\mathcal{X})$. Fix any $\varepsilon, \delta, \gamma > 0$. There exists an algorithm such that, with probability at least $1 - \delta$, it learns a classifier $\widehat{h}$ with Chow's excess error $\widetilde{O}(\varepsilon)$ after querying $\mathrm{poly}(\frac{1}{\gamma}) \cdot \mathrm{polylog}(\frac{1}{\varepsilon\delta})$ labels.*

*Proof.* The result is obtained by applying Algorithm 4 with line 1 be the set of neural networks $\mathcal{F}_{\mathsf{dnn}}$ generated from Proposition 15 with approximation level $\kappa \in (0, \gamma/32]$ (and constants $c, c'$ specified therein). The rest of the proof proceeds in a similar way as the proof Theorem 7. Since we have $r = 1$ and $L \leq 1$ (Parhi and Nowak, 2022b), we only need to choose a $\kappa > 0$ such that

$$\frac{1}{\kappa} = \check{c} \cdot \left(\frac{1}{\gamma}\right)^{\frac{d}{2}+1} \cdot \log\frac{1}{\varepsilon\gamma}$$

with a universal constant $\check{c} > 0$. With such choice of $\kappa$, we have

$$\mathrm{Pdim}(\mathcal{F}_{\mathsf{dnn}}) = O\left(\left(\frac{1}{\gamma}\right)^{\frac{d^2+2d}{d+3}} \mathrm{polylog}\left(\frac{1}{\varepsilon\,\gamma}\right)\right).$$

Plugging this bound on $\mathrm{Pdim}(\mathcal{F}_{\mathsf{dnn}})$ and the upper bound on $\theta^{\mathrm{val}}_{\mathcal{F}_{\mathsf{dnn}}}(\gamma/4)$ from Proposition 15 into the guarantee of Theorem 14 leads to $\mathrm{excess}_\gamma(\widehat{h}) = O(\varepsilon \cdot \log(\frac{1}{\varepsilon\,\gamma\,\delta}))$ after querying

$$O\left(\left(\frac{1}{\gamma}\right)^{d+2+\frac{d^2+2d}{d+3}} \cdot \mathrm{polylog}\left(\frac{1}{\varepsilon\,\gamma\,\delta}\right)\right)$$

labels. $\qquad\square$