# OpenReview forum: "Active Learning with Neural Networks: Insights from Nonparametric Statistics"
_NeurIPS.cc/2022/Conference — NeurIPS 2022 Accept_

### Official Review · Reviewer_ynw4 · 2022-07-08

**Rating:** 7
**Confidence:** 3
**Soundness:** 3 good
**Presentation:** 4 excellent
**Contribution:** 3 good

**Summary:**


The authors investigate the label complexity guarantees for deep active learning from the nonparametric classification perspective. They first look into the RobustCAL algorithm which only queries data points lying in the region of disagreement and provides the near minimax optimal label complexity, bringing closer the gap between theory and practice of deep active learning. Next they replace the low noise condition with abstention and Chow’s excess error, which enables exponential speedups and leads to sharper concentration results. Last, they conclude with extensions from Sobolev/H\"{o}lder space to Radon BV^2 space.

**Questions:**

I am not entirely familiar with the field, but I enjoy reading the paper. I wonder how the theorem can be carried over to implementation like how to decide whether to implement abstention, and like how to verify the conditions/assumptions of the theorems.

**Limitations:**

The current work mainly focuses on classification with binary labels. And the main theorems have dependence on the disagreement coefficient or Chow’s errors (the authors identify as well).

**Strengths And Weaknesses:**

The authors provide a detailed overview of related literature in the field. The presentation of the paper is clear and easy to follow. The contributions are theoretical. Theorem 6 provides near optimal label complexity up to disagreement coefficients and logarithmic factors. Theorem 7 provides good theoretical intuition for why abstention is efficient in practice.

---

> ### Author Response · Authors · 2022-08-02
> **Response to Reviewer ynw4**
>
> Thank you for the positive comments. Please see our response to your question below.
>
>
> > I am not entirely familiar with the field, but I enjoy reading the paper. I wonder how the theorem can be carried over to implementation like how to decide whether to implement abstention, and like how to verify the conditions/assumptions of the theorems.
>
> **Response:** We recommend implementing the algorithm with abstention (i.e., Algorithm 2) in practice due to the following three reasons: (1) Abstention helps avoid the noise-seeking behavior of the active learner (i.e., over-sampling from the high-noise regions) and thus helps achieve exponential speedups. (2) Under certain low noise conditions (e.g., Massart noise), Algorithm 2 also provides exponential label savings under standard excess error (see Proposition 12 on page 32 of supplementary material). (3) Algorithm 2 can be implemented with square loss regression wrt neural networks (which can be approximated using stochastic gradient descent in practice). However, Algorithm 1 (and many other existing standard active learning algorithms) requires conducting empirical risk minimization (ERM) wrt 0/1 loss (which is known to be NP-hard even in the case with linear functions) and characterization of the region of disagreement (which is also computationally challenging).
>
> The main assumptions used in our paper are (1) the true regression function $f^{\star}$ is smooth (e.g., it belongs to Sobolev spaces or Radon $\textsf{BV}^{2}$ spaces) and (2) the label noise satisfies some low noise conditions (not needed for Algorithm 2). Both assumptions are standard in the literature on active learning and are usually verified with domain knowledge (e.g., for particular applications, domain experts can generally shed light on the typical/approximate smoothness level of the true regression function and the typical/approximate noise level).

---

> > ### Comment · Reviewer_ynw4 · 2022-08-09
> > **Thank you for your response**
> >
> > My concerns have been addressed. The rating remains the same.

---

### Official Review · Reviewer_qadK · 2022-07-10

**Rating:** 7
**Confidence:** 4
**Soundness:** 4 excellent
**Presentation:** 3 good
**Contribution:** 3 good

**Summary:**

This paper shows that active learning with neural networks can provably achieve the minimax label complexity, up to disagreement coefficient, and other logarithmic terms under the assumption that the neural network is obtained from $\alpha$-Sobolev space. Then the authors try to generalize the idea up to the Radon $BV^2$ space.

As claimed by the authors, the main contributions of the paper are **1)** to extend the understanding of how approximation affects active learning to achieve fast rates (e.g., under Tsybakov noise in Theorem 1) which corresponds to Theorem 6, and **2)** to identify conditions that greatly relax the (previous) approximation requirement in the learning with abstention setup (to develop Theorem 2) which corresponds to Theorem 7.

I think the key component of the paper is building a bridge by extending the active learning analysis in the parametric setting to the non-parametric DNN setting leveraging the universal approximation property of the ReLU activated neural networks.

**Questions:**

This paper contains a lot of materials from existing literature, so contains a long proof to digest. I still have some questions to clarify. Once questions are clarified, I would rather consider updating my rating. Thank you.

1. In algorithm 2, why $\beta_m$ is chosen to be adaptively decreasing? This implies that the abstention region seems to be adaptively changing(decreasing) as the epoch $m$ increases. How is this decreasing $\beta_m$ facilitating the exponential gain toward the label complexity in Algorithm 2? I was not able to grasp the full intuition.
2. How is the geometrically increasing query length contributing to the exponential gain of the label complexity? In other words, what's the motivation of the geometrically increasing query length? And how is it related to the decreasing $\beta_m$? I understand it is necessary for analysis, but the cost of the geometrically increasing query seems to be expensive (at least for making a decision if it is under abstention or not) unless it produces an exponentially small size of the non-abstention selection for a real label query. I see some relation at Lemma 10, but the decreasing abstention region (as described in Question 1 above) seems to be conflicting with the geometrically increasing number of points in the abstention region because of the geometrically increasing query length.
3. At L955 of Appendix D5, the condition described in the equation $(19)$ seems to be important as it is also used in the proof of Theorem 7. But I am not able to find where the condition $(19)$ has been established in the proof of Theorem 15. Can you elaborate on where the equation $(19)$ has been established/or used in the proof? (I suspect L965, but it's not sufficient.)
4. In practice, the diversification of point selection is essential to improve active learning performance.  As far as I see, the Tsybakov noise condition controls the number of ambiguous (and redundant) points in the input domain. For exponential speed up, in Theorem 7, the parameter $\beta$ does not appear anymore. It might be embedded somewhere in the proof, but I was not able to find it. Can you explain the relationship between the Tsybakov noise parameter $\beta$ and the number of queries in Theorem 7? Intuitively, if the redundant points are concentrated around **the outside of the abstention region** but still inside the query range, algorithm 2 might be failed. I was wondering where to avoid such a case.
5. At L858, I was wondering if $M_t(f)$ is still a martingale with abstention option. In other words, I want to understand the relationship between the martingale condition and the exponential gain of the label complexity. If it is a martingale, the active learning process is a fair game in some sense. How is it possible to achieve the exponential speed-up under this martingale regime?

---------------------------------------------------------------------------------------------------------------------------------------------------
After rebuttal, the authors clarified all my questions properly.

Because of the limited review process time, I cannot assure to confirm the whole correctness of the paper by myself.
However, as a collective effort to confirm the result, I do not see any serious issues or problems in this paper.
Accordingly, I updated my rating given that
- the closely related work [1] needs to be confirmed officially because I see lots of overlapped material with the paper [1] as cited by the authors,
- the authors revise the paper properly.

Thank you.

[1] Zhu, Yinglun, and Robert Nowak. "Efficient Active Learning with Abstention." arXiv preprint arXiv:2204.00043 (2022).

**Limitations:**

1. In practice, geometrically increasing query length might be unrealistic (with/without abstention). Practical active learning typically assumes a constant size (or budget-constrained size) of the query for each epoch. I was wondering if there is a way to improve both the algorithm and the analysis with a constant query size.
2. As mentioned, it is not straightforward how to generalize the idea to the multi-class classification problem.

**Strengths And Weaknesses:**

** Strength
1. This paper provides an extensive theoretical insight into how deep active learning could be successful in the binary classification problem.
2. This paper proves the exponential gain of the label complexity in deep active learning with the abstention option.

** Weakness
1. The paper focuses on the binary classification problem, which is the fundamental case to build necessary theoretical tools. Still, it doesn't seem to be straightforward how to generalize the insight to the multi-class classification problems when we apply the proposed algorithm in practice.
2. The presentation could be improved. The non-parametric setting mainly focuses on the $\alpha$-Sobolev space (weakly differentiable space up to $\alpha$-th order) in the main article. But after reading the whole paper including the supplementary material, Radon $BV^2$ space is a better function space to consider. Because the contents of the main article are heavily relying on the supplementary material, it might be better to re-organize the paper by focusing on the Radon $BV^2$ space instead of the $\alpha$-Sobolev space. Then all relevant results from $\alpha$-Sobolev can be moved to the supplementary material.

** Minor points
1. At the footprint on page 3, "exponentially my sub-regions" should be "exponentially many sub-regions".
2. At Algorithm 2, $Pdim(\cdot)$ doesn't seem to be defined in the main article. I was able to locate it at L749 of Definition 6 in the supplementary material.
3. At L785, I cannot find the definition of $\mathcal{N}(\frac{\kappa'}{8L};\mathcal{X},\|\cdot\|_2)$. I understand it is the covering (or sphere-packing) number, but the definition seems to be missing.
4. At L793, balls of the radius $\frac{\kappa}{8L}$ should be $\frac{\bar{\kappa}}{8L}$.
5. At L807, Proposition 8 of Appendix D.1.3, the second condition should be $0$ if $f(x)\leq 0$ instead of $-1$ if $f(x)\geq 1$.
6. At L835 of Appendix D.2, $B_{1}^{d}$ should be $B_{r}^{d}$.
7. At L914 of Appendix D.4.2, $(a+b)^2\leq a^2+b^2$ should be $(a+b)^2 \leq 2(a^2+b^2)$.
8. At L1046 of Appendix E.2.2., the notation $\alpha$ is conflicting with the notion $\alpha$-Sobolev space. As far as I understand, this $\alpha$ in L1046 has nothing to do with $\alpha$-th order weak differentiability. Please reconsider using another letter in either case.
9. At L1057, $\bar{a}:=\frac{\gamma}{8M}$ should be $\bar{\alpha}:=\frac{\gamma}{8M}$.

---

> ### Author Response · Authors · 2022-08-02
> **Response to Reviewer qadK (1/3)**
>
> Thank you for your detailed review. We hope our responses below can help resolve your concerns.
>
> ### **Responses to Questions:**
>
> 1. > In algorithm 2, why $\beta_m$ is chosen to be adaptively decreasing? This implies that the abstention region seems to be adaptively changing(decreasing) as the epoch $m$ increases. How is this decreasing $\beta_m$ facilitating the exponential gain toward the label complexity in Algorithm 2? I was not able to grasp the full intuition.
>
> **Response:** We choose a decreasing $\beta_m$ to automatically ensure ${\cal F_{ m}} \subseteq {\cal F_{{m}\text{-1}}}$, i.e., the 5th statement in Lemma 9 (at line 898 and its proofs at lines 916-919) without adding a hard constraint ${\cal F_{ m}} \subseteq {\cal F_{{m}\text{-1}}}$. Since the choice of  $\beta_m$ ensures that ${\cal F_{ m}} \subseteq {\cal F_{{m}\text{-1}}}$, the abstention region (i.e., {$ x \in {\cal X}: \widehat h_m(x) = \bot$}) is actually **increasing** as  $m$ increases (due to construction of $\widehat h_m(x)$ at line 7 of Algorithm 2). Since labels are not queried whenever the classifier abstains (also see the construction $g_m$ at line 7 of Algorithm 2), this **decreases** the probability of issuing a label query, which eventually helps achieves exponential gains in label complexity.
>
> 2. > How is the geometrically increasing query length contributing to the exponential gain of the label complexity? In other words, what's the motivation of the geometrically increasing query length? And how is it related to the decreasing $\beta_m$? I understand it is necessary for analysis, but the cost of the geometrically increasing query seems to be expensive (at least for making a decision if it is under abstention or not) unless it produces an exponentially small size of the non-abstention selection for a real label query. I see some relation at Lemma 10, but the decreasing abstention region (as described in Question 1 above) seems to be conflicting with the geometrically increasing number of points in the abstention region because of the geometrically increasing query length.
>
> **Response:** The geometrically increasing epoch length is not a problem **since the probability of issuing a label query is indeed geometrically decreasing**. Note that there is no conflict: As emphasized in our response to your first question, the abstention region is increasing, and the probability of issuing a label query is decreasing. More specifically, the probability of issuing a label query at epoch $m$ can be upper bounded by $O(\frac{\beta_m \bar \theta}{\tau_{m-1} \gamma^{2}})$ (recall that $\tau_m = 2^{m}$); we thus can bound the total number of labels queried in epoch $m$ by $\tau_m \cdot O(\frac{\beta_m \bar \theta}{\tau_{m-1} \gamma^{2}}) = O(\frac{\beta_m \bar \theta}{\gamma^{2}})$. The geometrically increasing epoch length ensures that there are at most $M = O(\log T)$ epochs, which further leads to label complexity  $ \sum_{m=1}^{M} \frac{\beta_m \bar \theta}{\gamma^{2}} = \widetilde O(\frac{ \textsf{polylog}(T) \cdot \textsf{Pdim}({\cal F}) \bar \theta}{\gamma^{2}})$ (this idea is formalized in the proof of Theorem 15, on pages 29-30 of supplementary material). By properly controlling the disagreement coefficient $\bar \theta$ and the Pseudo dimension  $\textsf{Pdim}({\cal F})$ (see proof of Theorem 7 on page 31 of supplementary material for details), we derive the desired $ \textsf{poly} (\frac{1}{\gamma}) \cdot \textsf{polylog}(\frac{1}{\epsilon}) $ label complexity for Theorem 7.
>
> 3. > At L955 of Appendix D5, the condition described in the equation (19) seems to be important as it is also used in the proof of Theorem 7. But I am not able to find where the condition (19) has been established in the proof of Theorem 15. Can you elaborate on where the equation (19) has been established/or used in the proof? (I suspect L965, but it's not sufficient.)
>
> **Response:** Eq. (19) is mainly used to derive the first statement in Lemma 9 (i.e., $\widehat R_m(\bar f) - \widehat R_m(f^{\star}) \leq \frac{3}{2} C_\delta$, proved at lines 903-905). Lemma 9 is essential in deriving Lemmas 10-12 (in an inductive way) and eventually Theorem 15.
>
> **Continued in the next response.**

---

> > ### Author Response · Authors · 2022-08-02
> > **Response to Reviewer qadK (2/3)**
> >
> > ### **Responses to Questions (Cont'd):**
> >
> > 4. > In practice, the diversification of point selection is essential to improve active learning performance. As far as I see, the Tsybakov noise condition controls the number of ambiguous (and redundant) points in the input domain. For exponential speed up, in Theorem 7, the parameter  does not appear anymore. It might be embedded somewhere in the proof, but I was not able to find it. Can you explain the relationship between the Tsybakov noise parameter  and the number of queries in Theorem 7? Intuitively, if the redundant points are concentrated around **the outside of the abstention region** but still inside the query range, algorithm 2 might be failed. I was wondering where to avoid such a case.
> >
> > **Response:** Theorem 7 is established **without any low noise assumption**, but instead under Chow's excess error defined at line 295. It is known that the minimax lower bound under the standard excess error and Tsybakov noise scales as $\Omega(\epsilon^{-\frac{d+2\alpha}{\alpha + \alpha \beta}})$  (Locatelli et al. 2017), which is polynomial in $\frac{1}{\epsilon}$. With Chow's excess error, we achieve label complexity that scales as $\textsf{polylog}(\frac{1}{\epsilon})$, which is an exponential improvement (and has no dependence on $\beta$). We achieve a similar exponential improvement under standard excess error and Massart noise, and the noise parameter appears in the label complexity in this case (see Proposition 12, at lines 1030-1032).
> >
> > 5. > At L858, I was wondering if $M_t(f)$ is still a martingale with abstention option. In other words, I want to understand the relationship between the martingale condition and the exponential gain of the label complexity. If it is a martingale, the active learning process is a fair game in some sense. How is it possible to achieve the exponential speed-up under this martingale regime?
> >
> > **Response:** With notations defined at lines 857-860, the martingale used to prove Lemma 6 is $ \sum_{t}^{} M_t(f) - {\mathbb E_t} [M_t(f)]$, which is a legit martingale even with query variable $Q_t$: Since  $(M_t(f) - {\mathbb E_t} [M_t(f)])_t$ is a martingale difference sequence **by construction**, i.e., ${\mathbb E_t} [M_t(f) - {\mathbb E_t} [M_t(f)] ] = {\mathbb E_t} [M_t(f)] - {\mathbb E_t} [M_t(f)] = 0$. Such martingale is only used to show that the empirical square error difference (i.e., $ \sum_t M_t(f) $) concentrates around its expected version (i.e., $\sum_t {\mathbb E_t} [M_t (f)] $, see Lemma 6 for precise statements). The martingale concentration results are not directly related to the exponential improvement of label complexity, but are mainly used to prove Lemma 9 (see pages 25-27 of supplementary material). In our response to your second question (in response (1/3) posted before this one), we sketch the key idea behind the exponential saving; we prove the formal guarantees in Theorem 15 (pages 29-30 of supplementary material).
> >
> > **Continued in the next response.**

---

> > > ### Author Response · Authors · 2022-08-02
> > > **Response to Reviewer qadK (3/3)**
> > >
> > > ### **Responses to Weaknesses:**
> > >
> > > 1. >The paper focuses on the binary classification problem, which is the fundamental case to build necessary theoretical tools. Still, it doesn't seem to be straightforward how to generalize the insight to the multi-class classification problems when we apply the proposed algorithm in practice.
> > >
> > > **Response:** Yes, the main focus of the current paper is to establish theoretical guarantees for active learning with neural networks in the binary classification case (which is fundamental). We leave the study of multi-class for future work.
> > >
> > > 2. > The presentation could be improved. The non-parametric setting mainly focuses on the $\alpha$-Sobolev space (weakly differentiable space up to $\alpha$-th order) in the main article. But after reading the whole paper including the supplementary material, Radon $\textsf{BV}^{2}$ space is a better function space to consider. Because the contents of the main article are heavily relying on the supplementary material, it might be better to re-organize the paper by focusing on the Radon $\textsf{BV}^{2}$ space instead of the $\alpha$-Sobolev space. Then all relevant results from $\alpha$-Sobolev can be moved to the supplementary material.
> > >
> > > **Response:** While Radon $\textsf{BV}^{2}$ space is a natural space to analyze shallow neural networks, we present our main results for functions in Sobolev spaces since that is the standard/classical setting studied in the literature of nonparametric active learning. Our main contribution is establishing a bridge between approximation theory and active learning: Our algorithms/analyses provide active learning guarantees as long as one can provide appropriate neural network approximation guarantees (in whatever spaces). In fact, we provide such general guarantees in Appendix B for standard excess error (see pages 15-18 of supplementary material) and Appendix D for Chow's excess error (see pages 20-30 of supplementary material). We plan to further highlight this contribution in the revision.
> > >
> > > ---
> > >
> > > ### **Responses to Minor Points:**
> > >
> > >
> > > 2. > At Algorithm 2, $\textsf{Pdim}(\cdot)$ doesn't seem to be defined in the main article. I was able to locate it at L749 of Definition 6 in the supplementary material.''
> > >
> > > **Response:** Thank you for pointing this out. In the revision, we'll define/introduce the Pseudo dimension $\textsf{Pdim}({\cal F})$ in the main content as well.
> > >
> > > 3. > At L785, I cannot find the definition of ${\cal N}(\frac{\kappa^{\prime}}{8L}; {\cal X}, \|\cdot\|_2)$. I understand it is the covering (or sphere-packing) number, but the definition seems to be missing.
> > >
> > > **Response:** We use the (relatively standard) notation ${\cal N}(\epsilon; {\cal X}, \|\cdot\|_2)$ to denote the $\epsilon$-covering number of  ${\cal X}$ under metric  $\|\cdot \|_2$. We apologize for any confusion and will add an explicit definition in the revision.
> > >
> > > 7. > At L914 of Appendix D.4.2, $(a+b)^{2} \leq a^{2} + b^{2}$ should be $(a+b)^{2} \leq 2(a^{2} + b^{2})$.
> > >
> > > **Response:** Yes, this is a typo. The inequality, however, is correctly applied on the second line of the display before line 914: We have the constant $2$ in the inequality.
> > >
> > > 8. > At L1046 of Appendix E.2.2., the notation $\alpha$ is conflicting with the notion $\alpha$-Sobolev space. As far as I understand, this $\alpha$ in L1046 has nothing to do with $\alpha$-th order weak differentiability. Please reconsider using another letter in either case.''
> > >
> > > **Response:** Yes, the notation $\alpha$ at line 1045/1046 represents the approximation error but not the smoothness parameter. We'll change it to another notation in the revision to remove potential confusion.
> > >
> > >
> > > > Minor points 1, 4, 5, 6, 9 (regarding typos).
> > >
> > > **Response:** Thank you for catching these typos. We will correct them in the revision.

---

> > > > ### Comment · Reviewer_qadK · 2022-08-06
> > > > **Thank you very much for your detailed answers.**
> > > >
> > > > Thank you very much for your detailed answers. All my questions are resolved properly.
> > > >
> > > > For example,
> > > > - I also confirm that in the abstract, $\text{ploylog}(\frac{1}{\epsilon})$ label complexity is possible without low noise assumptions.
> > > > - Equation (19) is the assumption for the lines of the proof in Lemma 9.
> > > > - The martingale condition is related to the proof of Theorem 15, which is the key part for exponential savings.
> > > >
> > > > I will update my rating accordingly.
> > > >
> > > > This is a fantastic paper. I appreciate the authors having this research work. But it's somewhat difficult to follow the entire flow for non-expert readers in this domain. To improve readability further, I suggest considering to
> > > > - add *a diagram* that shows the relationship between Lemmas and Theorems,
> > > > - add a *conclusion* and *limitations* of this work (even in the supplementary material if you cannot find any further space).

---

> > > > > ### Author Response · Authors · 2022-08-06
> > > > > **Thanks for your reply**
> > > > >
> > > > > Thank you very much for your prompt response! We will take your suggestions/comments into serious consideration and properly revise our paper to make it more readable.

---

### Official Review · Reviewer_TeP9 · 2022-07-13

**Rating:** 6
**Confidence:** 3
**Soundness:** 2 fair
**Presentation:** 2 fair
**Contribution:** 3 good

**Summary:**

This paper studies the deep active learning problem and focuses on the rigorous label complexity guarantees. Deep active learning has received practical success, however, there lack of rigorous theoretical guarantees of deep active learning. In this paper, they consider a nonparametric setting. They show the relationship between approximation and active learning. They also give an analysis of the disagreement coefficient.

**Questions:**

 - Their paper has a lot of results. Which one is the most significant result?

**Limitations:**

 - their result relies on the disagreement coefficient, which does not have explicit expression.

**Strengths And Weaknesses:**

Pros:
 - Their result is easy to extend and can be applied to a large class of smooth functions that have recently been proposed. Their result can be applied to ReLU neural networks.
 - In both passive and active settings, they can learn by the neural network to achieve near minimax optimal label complexities, which gives a new perspective on nonparametric learning

Cons:
 - Their result is meaningful when the disagreement coefficient is well-bounded. However, they left the analysis of the disagreement coefficient for future work. The disagreement coefficient does not have an explicit expression, which makes it hard to judge how important their results are.

---

> ### Author Response · Authors · 2022-08-02
> **Response to Reviewer TeP9**
>
> Thank you for your positive review. We hope our responses below can help resolve your concerns.
>
> 1. > Their result is meaningful when the disagreement coefficient is well-bounded. However, they left the analysis of the disagreement coefficient for future work. The disagreement coefficient does not have an explicit expression, which makes it hard to judge how important their results are.
>
> **Response:** We theoretically analyze both the classifier-based disagreement coefficient (for Section 2) and the value function-based disagreement coefficient (for Section 3). Although we left a comprehensive analysis of the classifier-based disagreement coefficient for future work, we provide the following label complexity guarantees based on our current analyses of the disagreement coefficient.
>
> Under standard excess error (Section 2), our label complexity is never worse than passive learning even with a trivial bound on the disagreement coefficient; when there exist additional structures, we further bound the disagreement coefficient by $\theta_{{\cal H_{\textsf{dnn}}}} ( \epsilon) = o(\epsilon^{-1})$ (implying strict improvement over passive learning) and  $\theta_{{\cal H_{\textsf{dnn}}}} ( \epsilon) = O(1)$ (implying matching the active learning lower bound). Please see lines 255-262 and Appendix C.2 (page 18 of supplementary material) for a detailed discussion.
>
> Under Chow's excess error (Section 3), we provably upper bound the disagreement coefficient by $O(\textsf{poly}(\frac{1}{\gamma}))$ (see Proposition 9 on page 22 of supplementary material), which is a key step to achieve exponential savings in the label complexity (i.e., the $\textsf{polylog}(\frac{1}{\epsilon})$ label complexity in Theorem 7). In the revision, we plan to highlight this bound on disagreement coefficient in the main content.
>
> 2. > Their paper has a lot of results. Which one is the most significant result?
>
> **Response:** We summarize our main contributions in Section 1.2, i.e., label complexity guarantees for active learning with neural networks under both standard excess error (Theorem 1, the informal version of Theorem 6) and Chow's excess error (Theorem 2, the informal version of Theorem 7). To the best of our knowledge, Theorem 1 (Theorem 6) provides the first near minimax optimal guarantees for deep active learning, and Theorem 2 (Theorem 7) provides the first guarantee with exponential savings for deep active learning.
>
> More generally, we establish a bridge from approximation theory to active learning: We provide the general guarantees under standard excess error in Appendix B (pages 15-18 of supplementary material) and the general guarantees under Chow's excess error in Appendix D (pages 20-30 of supplementary material). We plan to further highlight this contribution in the revision.

---

> > ### Author Response · Authors · 2022-08-08
> > **Following up**
> >
> > Thanks again for your thoughtful review. We hope our rebuttal addresses the points you raised. Please let us know if there is anything else we can clarify.

---

### Official Review · Reviewer_MozJ · 2022-07-13

**Rating:** 6
**Confidence:** 3
**Soundness:** 3 good
**Presentation:** 2 fair
**Contribution:** 3 good

**Summary:**

This paper provides the first near minimax optimal label complexity guarantees for deep active learning.
Section 2 analyzes the label complexity of deep active learning under Tsybakov noise. Section 3 analyzes it under another setting with an abstention option and without any low noise assumption. Section 4 extends the results to Radon BV2 space.

**Questions:**

1. In line 202 under Theorem 3, the paper says "The architecture of the neural network $f_{dnn}$ appearing in the above theorem is independent of $f^*$". Could the authors elaborate more on this conclusion? Intuitively, given a function $f^*$, why is the architecture of the neural network that can best approximate $f^*$ within a small error **independent** of $f^*$?
2. Following the above question, since both $H_{dnn}$ and $F_{dnn}$ are specific sets of neural networks defined based on the architecture of $f_{dnn}$, the problem setting becomes very limited if $f_{dnn}$ is chosen based on the true function $f^*$.
3. How important is the smooth assumption on $\eta$? Section 4 demonstrates how the results can be extended to Radon BV2 space. Can we assume that the results can be extended to any space as long as the approximation power of the neural networks can be proved?
4. Section 3 states that the results are generalized from that of Zhivotovskiy (2021); Zhu and Nowak (2022). Line 316-323 presents how this paper is new compared to Zhu and Nowak (2022). How about the work of Zhivotovskiy (2021)? How is section 3 technically novel compared to Zhivotovskiy (2021)?
5. Any numerical simulations?
6. There are many grammar errors:
- line 181: exhibit -> exhibits
- line 212: has well-controlled -> has a well-controlled
- line 287: analyze -> analyzes
- line 323: function -> functions
- ...

**Limitations:**



**Strengths And Weaknesses:**

Strength:
1. This paper contributes new theoretical results to the area of deep active learning.
2. The proof techniques could be potentially useful for the theoretical study in this area. I like how the results do not require the Bayes classifier to be in the hypothesis class. Instead, it only requires a good approximation to be in the hypothesis class.
3. A new algorithm with theoretical support has been proposed.

Weakness:
1. The paper does not have a conclusion section.
2. No experiment is provided to support the claims. Especially, a new algorithm (Algorithm 2) is proposed, but no numerical simulation is demonstrated.
3. The limitation of the chosen set of neural networks $H_{dnn}$ and $F_{dnn}$ has not been discussed.
4. Results in Section 2 and Section 3 appear to be a bit disconnected.

---

> ### Author Response · Authors · 2022-08-02
> **Response to Reviewer MozJ (1/2)**
>
> Thank you for your positive review. We hope our responses below can help resolve your concerns.
>
> ### **Responses to Questions:**
>
> 1. >In line 202 under Theorem 3, the paper says ``The architecture of the neural network $f_{\textsf{dnn}}$ appearing in the above theorem is independent of $f^{\star}$ ''. Could the authors elaborate more on this conclusion? Intuitively, given a function $f^{\star}$, why is the architecture of the neural network that can best approximate $f^{\star}$ within a small error **independent** of $f^{\star}$?
>
> **Response:**
> We consider any **unknown** $f^\star$ belonging to a **known** smooth function space ${\cal W_{\text 1}^{\alpha,\infty}} ([-1,1]^{d})$ (knowing the function space is a standard assumption).
> Fix any user-specified approximation error $\kappa > 0$, the architecture of (a good approximating neural network) $f_{\textsf{dnn}}$ is determined by smoothness parameter $\alpha$, dimension $d$, and the desired approximation error $\kappa$ (see Theorem 3 for the specific architecture), but is otherwise completely independent of the **unknown** $f^\star$.
>
> We also want to clarify that the architecture of a neural network only specifies the depth/width and the total number of parameters, but not the exact parameters. In other words, we know there exists a $f_{\textsf{dnn}}$ with specific architecture that approximates $f^\star$ well, but we don't know the exact parameters of $f_{\textsf{dnn}}$: We need to learn the parameters of $f_{\textsf{dnn}}$ by ourselves.
>
> 2. > Following the above question, since both ${\mathcal H_{\textsf{dnn}}}$ and ${\cal F_{\textsf{dnn}}}$ are specific sets of neural networks defined based on the architecture of $f_{\textsf{dnn}}$, the problem setting becomes very limited if $f_{\textsf{dnn}}$ is chosen based on the true function $f^{\star}$.
>
> **Response:** Please see our response to the above question. Our problem setting is not limited since the smoothness parameter $\alpha$ and the dimension $d$ are generally considered known parameters, and the approximation error $\kappa$ is chosen by our algorithms (based on the desired accuracy level $\epsilon$). ${\mathcal F}_{\textsf{dnn}}$ simply contains all possible neural networks with the same architecture (e.g., all 3-layer fully-connected neural networks with 100 total parameters).
>
> 3. > How important is the smooth assumption on $\eta$? Section 4 demonstrates how the results can be extended to Radon BV2 space. Can we assume that the results can be extended to any space as long as the approximation power of the neural networks can be proved?
>
> **Response:** This is a great observation. Our algorithms/analyses provide active learning guarantees as long as one can provide appropriate neural network approximation guarantees. In other words, we establish a bridge between approximation theory and active learning. We provide such general guarantees in Appendix B for standard excess error (see pages 15-18 of supplementary material) and Appendix D for Chow's excess error (see pages 20-30 of supplementary material). We plan to further emphasize this contribution in the revision.
>
> 4. > Section 3 states that the results are generalized from that of Zhivotovskiy (2021); Zhu and Nowak (2022). Line 316-323 presents how this paper is new compared to Zhu and Nowak (2022). How about the work of Zhivotovskiy (2021)? How is section 3 technically novel compared to Zhivotovskiy (2021)?
>
> **Response:** Our algorithm is adapted from Zhu and Nowak (2022), so we highlight the novelty compared to Zhu and Nowak (2022) on lines 316-323. Similar advantages carried over when compared to the work of Puchkin and Zhivotovskiy (2021): We can provably bound the label complexity by $\textsf{polylog}(\frac{1}{\epsilon})$ yet their label complexity can scale as $\textsf{poly}(\frac{1}{\epsilon})$ due to the use of classifier-based disagreement coefficient (even under Chow's excess error)---this demonstrates an exponential improvement in label complexity over existing approaches. Our algorithm relies on fundamentally different techniques/analyses from the one in Puchkin and Zhivotovskiy (2021): Their algorithm relies on empirical risk minimization (ERM) on 0/1 loss (which is NP-hard even for linear functions), yet we can implement our algorithm using efficient regression oracle (see line 334-336 for detailed discussion).
>
> 5. > Any numerical simulations?
>
> **Response:** We agree that empirically examining the proposed algorithms is an interesting future direction. However, as it stands, our contributions are theoretical, and we would like them to be viewed as such.
>
> 6. > There are many grammar errors: ...
>
> **Response:** Thanks for catching these typos. We will correct them in the revision.
>
> **Continued in the next response.**

---

> > ### Author Response · Authors · 2022-08-02
> > **Response to Reviewer MozJ (2/2)**
> >
> > ### **Responses to Weaknesses:**
> >
> > 1. > The paper does not have a conclusion section.
> >
> > **Response:** We summarize our main contributions in Section 1.2 to give readers an overview of the whole paper. However, we are happy to add a conclusion/discussion section in the revision.
> >
> > 2. > No experiment is provided to support the claims. Especially, a new algorithm (Algorithm 2) is proposed, but no numerical simulation is demonstrated.
> >
> > **Response:** Please see our response to point 5 in the **Responses to Questions** section (in the first response).
> >
> >
> > 3. > The limitation of the chosen set of neural networks ${\mathcal H_{\textsf{dnn}}}$ and ${\mathcal F_{\textsf{dnn}}}$ has not been discussed.
> >
> > **Response:** Please see our responses to points 1 and 2 in the **Responses to Questions** section (in the first response).
> >
> >
> > 4. > Results in Section 2 and Section 3 appear to be a bit disconnected.
> >
> > **Response:** We aim to provide a theoretical understanding of active learning with neural networks. Both Section 2 and Section 3 contribute to this goal: Section 2 analyzes the label complexity under the standard excess error (where we nearly match the minimax lower bound), and Section 3 analyzes the label complexity under Chow's excess error (where we provably achieve exponential label savings). We will add clarifications in the revision.

---

> > > ### Author Response · Authors · 2022-08-08
> > > **Following up**
> > >
> > > Thanks again for your thoughtful review. We hope our rebuttal addresses the points you raised. Please let us know if there is anything else we can clarify.

---

### Meta-Review · Area_Chair_uRY6 · 2022-08-27

**Recommendation:** Accept
**Confidence:** Certain

**Metareview:**

This paper presents two contributions in the area of deep active learning: 1) minimax optimal label complexity analysis (under low noise conditions), and 2) an exponentially improved label complexity without low noise conditions provided an abstention option is provided.

This is a theory paper and all the reviewers agreed that this paper presents novel results and significant improvement over prior work. The paper is recommended for acceptance. However the authors should consider the reviewers' comments about improving the readability of the paper.

**Award:**

No

---

### Decision · Program_Chairs · 2022-09-14

Accept